# Wasserstein Distributionally Robust Minimax Regret Optimization for Multimodal Machine Learning

## Abstract

Learning robust multimodal predictors under distributional uncertainty remains challenging, as empirical risk minimization (ERM) is brittle to modality-specific perturbations and standard distributionally robust optimization (DRO), by minimizing worst-case risk, may yield overly conservative solutions under heterogeneous noise. We introduce **Wasserstein Distributionally Robust Minimax Regret Optimization (WDRO-MRO)**, a framework that unifies Wasserstein DRO with minimax regret. By minimizing worst-case *regret* relative to the oracle predictor, WDRO-MRO provides a decision-centric robustness notion that directly bounds performance degradation under heterogeneous shifts. A modality-weighted Wasserstein cost further enables selective protection of vulnerable modalities. Theoretically, WDRO-MRO establishes a solid foundation: existence and uniqueness of minimax regret solutions under convex losses, convexity and strong duality of the formulation, and sensitivity characterizations of optimal regret with respect to ambiguity radii and modality weights. We also provide statistical guarantees including consistency, finite-sample generalization bounds, $O(N^{-1/2})$ convergence rates, and explicit sample complexity. Algorithmically, WDRO-MRO admits tractable convex reformulations (LP, SOCP, SDP, and power-cone programs) and introduces a dual-game algorithm that couples strong-dual reformulations with an exponentiated-weights adversary update, yielding an oracle-free no-regret procedure. Empirically, on the HANCOCK multimodal healthcare dataset, WDRO-MRO maintains competitive average accuracy and improves robustness and fairness compared to ERM and standard DRO, without incurring excessive conservatism.

## 1 Introduction

Multimodal machine learning (MML) has achieved strong progress by integrating data from multiple modalities (e.g., images, text, audio, video), distribution shift is a core robustness challenge (Qiu et al., 2022). Since empirical risk minimization (ERM) assumes training and test distributions coincide and thus fails under distribution shift, several studies address robustness by introducing auxiliary losses to reduce spurious correlations among signals (Yang et al., 2023), by de-bias training via a group distributionally robust optimization (DRO) objective (Kim et al., 2024), and by pre-training with DRO to optimize worst-case performance (Shuai et al., 2025). These approaches based on DRO improve empirical robustness but lack theoretical analysis.

DRO focuses on absolute risk (Kuhn et al., 2025), which may yield conservative solutions and overlook *oracle performance* (Agarwal & Zhang, 2022). DRO mitigates ERM's limitations by minimizing worst-case risk over an ambiguity set $\mathcal{U}_\rho(\hat{P}_N)$ centered at the empirical distribution:

$$\min_{f \in \mathcal{F}} \sup_{Q \in \mathcal{U}_\rho(\hat{P}_N)} \mathbb{E}_Q[\ell(z, f(z))],$$

where $\ell$ is a convex loss. To solve the conservativity issue, recent studies minimize regret instead of risk, either in the form of ex-post regret (Al Taha et al., 2023; Hajar et al., 2023; Kargin et al., 2024; Bitar, 2024) or ex-ante regret (Agarwal & Zhang, 2022; Cho & Yang, 2024; Poursoltani et al., 2024; Fiechtner & Blanchet, 2025). However, these approaches define ambiguity sets in a single-modal space, which fails to capture modality-specific distribution shifts common in multimodal applications.

Different modalities often show distinct noise structures and varying importance, and treating all modalities uniformly ignores this heterogeneity and may either over-regularize stable modalities or under-protect vulnerable ones.

We therefore introduce **Wasserstein Distributionally Robust Minimax Regret Optimization (WDRO-MRO)** for the multimodal setting, a framework that redefines robustness by minimizing worst-case *regret*:

$$\min_{f \in \mathcal{F}} \sup_{Q \in \mathcal{U}_\rho(\hat{P}_N)} \left( \mathbb{E}_Q[\ell(z, f(z))] - \inf_{f' \in \mathcal{F}} \mathbb{E}_Q[\ell(z, f'(z))] \right).$$

This approach bounds the performance gap relative to the oracle predictor, providing a decision-centric robustness measure. WDRO-MRO employs a modality-weighted Wasserstein cost, $c(z, z') = \sum_{k=1}^{K} \alpha_k d_k(z_k, z'_k)$, with nonnegative weights $\alpha_k$ and modality-specific metrics $d_k$ to prioritize robustness for critical modalities (e.g., noisy histological images in oncology). By leveraging convexity and strong duality, WDRO-MRO reformulates into tractable convex programs, including linear programs (LP), second-order cone programs (SOCP), and semidefinite programs (SDP), ensuring computational efficiency and scalability.

This paper has four main contributions:

- **Framework: WDRO-MRO**, the first regret-based multimodal learning framework, unifying modality-weighted Wasserstein ambiguity sets with minimax regret optimization.
- **Theory:** Proofs of existence and uniqueness of minimax regret solutions under convex losses, convexity and strong duality, and statistical guarantees, including consistency, finite-sample bounds, and $O(N^{-1/2})$ convergence rates.
- **Algorithms:** Tractable convex reformulations (e.g., linear programs (LP), second-order cone programs (SOCP), semidefinite programs (SDP), power-cone programs) across different loss functions and $p$-Wasserstein norms, together with a dual-game solver (Alg. 1) that couples strong-dual reformulations with an exponentiated-weights adversary update, yielding an oracle-free no-regret procedure balancing robustness and generalization.
- **Empirics:** Validation on the real-world HANCOCK dataset shows that WDRO-MRO achieves competitive accuracy, robustness and fairness.

## 2 PROBLEM FORMULATION AND PRELIMINARIES

In multimodal machine learning, data is represented as $z \in \mathcal{Z} = \mathcal{Z}_1 \times \cdots \times \mathcal{Z}_K$ (e.g., $\mathcal{Z}_1$ for images, $\mathcal{Z}_2$ for text). The function class $\mathcal{F}$ consists of cross-modal predictors $f : \mathcal{Z} \to \mathbb{R}$ (e.g., multimodal fusion networks that integrate features across modalities). The nominal distribution $P_0$ is unknown, but we observe $N$ i.i.d. samples $\{z_i = (z_{i1}, \ldots, z_{iK})\}_{i=1}^{N} \sim P_0$, forming the empirical distribution $\hat{P}_N = \frac{1}{N} \sum_{i=1}^{N} \delta_{z_i}$.

**Definition 2.1** (Multimodal Ambiguity Set)**.** To capture distribution shifts, we define the Wasserstein ambiguity set as $\mathcal{U}_\rho(\hat{P}_N) = \{Q \in \mathcal{P}(\mathcal{Z}) : W_p(\hat{P}_N, Q) \leq \rho\}$, with transportation cost $c(z, z') = \sum_{k=1}^{K} \alpha_k d_k(z_k, z'_k)$, where $\alpha_k \geq 0$ weights the importance of modality $k$, and $d_k$ is a modality-specific metric (e.g., pixel distance for images). This weighted cost allows for heterogeneous robustness across modalities.

**Definition 2.2** (Risk, Regret, and Core Problem)**.** The risk under $Q$ is $R_Q(f) = \mathbb{E}_Q[\ell(z, f(z))]$, and the regret is $\mathrm{Regret}_Q(f) = R_Q(f) - \inf_{f' \in \mathcal{F}} R_Q(f')$. The multimodal WDRO-MRO problem minimizes the worst-case regret: $\inf_{f \in \mathcal{F}} \sup_{Q \in \mathcal{U}_\rho(\hat{P}_N)} \mathrm{Regret}_Q(f) = \inf_{f \in \mathcal{F}} \sup_{Q \in \mathcal{U}_\rho(\hat{P}_N)} [\mathbb{E}_Q[\ell(z, f(z))] - \inf_{f' \in \mathcal{F}} \mathbb{E}_Q[\ell(z, f'(z))]]$, where the loss $\ell(z, v)$ is convex in $v = f(z)$ (e.g., cross-modal squared loss). This formulation captures multimodal shifts, such as inter-modal inconsistencies (e.g., image noise vs. text misalignment).

We have the standard assumptions of the multimodal setting:

*Assumption* 2.1 (Space & transport)*.* $\mathcal{Z}$ is a Polish (separable metric) space with its Borel $\sigma$-algebra. The transport cost $c : \mathcal{Z} \times \mathcal{Z} \to [0, \infty]$ is lower semicontinuous and modality-additive (e.g. $c(z, z') = \sum_{k=1}^{K} \alpha_k d_k(z_k, z'_k)$ with $\alpha_k \geq 0$).

*Assumption* 2.2 (Loss). For every $z$, the map $v \mapsto \ell(z, v)$ is convex and bounded; moreover it is $L$-Lipschitz on the prediction range.

*Assumption* 2.3 (Model class). $\mathcal{F}$ is a closed convex class. One of the following (sufficient) regularity conditions holds:

(a) (*Curvature*) $\ell(z, \cdot)$ is strictly/strongly convex $\Rightarrow$ uniqueness/stability, or

(b) (*Level-boundedness*) the outer objective has bounded lower level sets (e.g. via explicit regularization).

**Proposition 2.1** (Interchangeability / Strong Duality for Wasserstein DRO). *Under Assumptions 2.1–2.3, for any empirical reference $\hat{P}_N$ we have* $\sup_{Q: W_p(Q, \hat{P}_N) \leq \rho} \mathbb{E}_Q[\ell(z, f(z))] = \inf_{\lambda \geq 0} \Big\{ \lambda \rho + \mathbb{E}_{\hat{P}_N} \big[ \sup_{z' \in \mathcal{Z}} \{ \ell(z', f(z')) - \lambda c(\hat{z}, z') \} \big] \Big\}$.

This is a standard strong duality result for Wasserstein distributionally robust optimization; see, e.g., Mohajerin Esfahani & Kuhn (2018) and Kuhn et al. (2025, Lemma 4.16) for general statements.

## 3 Theoretical Analysis

This section develops the theoretical foundation of WDRO-MRO. Section 3.1 presents the core optimization properties: the existence of inner worst-case distributions, convexity of the outer objective, existence and uniqueness of solutions, and a strong dual formulation which supports the tractable reformulations in Sec. 3.2. Section 3.2 builds on these properties to obtain finite-dimensional convex programs and provides convergence and sensitivity analysis. Section 3.3 establishes statistical guarantees, including consistency, finite-sample bounds, and convergence and sensitivity analysis. Finally, Section 3.4 links WDRO-MRO to implicit regularization and robustness, and shows its continuous limit to ERM as the ambiguity radius vanishes. Detailed proofs can be found in Appendices F to I.

### 3.1 Basic Optimization Properties

Before deriving tractable convex programs in Sec. 3.2.1, we must ensure that the WDRO-MRO problem is well-defined and solvable, in the sense that worst-case distributions exist, the objective is convex, solutions exist and are unique, and the formulation admits a strong dual representation.

**Proposition 3.1** (Existence of Worst-Case Distribution). *Under Assumption 2.2 and 2.3 and Proposition 2.1, for any fixed $f \in \mathcal{F}$, there exists a worst-case distribution $Q^\star \in \mathcal{U}_\rho(\hat{P}_N)$ that attains $\sup_{Q \in \mathcal{U}_\rho(\hat{P}_N)} \mathrm{Regret}_Q(f)$. Moreover, $Q^\star$ is characterized by an optimal transport plan $\pi^\star$ respecting the weighted modality costs $\alpha_k d_k(z_k, z'_k)$, where $\pi^\star$ solves the Kantorovich problem with cost $c(z, z') = \sum_k \alpha_k d_k(z_k, z'_k)$.*

**Proposition 3.2** (Convexity of the Problem). *Under Assumption 2.2 and 2.3 and Proposition 2.1, the WDRO-MRO objective $\phi(f)$ is convex in $f \in \mathcal{F}$. Furthermore, if $\ell(z, v)$ is strongly convex in $v$ with modulus $\kappa > 0$, and the modality-specific assumptions hold (e.g., additive convexity across modalities), then $\phi(f)$ is strongly convex in $f$.*

**Proposition 3.3** (Existence and Uniqueness of Solutions). *Under Assumption 2.2 and 2.3 and Proposition 2.1, the infimum in the WDRO-MRO problem is attained. Furthermore, if the loss function $\ell(z, v)$ is strictly convex in $v$, then the solution is unique.*

**Proposition 3.4** (Strong Duality). *Under Assumption 2.2 and 2.3 and Proposition 2.1, the WDRO-MRO problem admits a strong dual formulation with zero duality gap. Specifically, for any fixed $f \in \mathcal{F}$, the inner maximization $\sup_{Q \in \mathcal{U}_\rho(\hat{P}_N)} \mathrm{Regret}_Q(f)$ equals $\inf_{\lambda \geq 0} \lambda \rho + \mathbb{E}^{\hat{P}_N} [\sup_{z' \in \mathcal{Z}} (\ell(z, f(z')) - \lambda c(\hat{z}, z')) - \inf_{f' \in \mathcal{F}} \sup_{z'' \in \mathcal{Z}} (\ell(z, f'(z'')) - \lambda c(\hat{z}, z''))]$, where the overall problem reformulates as a finite-dimensional convex optimization problem over dual variables.*

### 3.2 Computational Properties

By strong duality, WDRO-MRO reduces to finite-dimensional convex programs whose type is determined by the loss and the Wasserstein norm: LP, SOCP, SDP, or power/exponential-cone

(Sec. 3.2.1). These programs are handled by standard solvers. Beyond these direct solves, Sec. 3.2.2 presents an oracle-free dual-game solver that operates on the same tractable envelopes and alternates adversarial exponentiated-weights updates with learner/oracle best responses and a projected update for the radius dual variable. Sec. 3.2.3 states the convergence guarantees, and Sec. 3.2.4 characterizes how the ambiguity radius and modality weights influence the optimum and informs tuning.

### 3.2.1 TRACTABLE REFORMULATIONS FOR GENERAL $p$

Throughout this subsection, we assume $f$ is affine, i.e., $f(z') = \sum_m F_m z'_m + g$, standard in multimodal machine learning, ensuring finite suprema. The transportation cost is $c(z, z') = \sum_{m=1}^{K} \alpha_m \|z_m - z'_m\|_p^p$, with weights $\alpha_m \geq 0$ modulating robustness across modalities, prioritizing those with higher $\alpha_m$. Assumptions 2.2–2.1 hold, ensuring convexity and measurability, with Proposition 2.1 guaranteeing interchange; see (Zhang et al., 2025). All reformulations are finite-dimensional convex programs with zero duality gap (Section 3.2). We organize the results by loss type (piecewise linear, quadratic, general convex) and Wasserstein norm $p$. These reformulations provide tractable solutions for WDRO-MRO across different $p$-norms and loss types, leveraging LP for polyhedral constraints, SOCP/SDP for quadratic terms, and power/exponential cones for general $p$. The reformulations are summarized in Table 1. For brevity, the main results for general convex loss, piecewise linear and quadratic cases are given in Appendix B.

Table 1: Tractable reformulations for WDRO-MRO under different losses and Wasserstein norms.

| Loss Type | $p$-norm | Constraints | Cone / Program Type |
|---|---|---|---|
| Piecewise Linear | $p = 1$ | Linear constraints with aux. vars. | LP |
| | $p = 2$ | Rotated quadratic constraints | SOCP |
| | $2 < p < \infty$ | Power cone constraints | Convex (Power Cone) |
| | $p = \infty$ | Vertex-enumeration constraints | LP |
| Quadratic | $p = 1$ | Matrix inequality (block PSD) | SDP (SOCP if diag.$Q$) |
| | $p = 2$ | Matrix inequality (block PSD) | SDP (SOCP if diag.$Q$) |
| | $2 < p < \infty$ | Conjugate representation | SDP / Exp. Cone |
| | $p = \infty$ | Vertex-PSD constraints | SDP |
| General Convex | $p = 1$ | Convex conjugate constraints | SDP / LP (Lipschitz case) |
| | $p = 2$ | S-lemma based constraints | SDP |
| | $2 < p < \infty$ | Conjugate + power cone | Convex (Power/Exp. Cone) |
| | $p = \infty$ | Polyhedral or dual vertex constraints | LP / SDP |

**Canonical Objective.** All tractable reformulations in Lemmas B.1 to B.12 share the following canonical objective: $\min_{f \in \mathcal{F}, \lambda, \lambda' \geq 0, s_i, s'_i} \quad \lambda \rho + \frac{1}{N} \sum_{i=1}^{N} s_i - \left( \lambda' \rho + \frac{1}{N} \sum_{i=1}^{N} s'_i \right)$, where $\{s_i\}$ correspond to the regret constraints for the candidate predictor $f$, and $\{s'_i\}$ are defined analogously for the oracle predictor in the infimum term.

### 3.2.2 ORACLE-FREE DUAL-GAME HYBRID SOLVER

The WDRO–MRO problem can be cast as a two-player zero-sum game between the learner and an adversarial nature. Leveraging the strong-dual reformulations in Section 3.2.1, we construct an oracle-free iterative scheme in Algorithm 1: dual envelopes are computed via tractable convex programs, nature updates its distribution using exponentiated weights, and the learner/oracle predictors are updated accordingly.

*Remark* 3.1 (Non-convex deep models). Our tractability and convergence results rely on convexity of the learner and oracle objectives. For non-convex deep architectures, Algorithm 1 can be instantiated as a first-order min–max procedure: in each iteration, the "Learner / Oracle updates" are implemented by one or a few stochastic gradient steps on mini-batches, while the adversary distribution is updated via the same exponentiated-weights rule. This corresponds to replacing exact best responses with

---

**Algorithm 1** WDRO–MRO: Oracle-Free Dual-Game Solver with Exponentiated Weights

---

**Require:** samples $\{\hat{z}_i\}_{i=1}^N$, radius $\rho$, cost $c$, loss $\ell$, steps $\eta, \eta_\lambda$
 1: Initialize $w_1(i) \leftarrow 1/N$, $\lambda_1 \leftarrow 0$, pick $f_1, g_1 \in \mathcal{F}$
 2: **for** $t = 1, 2, \ldots, T$ **do**
 3:     **Dual envelopes:** for each $i$, compute $s_i(f_t, \lambda_t)$ and $s_i(g_t, \lambda_t)$ from the canonical objective (Section 3.2.1).
 4:     **Nature update:** let $\Delta_i \leftarrow s_i(f_t, \lambda_t) - s_i(g_t, \lambda_t)$ and update $w_{t+1}(i) \leftarrow \frac{w_t(i)\exp(\eta\,\Delta_i)}{\sum_{j=1}^N w_t(j)\exp(\eta\,\Delta_j)}$.
 5:     **Learner / Oracle updates:** $f_{t+1} \in \arg\min_{f \in \mathcal{F}} \left\{ \lambda_t \rho + \sum_{i=1}^N w_{t+1}(i)\, s_i(f, \lambda_t) \right\}$, $g_{t+1} \in$
     $\arg\min_{g \in \mathcal{F}} \left\{ \lambda_t + \sum_{i=1}^N w_{t+1}(i)\, s_i(g, \lambda_t) \right\}$.
 6:     **Radius dual:** update $\lambda_{t+1} \leftarrow \Pi_{[0, \lambda_{\max}]}\left( \lambda_t + \eta_\lambda(\rho - \hat{\rho}_t) \right)$, where $\hat{\rho}_t$ is the empirical dual subgradient.
 7: **end for**
 8: **Output:** averaged predictor $\bar{f} \leftarrow \frac{1}{T} \sum_{t=1}^T f_t$

---

approximate SGD-based updates, in line with standard practice in non-convex DRO and adversarial training.

### 3.2.3 ALGORITHMIC CONVERGENCE GUARANTEES

We next establish convergence guarantees for the convex subproblems introduced in Section 3.2. These include LP, SOCP, SDP, and power or exponential cone programs. Under standard assumptions, interior-point or first-order methods achieve either linear or sublinear rates. The modality weights $\alpha_m \geq 0$ in the transportation cost $c(z, z') = \sum_{m=1}^K \alpha_m \|z_m - z'_m\|_p^p$ affect the associated Lipschitz constants and thereby influence convergence rates. All subproblems are convex with zero duality gap (Proposition 3.4), and attain their optima by Proposition 3.1.

**Proposition 3.5** (Global convergence of the Dual-Game Hybrid Solver). *Suppose Assumption 2.2 and 2.3 and Proposition 2.1 hold and the tractable reformulations in Section 3.2.1 admit zero duality gap with attained optima. Let the nature weights $w_t \in \Delta([N])$ be updated by exponentiated weights with step size $\eta = \Theta(\sqrt{\ln N / T})$. Assume learner and oracle best-responses are computed to accuracy $\varepsilon_t \geq 0$, and that the dual variable $\lambda_t \in [0, \lambda_{\max}]$ is updated by projected subgradient ascent with steps $\eta_{\lambda,t} = \Theta(1/\sqrt{T})$ and bounded subgradients $\|g_t\| \leq G$.*

*Define the saddle objective $\Phi(f, g, w, \lambda) = \lambda\rho + \sum_{i=1}^N w(i)\, s_i(f, \lambda) - \left( \lambda\rho + \sum_{i=1}^N w(i)\, s'_i(g, \lambda) \right)$, and the averaged iterates $\bar{f} = \frac{1}{T}\sum_{t=1}^T f_t$, $\bar{g} = \frac{1}{T}\sum_{t=1}^T g_t$, $\bar{w} = \frac{1}{T}\sum_{t=1}^T w_t$, $\bar{\lambda} = \frac{1}{T}\sum_{t=1}^T \lambda_t$. Then $\max_{w \in \Delta([N]), \lambda \in [0, \lambda_{\max}]} \Phi(\bar{f}, \bar{g}, w, \lambda) - \min_{f, g \in \mathcal{F}} \Phi(f, g, \bar{w}, \bar{\lambda}) = \mathcal{O}\left( \sqrt{\frac{\ln N}{T}} \right) + \mathcal{O}\left( \frac{1}{\sqrt{T}} \right) + \frac{1}{T}\sum_{t=1}^T \varepsilon_t$. In particular, if all best-responses are solved exactly ($\varepsilon_t = 0$), the averaged iterate $(\bar{f}, \bar{g}, \bar{w}, \bar{\lambda})$ constitutes an $\tilde{\mathcal{O}}(1/\sqrt{T})$ saddle point of the hybrid dual game.*

**Proposition 3.6** (Global convergence with continuous $\mathcal{W}$). *Assume the setting of Proposition 3.5, but let nature's strategy set be the continuous density-ratio class $\mathcal{W}_B = \{w : 0 \leq w(z) \leq B, \mathbb{E}_{P_0}[w] = 1\}$. Suppose at each iteration the adversary's update is implemented by the exact closed form $w_t^\star \in \arg\max_{w \in \mathcal{W}_B} \Phi(f_t, g_t, w, \lambda_t)$. Then with the same learner/oracle updates and dual steps as in Proposition 3.5, the averaged iterate $(\bar{f}, \bar{g}, \bar{w}, \bar{\lambda})$ satisfies $\max_{w \in \mathcal{W}_B, \lambda \in [0, \lambda_{\max}]} \Phi(\bar{f}, \bar{g}, w, \lambda) - \min_{f, g \in \mathcal{F}} \Phi(f, g, \bar{w}, \bar{\lambda}) = \tilde{\mathcal{O}}\left( \frac{1}{\sqrt{T}} \right)$.*

### 3.2.4 SENSITIVITY ANALYSIS

We analyze the sensitivity of the optimal regret $R(\epsilon) = \inf_{f \in \mathcal{F}} \sup_{Q : W_p(Q, \hat{P}_N) \leq \epsilon} \text{Regret}_Q(f)$ to the ambiguity radius $\epsilon$, critical for tuning robustness in multimodal settings with heterogeneous noise (e.g., images vs. text). We derive continuity and Lipschitz bounds, extended to high-dimensional regimes via a reformulation equivalent to a low-dimensional optimization, avoiding costly cross-validation (Aolaritei et al., 2022).

*Lemma* 3.1 (Sensitivity of Optimal Regret). The optimal regret $R(\rho) = \inf_{f \in \mathcal{F}} \sup_{Q:W_p(Q,\hat{P}_N) \leq \rho} \text{Regret}_Q(f)$ is continuous on $\rho > 0$. It is Lipschitz continuous with constant $L$, the Lipschitz modulus of $\ell(z, v)$ in $v$. The subgradient satisfies $\partial R(\rho) \subseteq [0, \lambda^\star]$, where $\lambda^\star \geq 0$ is the optimal dual variable in the Kantorovich-Rubinstein dual from Proposition 3.4. For multimodal costs $c(z, z') = \sum_{m=1}^{K} \alpha_m \|z_m - z'_m\|_p^p$, the weights $\alpha_m \geq 0$ modulate the subgradient via the transportation cost gradient $\|\nabla c(z, z')\| \leq \sum_m \alpha_m \|z_m - z'_m\|_{p-1}^{p-1}$.

*Lemma* 3.2 (High-Dimensional Error Equivalence). For high-dimensional multimodal data, the WDRO estimation error $\|\hat{f}_{DRE} - f_0\|^2/d$ in the proportional regime ($d, n \to \infty$, $d/n \to \rho$) is equivalent to the solution of a convex-concave optimization over four scalar variables:

$$\min_{0 \leq \alpha \leq \sigma_{f_0}} \max_{\substack{\beta \geq 0 \\ \tau_1, \tau_2 > 0}} \left\{ \frac{\beta \tau_1}{2} + \frac{\rho_0 \beta \tau_2}{2} - \frac{\beta^2}{2M} + \mathcal{L}\left(\alpha, \frac{\tau_1}{\beta}\right) + \frac{\sqrt{\rho_0}\beta \rho(\sigma_{f_0}^2 + \alpha^2)}{2\tau_2} - \alpha \beta \sqrt{\rho} \sqrt{\frac{\rho \rho_0 \sigma_{f_0}^2}{\tau_2^2} + 1} \right\}.$$

where $\mathcal{L}$ is the smoothed loss function, $\sigma_{f_0}^2$ is the oracle predictor's variance scaled by modality weights $\alpha_m$, and $\rho = \rho_0/n^{p/2}$.

## 3.3 STATISTICAL PROPERTIES

This section develops statistical guarantees that show the estimator trained on finite data generalizes to the underlying distribution. Specifically, we derive consistency, finite-sample bounds, convergence rates, sample complexity requirements, and the asymptotic unbiasedness of WDRO-MRO estimator.

**Theorem 3.1** (Statistical Consistency of WDRO-MRO). *Let* $\hat{f}_{DRE} = \arg\min_{f \in \mathcal{F}} \sup_{Q \in U_\rho(\hat{P}_N)} \text{Regret}_Q(f)$ *be the WDRO-MRO estimator, where* $U_\rho(\hat{P}_N) = \{Q \in \mathcal{P}(\mathcal{Z}) : W_p(\hat{P}_N, Q) \leq \rho\}$ *with* $\rho = \rho_0/N^{p/2}$, *and* $\hat{P}_N = \frac{1}{N} \sum_{i=1}^{N} \delta_{z_i}$ *is the empirical distribution from* $N$ *i.i.d. samples* $z_i \sim P_0$. *Let* $f_0 = \arg\min_{f \in \mathcal{F}} \sup_{Q \in B_\rho(P_0)} \text{Regret}_Q(f)$ *be the population minimax regret minimizer. Under Assumption 2.2 and 2.3 and Proposition 2.1,* $\hat{f}_{DRE} \to f_0$ *in probability as* $N \to \infty$, *i.e., for any* $\epsilon > 0$, $\mathbb{P}\left( \|\hat{f}_{DRE} - f_0\|_{\mathcal{F}} > \epsilon \right) \to 0$, *where* $\| \cdot \|_{\mathcal{F}}$ *is the sup-norm on the compact function class* $\mathcal{F}$.

**Theorem 3.2** (Finite-Sample Guarantees for Out-of-Sample Regret). *Let* $\hat{f}_{DRE} = \arg\min_{f \in \mathcal{F}} \sup_{Q \in U_\rho(\hat{P}_N)} \text{Regret}_Q(f)$ *be the WDRO-MRO estimator.* $\sup_{Q \in B_\rho(P_0)} \text{Regret}_Q(\hat{f}_{DRE}) \leq \inf_{f \in \mathcal{F}} \sup_{Q \in B_\rho(P_0)} \text{Regret}_Q(f) + LW_p(\hat{P}_N, P_0) + 2\mathcal{R}_N(\mathcal{F}) + \sqrt{\frac{2\log(2/\delta)}{N}}$, *where* $L$ *is the effective Lipschitz modulus defined in Appendix H.2,* $\mathcal{R}_N(\mathcal{F})$ *is the Rademacher complexity of* $\{\ell(z, f(z)) : f \in \mathcal{F}\}$, *and the weights* $\alpha_k$ *scale the bound through the variance* $\sigma^2 = \sum_{k=1}^{K} \alpha_k^2 \sigma_k^2$ *in the multimodal cost.*

*Lemma* 3.3 (Convergence Rates for Regret). Under Assumption 2.2 and 2.3 and Proposition 2.1, let $\hat{f}_{DRE} = \arg\min_{f \in \mathcal{F}} \sup_{Q \in U_\rho(\hat{P}_N)} \text{Regret}_Q(f)$ be the WDRO-MRO estimator. The out-of-sample regret satisfies, with probability at least $1 - \delta$, $\sup_{Q \in B_\rho(P_0)} \text{Regret}_Q(\hat{f}_{DRE}) - \inf_{f \in \mathcal{F}} \sup_{Q \in B_\rho(P_0)} \text{Regret}_Q(f) = O\left( \sqrt{\frac{\log(1/\delta)}{N}} \right)$, leveraging the Rademacher complexity $\mathcal{R}_N(\mathcal{F}) = O(1/\sqrt{N})$ of the multimodal function class $\mathcal{F}$, scaled by modality weights $\alpha_k$ through the variance $\sigma^2 = \sum_k \alpha_k^2 \sigma_k^2$.

*Lemma* 3.4 (Sample Complexity for $\epsilon$-Optimal Regret). Let $d = \sum_{k=1}^{K} d_k$ be the total dimension and $\text{vc}(\mathcal{G})$ the VC dimension of $\mathcal{G} = \{\ell(z, f(z)) : f \in \mathcal{F}\}$. Under Assumption 2.2 and 2.3 and Proposition 2.1, let $\hat{f}_{DRE} = \arg\min_{f \in \mathcal{F}} \sup_{Q \in U_\rho(\hat{P}_N)} \text{Regret}_Q(f)$. There exist constants $C_1, C_2 > 0$ such that if $N \geq C_1 \frac{\text{vc}(\mathcal{G}) + \log(2/\delta)}{\epsilon^2}$ and $N \geq C_2 \left(\frac{L}{\epsilon}\right)^{\max\{2, d/p\}}$, where $L = L_\ell \sum_{k=1}^{K} \alpha_k$ is the Lipschitz constant scaled by modality weights $\alpha_k$, then with probability at least $1 - \delta$, $\sup_{Q \in B_\rho(P_0)} \text{Regret}_Q(\hat{f}_{DRE}) - \inf_{f \in \mathcal{F}} \sup_{Q \in B_\rho(P_0)} \text{Regret}_Q(f) \leq \epsilon$.

*Lemma* 3.5 (Asymptotic Unbiasedness of Debiased WDRO-MRO). Let $\hat{f}_{DRE} = \arg\min_{f \in \mathcal{F}} \sup_{Q \in U_\rho(\hat{P}_N)} \text{Regret}_Q(f)$ be the WDRO-MRO estimator. Define the debiased estimator $\hat{f}_{deb} = \hat{f}_{DRE} + b_N$, where $b_N = O(1/N)$ is a bias correction term scaled by modality weights $\alpha_k$ through the variance $\sigma^2 = \sum_k \alpha_k^2 \sigma_k^2$. Under Assumption 2.2 and 2.3 and Proposition 2.1,

as $N \to \infty$, $\mathbb{E}[\hat{f}_{deb}] \to f_0$, where $f_0 = \arg\min_{f \in \mathcal{F}} \sup_{Q \in B_\rho(P_0)} \mathrm{Regret}_Q(f)$ is the population minimax regret minimizer.

## 3.4 REGULARIZATION AND ROBUSTNESS PROPERTIES

This subsection interprets WDRO-MRO as a regularization mechanism and quantifies its robustness in multimodal settings. As the ambiguity radius increases, the solution becomes more conservative with respect to modality-specific shifts, while as the radius vanishes WDRO-MRO converges continuously to ERM.

In addition to Assumptions 2.1–2.3, we make the following standing assumptions for the regularization equivalences.

*Assumption* 3.1 (Geometry and tails for regularization). Let $\mathcal{Z} = \mathcal{Z}_1 \times \cdots \times \mathcal{Z}_K$ be a product Polish space and let the multimodal cost be $c(z, z') = \sum_{k=1}^{K} \alpha_k \|z_k - z'_k\|_p^p$, with $\alpha_k \geq 0$ and $p \in [1, \infty)$.

  (i) (*Loss regularity*) The loss $\ell(z, v)$ is convex in $v$ and $L$-Lipschitz in $v$ on the prediction range (Assumption 2.2); for the $p > 1$ variants we additionally assume that $\ell(z, \cdot)$ is differentiable with Lipschitz gradient in $v$ on bounded sets.

  (ii) (*Multimodal separability*) The model class is modality-separable in the sense that $f(z) = \sum_{k=1}^{K} f_k(z_k)$ for $f \in \mathcal{F}$, and each component $f_k$ belongs to a convex class $\mathcal{F}_k$.

  (iii) (*Finite variation / smoothness*) For $p = 1$ we assume that each $f_k$ has bounded total variation on $\mathcal{Z}_k$, so that $\mathrm{TV}_k(f_k) < \infty$. For $p > 1$ we assume that each $f_k$ lies in a Sobolev-type ball with finite gradient (or higher-order) seminorm, ensuring that the corresponding conjugate penalty is finite.

  (iv) (*Tail / moment condition*) The data distribution has finite $p$-th moments in each modality: $\mathbb{E}\big[\sum_{k=1}^{K} \alpha_k \|Z_k\|_p^p\big] < \infty$, or, equivalently for our purposes, the empirical support lies in a bounded set with respect to $c(\cdot, \cdot)$.

These conditions ensure that the Kantorovich dual representation is well defined, the inner suprema in the dual problems are finite, and the Fenchel-conjugate-based regularizers (total variation or Sobolev-type) are proper and lower semicontinuous.

*Lemma* 3.6 (Variational Regularization Equivalence). Under Assumptions 2.1–3.1, the WDRO-MRO problem $\inf_{f \in \mathcal{F}} \sup_{Q \in \mathcal{U}_\rho(\hat{P}_N)} \mathrm{Regret}_Q(f)$ is equivalent to the variation regularized problem $\inf_{f \in \mathcal{F}} \mathbb{E}_{\hat{P}_N}[\ell(z, f(z))] + \gamma \mathrm{Var}(f)$, where $\mathrm{Var}(f) = \sum_{k=1}^{K} \alpha_k \mathrm{TV}_k(f_k)$ is the multimodal total variation regularizer, $\mathrm{TV}_k(f_k)$ is the total variation norm for modality $k$, and $\gamma > 0$ depends on $\rho$ and the Lipschitz modulus of $\ell$.

*Proof sketch.* Under Assumptions 2.1–3.1, the inner Wasserstein-robust risk admits the Kantorovich dual representation (Proposition 2.1): $\sup_{Q: W_p(Q, \hat{P}_N) \leq \rho} \mathbb{E}_Q[\ell(z, f(z))] = \inf_{\lambda \geq 0} \big\{\lambda\rho + \mathbb{E}_{\hat{P}_N}\big[\sup_{z'}\{\ell(z', f(z')) - \lambda c(z, z')\}\big]\big\}$. For $p = 1$ and an $L$-Lipschitz loss, the inner supremum can be rewritten via the Fenchel conjugate of $\ell$ evaluated at dual vectors whose norm is controlled by the unit ball of the dual transport cost (e.g. Azizian et al., 2023; Gao et al., 2024). Because the cost is additive across modalities, $c(z, z') = \sum_k \alpha_k d_k(z_k, z'_k)$, the dual constraint decomposes by modality and yields a sum of total-variation seminorms $\mathrm{TV}_k(f_k)$, each weighted by $\alpha_k$. The resulting objective has the form $\mathbb{E}_{\hat{P}_N}[\ell(z, f(z))] + \gamma \sum_{k=1}^{K} \alpha_k \mathrm{TV}_k(f_k)$, with $\gamma$ proportional to the optimal dual variable $\lambda^\star$ and the radius $\rho$. Applying the same argument to the regret baseline term (the infimum over $f'$) gives the stated equivalence between the WDRO-MRO problem and a variation-regularized ERM problem. A detailed derivation is provided in Appendix I.1. □

*Lemma* 3.7 (Multimodal Lipschitz Regularization Equivalence). Under Assumption 3.1, consider the WDRO-MRO problem for classification losses $\ell(y, w^\top x)$ (e.g., logistic: $\ell(y, v) = \log(1 + \exp(-yv))$) that are convex and $L$-Lipschitz in $v$, with multimodal linear fusion model

$f(z) = w^\top z = \sum_{k=1}^{K} w_k^\top z_k$ where $z = (z_1, \ldots, z_K) \in \mathcal{Z}_1 \times \cdots \times \mathcal{Z}_K$. The transportation cost is $c(z, z') = \sum_{k=1}^{K} \alpha_k \|z_k - z'_k\|_p^p$ with $\alpha_k \geq 0$. Then, the WDRO-MRO problem $\inf_w \sup_{Q \in \mathcal{U}_\rho(\hat{P}_N)} \left[ \mathbb{E}_Q[\ell(y, w^\top x)] - \inf_{w'} \mathbb{E}_Q[\ell(y, (w')^\top x)] \right]$ is equivalent to the regularized empirical risk minimization $\min_w \mathbb{E}_{\hat{P}_N}[\ell(y, w^\top x)] + \gamma \|w\|_*$, where $\gamma > 0$ depends on $\rho$ and the Lipschitz modulus of $\ell$, and $\|w\|_* = \sup_{\|u\|_p \leq 1} w^\top u$ is the dual norm weighted by modalities: specifically, $\|w\|_* = \inf_{\beta_k \geq 0, \sum_k \beta_k = 1} \sum_{k=1}^{K} \frac{\|w_k\|_q}{\alpha_k \beta_k}$ with $q = p/(p-1)$ (Holder dual), ensuring modality-specific robustness modulated by $\alpha_k$.

*Lemma* 3.8 (Convergence to Multimodal ERM). Under Assumption 2.2 and 2.3 and Proposition 2.1, as the ambiguity radius $\rho \to 0$, the WDRO-MRO problem $\inf_{f \in \mathcal{F}} \sup_{Q \in \mathcal{U}_\rho(\hat{P}_N)} \mathrm{Regret}_Q(f)$ converges to the multimodal empirical risk minimization (ERM) $\inf_{f \in \mathcal{F}} \mathbb{E}_{\hat{P}_N}[\ell(z, f(z))]$, ensuring graceful degradation: the solution $\hat{f}_\rho$ approaches the ERM solution $\hat{f}_{\mathrm{ERM}}$ continuously in the sup-norm on $\mathcal{F}$, with the rate modulated by modality weights $\alpha_k$ through the sensitivity $\partial R / \partial \rho \subseteq [0, \lambda^\star]$, where $\lambda^\star$ scales with $\sum_k \alpha_k$.

## 4 APPLICATIONS AND EXPERIMENTS

### 4.1 APPLICATION: WDRO–MRO FOR LOGISTIC REGRESSION

We illustrate the framework on logistic regression. Throughout, $y \in \{\pm 1\}$ and $\ell(y, v) = \log(1 + \exp(-yv))$, which is 1-Lipschitz in $v$ and convex. Let $x = (x_1, \ldots, x_K)$ be multimodal features and $w = (w_1, \ldots, w_K)$ the linear classifier so that $f(x) = w^\top x$. The transportation cost is $c(x, x') = \sum_{k=1}^{K} \alpha_k \|x_k - x'_k\|_p^p$ as in Definition 4.1.

**Definition 4.1** (WDRO–MRO for logistic regression). The WDRO–MRO objective reads $\min_{w \in \mathbb{R}^d} \sup_{Q \in \mathcal{U}_\rho(\hat{P}_N)} \left\{ R_Q(w) - \inf_{w' \in \mathbb{R}^d} R_Q(w') \right\}, R_Q(w) = \mathbb{E}_Q[\ell(y, w^\top x)]$.

**A. Strong-dual envelopes and tractable reformulations.** Specializing Section 3.2.1 to the logistic loss ($\ell$ convex, $L=1$-Lipschitz) and affine $f(x') = w^\top x'$, we obtain the per-sample dual envelopes $s_i(w, \lambda) = \sup_{x'} \{ \ell(y_i, w^\top x') - \lambda c(\hat{x}_i, x') \}, s'_i(w', \lambda) = \sup_{x'} \{ \ell(y_i, w'^\top x') - \lambda c(\hat{x}_i, x') \}$, which instantiate the canonical objective in Eq. (Section 3.2.1).

**Proposition 4.1** (Envelopes for logistic; tractable per $p$). *Under Assumptions 2.2–2.3 and $f(x') = w^\top x'$:*

  (i) $p = 1$ ***(LP via Lipschitz).*** *Using the $L=1$ Lipschitz bound from Lemma B.1, the envelope admits an LP representation with auxiliary variables $t_{ikj} \geq 0$: $s_i \geq \ell(y_i, w^\top \hat{x}_i) + \lambda \sum_{k=1}^{K} \alpha_k \sum_{j=1}^{d_k} t_{ikj}, \quad |x'_{k,j} - \hat{x}_{i,k,j}| \leq t_{ikj}$.*

  (ii) $p = 2$ ***(SDP/SOCP via conjugate).*** *By Lemma B.2, using the convex conjugate of $\ell$ and $c^*(\cdot)$ for $p=2$, $s_i \geq \inf_{u \in \mathbb{R}} \ell^*(y_i, u) + \lambda \sum_{k=1}^{K} \left( \frac{1}{4\alpha_k} \|u w_k\|_2^2 + u w_k^\top \hat{x}_{i,k} \right)$, which yields an SDP; if blocks are diagonal it reduces to SOCP (rotated cones).*

  (iii) $2 < p < \infty$ ***(power/exp. cones).*** *By Lemma B.3, the envelope is representable via a convex program over power cones (rational $p$) or exponential cones (irrational $p$).*

  (iv) $p = \infty$ ***(LP/SDP via vertex dual).*** *Using Lemma B.4, we obtain an LP/SDP through polyhedral/vertex constraints of the box uncertainty region.*

*In all cases the WDRO–MRO objective with these envelopes is a finite-dimensional convex program with zero duality gap.*

**B. Regularized ERM view (upper bound, implementable).** For $p=2$ and logistic loss, the envelope in 4.1(ii) implies a tight implementable upper bound that yields a group-norm penalty.

*Corollary* 4.1 (Group-norm regularization upper bound, $p=2$). Let $y \in \{\pm 1\}$ and $\ell(y, \cdot)$ be 1-Lipschitz. For $c(x, x') = \sum_k \alpha_k \|x_k - x'_k\|_2^2$, $\sup_{Q \in \mathcal{U}_\rho(\hat{P}_N)} R_Q(w) \leq \frac{1}{N} \sum_{i=1}^{N} \ell(y_i, w^\top \hat{x}_i) +$

$\rho \sum_{k=1}^{K} \frac{\|w_k\|_2}{\sqrt{\alpha_k}}$. Consequently, $\min_w \ \sup_{Q \in \mathcal{U}_\rho(\hat{P}_N)} R_Q(w) \ \leq \ \min_w \ \frac{1}{N} \sum_{i=1}^{N} \ell(y_i, w^\top \hat{x}_i) + \gamma \sum_{k=1}^{K} \frac{\|w_k\|_2}{\sqrt{\alpha_k}}$, with $\gamma$ proportional to $\rho$ (the constant depends on the chosen conjugate calibration). Thus WDRO induces a *modality-weighted group-lasso* penalty.

*Remark* 4.1. The bound in Corollary 4.1 is exact for several Lipschitz losses and serves as a tight surrogate for logistic; it is useful for large-scale training and matches the intuition that larger $\alpha_k$ (more trusted modality) yields weaker shrinkage on $w_k$.

**C. Oracle-free dual-game solver (specialized to logistic regression)**   is provided in Appendix C.

## 4.2 EXPERIMENTAL EVALUATION

We next evaluate WDRO-MRO on the real world HANCOCK dataset (Dörrich et al., 2025), which contains multimodal records from 763 head and neck cancer patients (2005–2019).

### 4.2.1 EXPERIMENTAL SETUP

**Dataset and Preprocessing.**   This paper uses five modalities of HANCOCK in experiments, and the details can be found in Appendix J.1. We simulate robustness stress tests by injecting noise into both labels and features. Specifically, we consider noise rates $\rho \in \{0.0, 0.1, 0.2, 0.3, 0.4, 0.5\}$, applied as **label noise**, where a fraction $\rho$ of labels is randomly flipped, and **feature noise**, where Gaussian perturbations are injected at the group level, targeting one or more modalities. To address class imbalance, we apply SMOTE oversampling after noise injection. Each experiment is repeated with 5 random seeds. **Baselines.** We compare WDRO-MRO against three baselines: **ERM (Logistic/MLP)** - Empirical Risk Minimization with logistic regression or a multilayer perceptron, and **WDRO** - Standard DRO with Wasserstein distance.

**Evaluation Metrics.**   We group evaluation metrics into three categories: (A) **Performance** metrics, which measure the overall predictive accuracy (e.g., Average AUC); (B) **Robustness** metrics (Sagawa et al., 2020; Koh et al., 2021), such as Robust AUC ($\min_\rho \mathrm{AUC}(\rho)$), RR-AUC (Relative Robustness AUC, $\frac{\mathrm{Robust\ AUC}}{\max_\rho \mathrm{AUC}(\rho)}$), and Worst-Case Drop ($\max_\rho \mathrm{AUC}(\rho) - \mathrm{Robust\ AUC}$)); and (C) **Fairness** metrics, such as GNR (Group-Noise Robustness, $\min_{g,\rho}\{\mathrm{AUC}_g(\rho)\}$), GF Gap (Group-Fairness Gap, $\max_g \overline{\mathrm{AUC}}_g - \min_g \overline{\mathrm{AUC}}_g$). Detailed definitions of all metrics are provided in Table 4.

### 4.2.2 RESULTS

Table 2: WDRO-MRO shows strong performance, robustness and fairness on HANCOCK dataset. Best values (per split, per column) are in **bold**, with detailed visualizations provided in Figures 4 to 7.

| Model | Split | Performance | | Robustness | | | Fairness | | Stability | |
|---|---|---|---|---|---|---|---|---|---|---|
| | | Avg ↑ AUC ± Std ↓ | | Robust AUC ↑ | RR-AUC ↑ | W.C. Drop ↓ | GNR ↑ | GF Gap ↓ | NS Drop ↓ | \|NS Slope\| ↓ |
| ERM (Logistic) | ID | 0.635 ±0.105 | | 0.528 | 0.670 | 0.259 | 0.712 | 0.034 | 0.259 | -0.526 |
| | OOD | 0.613 ±0.095 | | 0.477 | 0.654 | 0.253 | 0.662 | 0.047 | 0.253 | -0.463 |
| | Oropharynx | 0.586 ±0.080 | | 0.470 | 0.707 | 0.195 | 0.620 | 0.016 | 0.195 | -0.383 |
| ERM (MLP) | ID | 0.602 ±0.090 | | 0.509 | 0.687 | 0.232 | 0.674 | 0.030 | 0.232 | -0.433 |
| | OOD | 0.564 ±0.075 | | 0.494 | 0.775 | 0.144 | 0.604 | 0.032 | 0.144 | -0.296 |
| | Oropharynx | 0.565 ±0.079 | | 0.463 | 0.723 | 0.178 | 0.613 | 0.017 | 0.178 | -0.341 |
| GDRO | ID | 0.633 ±0.062 | | 0.537 | 0.776 | 0.155 | 0.675 | 0.004 | 0.155 | -0.289 |
| | OOD | 0.599 ±0.086 | | 0.448 | 0.686 | 0.205 | 0.644 | **0.002** | 0.205 | -0.376 |
| | Oropharynx | 0.615 ±0.086 | | 0.505 | 0.738 | 0.179 | 0.677 | 0.003 | 0.179 | -0.371 |
| WDRO | ID | 0.578 ±0.063 | | 0.515 | 0.780 | 0.145 | 0.593 | 0.055 | 0.145 | -0.280 |
| | OOD | 0.554 ±0.046 | | 0.497 | 0.847 | 0.090 | 0.559 | 0.025 | 0.085 | -0.173 |
| | Oropharynx | 0.556 ±0.043 | | 0.494 | 0.822 | 0.107 | 0.569 | 0.010 | 0.096 | -0.187 |
| WDRO-MRO (ours) | ID | **0.684 ±0.028** | | **0.646** | **0.895** | **0.076** | **0.715** | 0.002 | **0.076** | **-0.141** |
| | OOD | **0.661 ±0.032** | | **0.621** | **0.907** | **0.064** | **0.671** | 0.007 | **0.060** | **-0.125** |
| | Oropharynx | **0.681 ±0.023** | | **0.655** | **0.929** | **0.050** | **0.697** | **0.002** | **0.050** | **-0.111** |

**Takeaway.** Across the aggregated evaluation results over random seeds and noise rates in Table 2, WDRO-MRO outperforms ERM, standard WDRO and group DRO(Figure 1). It achieves the highest average AUC with lower variance, improves robustness metrics (higher robust AUC and RR-AUC, smaller worst-case drop), and yields near-zero group fairness gap. These results demonstrate that

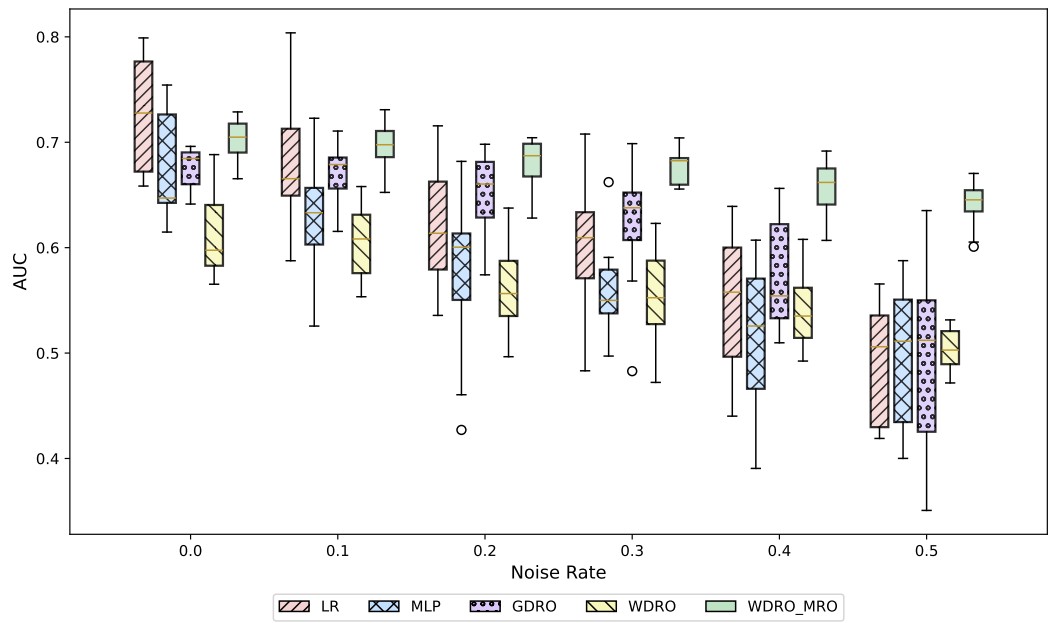

Figure 1: Boxplot of AUC across 3 data splits and 5 random seeds on the HANCOCK dataset. While LR achieves the highest AUC at $\rho = 0.0$, its performance degrades under noise. In contrast, WDRO_MRO maintains higher and more stable AUC distributions across noisy settings.

WDRO-MRO improves in performance, robustness, and fairness, whereas WDRO trades accuracy for conservativeness and ERM remains vulnerable to distribution shifts.

## 5    CONCLUSION AND FUTURE WORK

This paper introduces **WDRO-MRO**, a framework that unifies Wasserstein distributional robustness with minimax regret minimization to address multimodal learning under heterogeneous distributional shifts. By focusing on worst-case *regret* relative to the oracle predictor, WDRO-MRO provides a decision-centric notion of robustness that naturally connects performance and fairness within a tractable optimization framework. **Theory.** We establish a comprehensive foundation: worst-case distributions exist, minimax regret solutions are unique under strictly convex losses, and the objective is convex with strong duality. We further provide tractable reformulations (LP, SOCP, SDP, and power-cone programs) across a range of loss functions and $p$-Wasserstein norms, and design a dual-game solver (Alg. 1) that couples strong-dual reformulations with an exponentiated-weights adversary update, yielding an *oracle-free, no-regret* saddle-point scheme. These are supported by convergence guarantees, sensitivity analyses with respect to ambiguity radii and modality weights, and statistical guarantees including consistency, finite-sample bounds, and $O(N^{-1/2})$ convergence rates. **Practice.** On the HANCOCK multimodal dataset, WDRO-MRO demonstrates the strongest robustness to label noise with higher median AUC and lower variability across seeds and noise rates, and consistently outperforms both baselines on the Oropharynx split. **Outlook.** Future research directions include: (i) developing scalable stochastic and distributed solvers for large-scale multimodal data, (ii) extending the framework to nonconvex deep fusion models with approximate regret guarantees, (iii) exploring integration with generative and retrieval-augmented systems, and (iv) learning modality weights in a bilevel fashion to better trade off robustness and utility. Together, these directions point toward more reliable and interpretable multimodal AI systems built on minimax regret principles.

## 6 ETHICS STATEMENT AND REPRODUCIBILITY STATEMENT

### 6.1 ETHICS STATEMENT

This study uses only de-identified, publicly released data from the HANCOCK dataset. The original data collection was approved by the local ethics committee. The HANCOCK article reports that informed consent was waived because the data are retrospective, and it details the de-identification steps applied to clinical tables, blood measurements, pathology metadata, and surgery reports. We did not access any identifiable information, and we did not attempt re-identification.

### 6.2 REPRODUCIBILITY STATEMENT

We provide code, data processing pipelines, experimental settings, and theoretical derivations to ensure reproducibility: **Code and configurations.** All model training and evaluation scripts are submitted into supplementary materials together with environment files (env.yml). Scripts to regenerate every table and figure in the manuscript from raw logs are included. **Data access and randomness.** Our experiments are based on the publicly available HANCOCK dataset. All experiments are repeated with five random seeds. **Proofs.** Full derivations of the objectives and convex reformulations are provided in the Appendices E to I, together with convergence analysis of the dual-game solver. These materials enable reproduction of our results and validation of the theoretical components.

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

# A  NOTATION

Table 3: Summary of main notation used in the paper.

| Symbol | Description |
|---|---|
| $N$ | Number of training samples |
| $z = (x, y) \in \mathcal{Z}$ | Multimodal data point (features $x$ and label $y$) |
| $\hat{P}_N = \frac{1}{N} \sum_{i=1}^{N} \delta_{\hat{z}_i}$ | Empirical distribution |
| $P_0$ | Ground-truth data distribution on $\mathcal{Z}$ |
| $K$ | Number of modalities |
| $x = (x_1, \ldots, x_K)$ | Multimodal feature vector |
| $F_m$ | Linear map for modality $m$ in affine model $f(z) = \sum_{m=1}^{K} F_m z_m + g$ |
| $g$ | Bias vector in the affine model $f$ |
| $y \in \{\pm 1\}$ | Binary label in the logistic example |
| $d_k(\cdot, \cdot)$ | Ground metric on modality $k$ in the cost $c(z, z')$ |
| $c(z, z') = \sum_{k=1}^{K} \alpha_k d_k(z_k, z'_k)$ | Multimodal transport cost |
| $W_p(P, Q)$ | Order-$p$ Wasserstein distance between $P$ and $Q$ |
| $\mathcal{U}_\rho(\hat{P}_N)$ | Wasserstein ambiguity set centered at $\hat{P}_N$ |
| $\rho$ | Radius of the Wasserstein ambiguity set |
| $\Delta([N])$ | Probability simplex $\{w \in \mathbb{R}_+^N : \sum_{i=1}^{N} w_i = 1\}$ |
| $w_t \in \Delta([N])$ | Nature weights at iteration $t$ in the dual game |
| $\lambda$ | Dual variable for the Wasserstein radius constraint |
| $\lambda_{\max}$ | Upper bound for $\lambda$ in the projection $\Pi_{[0, \lambda_{\max}]}$ |
| $\sigma_k^2$ | Variance proxy (second-moment bound) for modality $k$ |
| $\sigma^2 = \sum_{k=1}^{K} \alpha_k^2 \sigma_k^2$ | Aggregate variance proxy in the generalization bounds |
| $L_\ell$ | Lipschitz constant of the loss in its prediction argument |
| $f \in \mathcal{F}$ | Predictor (e.g., multimodal fusion network) |
| $\ell(z, f(z))$ | Loss of predictor $f$ at sample $z$ |
| $R_Q(f) = \mathbb{E}_Q[\ell(z, f(z))]$ | Risk of $f$ under distribution $Q$ |
| $\text{Regret}_Q(f)$ | Regret $R_Q(f) - \inf_{f' \in \mathcal{F}} R_Q(f')$ |
| $\phi(f)$ | WDRO–MRO objective $\phi(f) = \sup_{Q \in B_\rho(\hat{P}_N)} \text{Regret}_Q(f)$ |
| $s_i(f, \lambda)$ | Dual envelope for sample $\hat{z}_i$: $s_i(f, \lambda) = \sup_{z'} \ell(\hat{z}_i, f(z')) - \lambda c(\hat{z}_i, z')$ |
| $T$ | Number of iterations in the dual-game solver |
| $\eta, \eta_\lambda$ | Step sizes for nature and radius-dual updates |

# B  TRACTABLE REFORMULATIONS FOR GENERAL $p$

**General Convex Loss:** $\ell(z, v)$ **proper, l.s.c., bounded in** $[0, M]$, $L$**-Lipschitz in** $v$.

*Lemma* B.1 ($p = 1$). With $c(z, z') = \sum_{m=1}^{K} \alpha_m \|z_m - z'_m\|_1$, the canonical objective is subject to SDP (or LP) constraints: $s_i \geq \inf_{u \in \mathbb{R}^{\dim(v)}} \ell^*(\hat{z}_i, u) + \lambda c^*(\hat{z}_i, -u/\lambda)$. For $L$-Lipschitz $\ell$, this reduces to linear constraints $s_i \geq \ell(\hat{z}_i, f(\hat{z}_i)) + L\lambda c(\hat{z}_i, z')$.

*Lemma* B.2 ($p = 2$). With $c(z, z') = \sum_{m=1}^{K} \alpha_m \|z_m - z'_m\|_2^2$, the canonical objective is subject to SDP constraints: $s_i \geq \inf_{u \in \mathbb{R}^{\dim(v)}} \ell^*(\hat{z}_i, u) + \lambda c^*(\hat{z}_i, -u/\lambda)$, where $\ell^*(z, u)$ is representable via the S-lemma, and $c^*(z, u) = \sum_m \frac{1}{4\alpha_m} \|u_m\|_2^2 + u_m^\top z_m$.

*Lemma* B.3 ($2 < p < \infty$). With $c(z, z') = \sum_{m=1}^{K} \alpha_m \|z_m - z'_m\|_p^p$, the canonical objective is subject to convex program constraints: $s_i \geq \inf_{u \in \mathbb{R}^{\dim(v)}} \ell^*(\hat{z}_i, u) + \lambda c^*(\hat{z}_i, -u/\lambda)$, where $c^*(z, u)$ is representable via power cones (rational $p$) or exponential cones (irrational $p$).

*Lemma* B.4 ($p = \infty$). With $c(z, z') = \sum_{m=1}^{K} \alpha_m \|z_m - z'_m\|_\infty$, the canonical objective is subject to LP/SDP constraints: $s_i \geq \inf_{u \in \mathbb{R}^{\dim(v)}} \ell^*(\hat{z}_i, u) + \lambda c^*(\hat{z}_i, -u/\lambda)$, where $\ell^*(z, u)$ is polyhedral for polyhedral support, and $c^*(z, u) = \sum_m u_m^\top z_m$ under $\ell_1$ bounds.

**Piecewise Linear Loss:** $\ell(z, v) = \max_{k=1,\ldots,J} (a_k^\top v + b_k)$.

*Lemma* B.5 ($p = 1$). With $c(z, z') = \sum_{m=1}^{K} \alpha_m \|z_m - z'_m\|_1$, the canonical objective is subject to linear constraints: $s_i \geq a_k^\top f(z') + b_k - \lambda \sum_{m=1}^{K} \sum_{j=1}^{\dim(\mathcal{Z}_m)} \alpha_m t_{i,k,m,j}$, $t_{i,k,m,j} \geq |z_{i,m,j} - z'_{m,j}|$, for all $i = 1, \ldots, N$, $k = 1, \ldots, J$, $m = 1, \ldots, K$, $j = 1, \ldots, \dim(\mathcal{Z}_m)$. This yields a LP.

*Lemma* B.6 ($p = 2$). With $c(z, z') = \sum_{m=1}^{K} \alpha_m \|z_m - z'_m\|_2^2$, the canonical objective is subject to SOCP constraints: $s_i \geq a_k^\top g + b_k - \lambda \sum_{m=1}^{K} \alpha_m \|\hat{z}_{i,m}\|_2^2 + \sum_{m=1}^{K} \frac{1}{4\lambda\alpha_m} \|a_k^\top F_m\|_2^2 + a_k^\top F \hat{z}_i$, for all $i = 1, \ldots, N$, $k = 1, \ldots, J$. This yields a SOCP.

*Lemma* B.7 ($2 < p < \infty$). With $c(z, z') = \sum_{m=1}^{K} \alpha_m \|z_m - z'_m\|_p^p$, the canonical objective is subject to power cone constraints: $s_i \geq a_k^\top f(z') + b_k - \lambda \sum_{m=1}^{K} \alpha_m t_{i,k,m}$, $\|\hat{z}_{i,m} - z'_m\|_p \leq t_{i,k,m}$, $t_{i,k,m} \geq 0$, for all $i = 1, \ldots, N$, $k = 1, \ldots, J$, $m = 1, \ldots, K$. This yields a convex program over power cones.

*Lemma* B.8 ($p = \infty$). With $c(z, z') = \sum_{m=1}^{K} \alpha_m \|z_m - z'_m\|_\infty$, the canonical objective is subject to vertex-enumeration constraints: $s_i \geq \max_{z' \in \mathcal{V}} \left[ a_k^\top f(z') + b_k \right]$, $\mathcal{V} = \{z' \in \mathcal{Z} : \sum_{m=1}^{K} \alpha_m \|\hat{z}_{i,m} - z'_m\|_\infty \leq \rho/\lambda\}$, for all $i = 1, \ldots, N$, $k = 1, \ldots, J$. This yields a LP.

**Quadratic Loss:** $\ell(z, v) = v^\top Q v + q^\top v + q_0$, $Q \succeq 0$.

*Lemma* B.9 ($p = 1$). With $c(z, z') = \sum_{m=1}^{K} \alpha_m \|z_m - z'_m\|_1$, the canonical objective is subject to SDP constraints: $\begin{pmatrix} Q & \frac{1}{2}(f(z') + q) \\ \frac{1}{2}(f(z') + q)^\top & s_i - q_0 + \lambda c(\hat{z}_i, z') \end{pmatrix} \succeq 0$, for all $i = 1, \ldots, N$. For diagonal $Q$, this reduces to SOCP constraints.

*Lemma* B.10 ($p = 2$). With $c(z, z') = \sum_{m=1}^{K} \alpha_m \|z_m - z'_m\|_2^2$, the canonical objective is subject to SDP constraints: $\begin{pmatrix} \lambda I & \frac{1}{2} \sum_{m=1}^{K} \alpha_m (F_m \hat{z}_{i,m} - f(z')) \\ \frac{1}{2} \left( \sum_{m=1}^{K} \alpha_m (F_m \hat{z}_{i,m} - f(z')) \right)^\top & s_i - q_0 - q^\top f(z') - \lambda \sum_{m=1}^{K} \alpha_m \|\hat{z}_{i,m}\|_2^2 \end{pmatrix} \succeq 0$, for all $i = 1, \ldots, N$. For diagonal $Q$, this reduces to SOCP constraints.

*Lemma* B.11 ($2 < p < \infty$). With $c(z, z') = \sum_{m=1}^{K} \alpha_m \|z_m - z'_m\|_p^p$, the canonical objective is subject to convex constraints: $s_i \geq \inf_{u \in \mathbb{R}^{\dim(v)}} \ell^*(\hat{z}_i, u) + \lambda c^*(\hat{z}_i, -u/\lambda)$, where $\ell^*(z, u)$ is representable via quadratic relaxation, and $c^*(z, u)$ via power or exponential cones depending on $p$.

*Lemma* B.12 ($p = \infty$). With $c(z, z') = \sum_{m=1}^{K} \alpha_m \|z_m - z'_m\|_\infty$, the canonical objective is subject to SDP constraints: $\begin{pmatrix} \lambda I & \frac{1}{2} \sum_{m=1}^{K} \alpha_m (F_m \hat{z}_{i,m} - f(z')) \\ \frac{1}{2} \left( \sum_{m=1}^{K} \alpha_m (F_m \hat{z}_{i,m} - f(z')) \right)^\top & s_i - q_0 - q^\top f(z') - \lambda \sum_{m=1}^{K} \alpha_m \|\hat{z}_{i,m}\|_\infty \end{pmatrix} \succeq 0$, for all $z' \in \mathcal{V}$, where $\mathcal{V}$ is the vertex set of the box uncertainty region.

## C  ORACLE-FREE DUAL-GAME SOLVER (SPECIALIZED TO LOGISTIC)

---

**Algorithm 2** WDRO–MRO Dual-Game Solver for Logistic Regression

---

**Require:** samples $\{(\hat{x}_i, y_i)\}_{i=1}^N$, radius $\rho$, stepsizes $\eta, \eta_\lambda$, projection bound $\lambda_{\max}$

1: Initialize nature weights $\pi_1(i) \leftarrow 1/N$, dual radius $\lambda_1 \geq 0$, predictors $w_1, w_1' \in \mathbb{R}^d$
2: **for** $t = 1, 2, \ldots, T$ **do**
3:     **Dual envelopes (common $\lambda_t$):** for each $i$, compute

$$s_i^t = s_i(w_t, \lambda_t), \qquad s_i'^t = s_i(w_t', \lambda_t),$$

   via the tractable LP/SDP/SOCP reformulations in Prop. 4.1
4:     **Nature update (no-regret):** let $\Delta_i \leftarrow s_i^t - s_i'^t$ (optionally mean-centered)

$$\pi_{t+1}(i) \leftarrow \frac{\pi_t(i)\exp(\eta\,\Delta_i)}{\sum_{j=1}^N \pi_t(j)\exp(\eta\,\Delta_j)}\,.$$

5:     **Learner / Oracle best-responses (same $\lambda_t$):**

$$w_{t+1} \in \arg\min_{w\in\mathbb{R}^d}\ \lambda_t\rho\ +\ \sum_{i=1}^N \pi_{t+1}(i)\,s_i(w, \lambda_t),$$

$$w_{t+1}' \in \arg\min_{w'\in\mathbb{R}^d}\ \lambda_t\rho\ +\ \sum_{i=1}^N \pi_{t+1}(i)\,s_i(w', \lambda_t)\,.$$

6:     **Radius dual update:**

$$\lambda_{t+1} \leftarrow \Pi_{[0,\lambda_{\max}]}\Big(\lambda_t + \eta_\lambda\big(\rho - \widehat{\rho}_t\big)\Big),$$

   where $\widehat{\rho}_t$ is the empirical dual subgradient (e.g., the average transport cost returned by the dual-envelope subproblems at $(w_{t+1}, \lambda_t)$).
7: **end for**
8: **Output:** averaged predictor $\bar{w} = \frac{1}{T}\sum_{t=1}^T w_t$

---

## D  RELATED WORK

### D.1  MULTIMODAL MACHINE LEARNING AND ROBUSTNESS CONSIDERATION

Multimodal machine learning (MML) investigates methods for learning from data that are represented in different modalities, such as images, text, audio (Yuan et al., 2025). Precision oncology is a particularly suitable application domain for MML, as patient data include medical images, radiological scans, multi-omics, and treatment histories (Zhou et al., 2024). Given that multimodal data are often noisy, incomplete, and imbalanced (Zhang et al., 2024b), ERM is not sufficient to handle the associated challenges.

**Robust Multimodal Learning.**  Qiu et al. (2022) evaluates the robustness of multimodal image–text models via 17 image perturbation and 16 text perturbation techniques. Among these, the character-level perturbation is the most effective for text, while zoom blur is the most effective for images. Yang et al. (2023) address robustness in multimodal finetuning by introducing four auxiliary losses-contrastive image and language losses, together with spurious-aware image and language losses-that use cross-moal signals to reduce reliance on spurious correlations. To mitigate bias in vision-language models, such as classifying "ants" with a "flower" background as "bees", Kim et al. (2024) propose the Bias-to-Text (B2T) framework. B2T extracts keywords from captions of misclassified images to interpret visual biases, and then assigns sample-wise bias labels. These inferred bias are incorporated into debiased training using a group DRO objective. Shuai et al. (2025) propose federated distributionally robust alignment framework to address client heterogeneity in medical data. They build a distribution family over client datasets and apply a DRO min-max objective to optimize the worst-case alignment risk.

To jointly handle multimodal and decision-dependent uncertainty, Yu & Basciftci (2024) propose a two-stage DRO framework in which the first stage chooses "here-and-now" decisions (e.g., which facilities to open) that are allowed to shift both the mixture weights (mode probabilities) and the per-mode distributions of future uncertainty. In the second stage, after the uncertain parameters (e.g., customer demand) are revealed, recourse actions are taken (e.g., determining how much demand to serve from each open facility) to minimize the resulting cost. They introduce a decision-dependent multimodal ambiguity set and use strong duality together with McCormick linearization to derive MILP/MISOCP reformulations that can be solved by existing solvers. These challenges motivate robust optimization frameworks like WDRO-MRO, which address modality-specific distributional shifts to ensure reliable performance.

## D.2    DISTRIBUTIONALLY ROBUST OPTIMIZATION

The pioneering work in distributionally robust optimization was introduced by Scarf (1958) in the newsvendor problem with an unknown exact demand distribution. The proposed min-max decision rule maximizes the expected profit under the worst-case distribution. For minimizing the worse-case risk, Namkoong & Duchi (2016) proposed the stochastic gradient in $f$-divergence DRO to improve efficiency. Based on the DRO idea, Shafieezadeh-Abadeh et al. (2019) proposed new regularization techniques using the Wasserstein distance and provided probabilistic interpretations of existing regularization methods. A tutorial on the theory and applications of Wasserstein-DRO in machine learning can be found in Kuhn et al. (2019). Motivated by the limitations of $\phi$-divergence ball fails to contain the true data distribution, while Wasserstein balls scale poorly with dimension, Staib & Jegelka (2019) introduced DRO based on the Maximum Mean Discrepancy (MMD). They proved that MMD-DRO is equivalent, up to small constants, to regularizing the empirical risk by the reproducing kernel Hilbert space norm of the loss function rather than the model itself. Since $\phi$-divergence measures only the relative probabilistic density ratio at identical support points, it ignores the metric between outcomes in the underlying metric space; consequently, it may exclude realistic distributions or include implausibly extreme ones, as illustrate in Example 1 (in Gao & Kleywegt (2023)). To overcome this limitation, Gao & Kleywegt (2023) use the Wasserstein distance to define the ambiguity set in distributionally robust stochastic optimization (DRSO) and derive the strong duality. Wu et al. (2023) use DRO to understand contrastive learning is equivalent to performing DRO over the negative-sample distribution, minizing the worst-case expected loss within a KL-divergence ball around the empirical distribution. The temperature parameter is not a heuristic constant but is the Lagrange multiplier that explicitly controls the radius of the uncertainty set. While DRO minimizes worst-case risk, it could be too conservative (Agarwal & Zhang, 2022); this motivates a shift to minimax regret optimization (MRO), which targets worst-case *regret* under the distributional uncertainty. WDRO-MRO overcomes this by minimizing worst-case regret, offering a less conservative, decision-centric approach for multimodal settings.

## D.3    MINIMAX REGRET OPTIMIZATION

Given that the risk is sensitive to heterogeneous noise, Agarwal & Zhang (2022) propose minimax regret optimization (MRO) using weight-based formulations to address distribution shift. This MRO formulation is less conservative than standard DRO, since it avoids overweighting distributions with intrinsically higher noise levels. However, a limitation of MRO is its computational demands: the empirical objective requires repeatedly solving inner ERM problems, which is impractical in large-scale settings. To address the computational bottleneck, Zhang et al. (2024a) present an efficient stochastic approximation of MRO via stochastic mirror descent with biased but controlled gradient estimates, which achieves near-optimal convergence rates. Beyond first-order methods, Gu & Xu (2024) develop zeroth-order stochastic mirror descent algorithms that rely solely on function evaluations. They prove $\mathcal{O}(1/\sqrt{t})$ convergence rate as well as $\mathcal{O}(1/\sqrt{t})$ optimization error. The minimax regret principle has been applied to causal inference with heterogeneous treatment effects. Zhang et al. (2024c) study the problem of aggregating conditional average treatment effect (CATE) estimates across multiple sites. Under assumptions that target-population CATEs lie in the convex hull of site-specific CATEs and that target covariate distributions are identifiable, the authors derive a closed-form minimax regret estimator. This estimator corresponds to a weighted average of site-level CATEs, with weights depending only on within-site estimates, thereby enabling robust generalization to unseen target populations without requiring individual-level data sharing. To minimize ex-ante

expected regret under distributional uncertainty, Fiechtner & Blanchet (2025) presents the Wasserstein distributionally robust regret optimization (DRRO). They prove that under smoothness and regularity conditions, the DRRO solution is consistent with ERM up to first-order terms, and exactly matches ERM for convex quadratic losses. For the classical newsvendor problem, regret has a closed-form characterization via maximizing two one-dimensional concave functions. For general max-affine losses, they show that regret evaluation is NP-hard and propose a convex relaxation with a provably tighter bound on the optimality gap.

# E  Proofs of Section 2

## E.1  Proof of Proposition 2.1

*Proof sketch.* By the Interchangeability Principle on Polish spaces, the supremum moves inside the expectation even under mild semicontinuity; see Kuhn et al. (2025, Lemma 4.16). $\square$

# F  Proofs of Section 3.1(Basic Optimization Properties)

## F.1  Proof of Proposition 3.1(Existence of Worst-Case Distribution)

*Proof.* We establish existence by leveraging the compactness of the ambiguity set and continuity properties, then characterize via duality.

**Existence.** The ambiguity set $\mathcal{U}_\rho(\hat{P}_N)$ is compact in the weak topology $\sigma(\mathcal{M}(\mathcal{Z}), C_b(\mathcal{Z}))$ (Villani et al., 2008), as it is closed (by lower semicontinuity of $c$) and tight (finite support of $\hat{P}_N$ implies Prohorov's theorem applies) (Billingsley, 2013).

For fixed $f$, $\mathrm{Regret}_Q(f) = \mathbb{E}_Q[\ell(z, f(z))] - \inf_{f' \in \mathcal{F}} \mathbb{E}_Q[\ell(z, f'(z))]$. Define $\ell_f(z) := \ell(z, f(z))$ and $\underline{\ell}(z) := \inf_{f' \in \mathcal{F}} \ell(z, f'(z))$. By Assumption 2.2, $\ell_f(z)$ is continuous and bounded; by Assumption 2.3 (compactness) and IP (Assumption 2.1), $\underline{\ell}(z)$ is weakly continuous in $Q$ (Mohajerin Esfahani & Kuhn, 2018). Thus, $\mathrm{Regret}_Q(f)$ is weakly continuous in $Q$.

By Berge's maximum theorem (Berge, 1877), as $\mathcal{U}_\rho(\hat{P}_N)$ is compact and $\mathrm{Regret}_Q(f)$ continuous, the supremum is attained.

**Characterization.** By Kantorovich-Rubinstein duality for multimodal costs (extended via separability: $c(z, z') = \sum_k \alpha_k d_k(z_k, z'_k)$) (Zhang et al., 2025; Mohajerin Esfahani & Kuhn, 2018), under Assumption 2.1 (convexity, lsc of costs) and IP,

$$\sup_{Q \in \mathcal{U}_\rho(\hat{P}_N)} \mathbb{E}_Q[\ell_f(z)] = \inf_{\lambda \geq 0} \lambda \rho + \mathbb{E}^{\hat{P}_N}\left[\sup_{z'} \ell_f(z') - \lambda c(\hat{z}, z')\right].$$

The dual attains at $\lambda^\star$, yielding optimal transport plan $\pi^\star$ minimizing transport cost for mass from $\hat{P}_N$ to $Q^\star$, with $\pi^\star(\hat{z}, z') > 0$ only if $z'$ maximizes $\ell_f(z') - \lambda^\star c(\hat{z}, z')$.

Similarly for the infimum term. The regret supremum is attained at $Q^\star$ induced by $\pi^\star$ respecting weighted $\alpha_k d_k$ (modality-specific metrics) Kuhn et al. (2019). Then existence of $Q^\star$ follows from compactness and continuity. $\square$

## F.2  Proof of Proposition 3.2(Convexity of the Problem)

*Proof.* We establish convexity and strong convexity leveraging the additive structure from modalities and the convexity of the ambiguity set.

**Convexity.** For fixed $Q \in \mathcal{U}_\rho(\hat{P}_N)$, consider $R_Q(f) = \mathbb{E}_Q[\ell(z, f(z))]$. By Assumption 2.2, $\ell(z, v)$ is convex in $v$, and additive across modalities: $\ell(z, v) = \sum_k \ell_k(z_k, v)$ with each $\ell_k$ convex. As $f(z)$ is affine in $f$ (linear composition), and expectation preserves convexity Rahimian & Mehrotra (2022), $R_Q(f)$ is convex in $f$.

The regret $\mathrm{Regret}_Q(f) = R_Q(f) - \inf_{f' \in \mathcal{F}} R_Q(f')$ is convex in $f$, since the infimum term is constant for fixed $Q$.

The ambiguity set $\mathcal{U}_\rho(\hat{P}_N)$ is convex Kuhn et al. (2019), as the Wasserstein ball is convex under convex transportation cost $c(z, z')$ (Assumption 2.1). The pointwise supremum over a convex set preserves convexity (Rockafellar, 1970), so $\phi(f) = \sup_Q \text{Regret}_Q(f)$ is convex in $f$.

**Strong Convexity.** Assume $\ell(z, v)$ is strongly convex in $v$ with modulus $\kappa > 0$. Then, each modality-specific $\ell_k(z_k, v)$ is strongly convex, implying overall strong convexity of $\ell$. Thus, $R_Q(f)$ is strongly convex in $f$ with modulus $\kappa$ (strong convexity preserved under affine composition and expectation) (Rahimian & Mehrotra, 2022).

$\text{Regret}_Q(f)$ inherits strong convexity, as the subtracted term is constant. The supremum over $Q$ preserves strong convexity (Zhang et al., 2025), yielding $\phi(f)$ strongly convex in $f$. $\qquad\square$

### F.3 PROOF OF PROPOSITION 3.3(EXISTENCE AND UNIQUENESS OF SOLUTIONS)

*Proof.* We proceed in two main steps: first, establish existence by proving lower semicontinuity of the objective on a compact domain; second, prove uniqueness via strict convexity.

**Existence.** By Assumption 2.3, $\mathcal{F}$ is convex and compact in the sup-norm topology (uniform topology) on $C(\mathcal{Z})$, the space of continuous functions on $\mathcal{Z}$ (Kuhn et al., 2019). It suffices to show $\phi$ is lower semicontinuous on $\mathcal{F}$; then, by Weierstrass' theorem (Rockafellar, 1970), the minimum is attained.

By Proposition 3.1, for each $f$, $\sup_{Q \in \mathcal{U}_\rho(\hat{P}_N)} \text{Regret}_Q(f)$ is attained, ensuring $\phi(f)$ is well-defined as a maximum (not just supremum).

By Proposition 3.2, $\phi(f)$ is convex, and thus continuous on the interior of $\mathcal{F}$. Lower semicontinuity on the boundary follows from the compactness of $\mathcal{U}_\rho(\hat{P}_N)$ and weak$^\star$ continuity of $\text{Regret}_Q(f)$ in $Q$ (as established in Proposition 3.1 proof), combined with joint continuity in $(f, Q)$ under boundedness (Assumption 2.2).

**Uniqueness.** Assume $\ell(z, v)$ strictly convex in $v$. Then, by Proposition 3.2, $\phi(f)$ is strictly convex on $\mathcal{F}$, yielding a unique minimizer (Gao et al., 2024). $\qquad\square$

### F.4 PROOF OF PROPOSITION 3.4(STRONG DUALITY)

*Proof.* By Proposition 3.3, the primal WDRO-MRO attains its infimum, ensuring the problem is well-posed for duality analysis.

We establish strong duality in the following steps: first, duality for the risk maximization under a fixed predictor; second, duality for the inner minimization over predictors; third, minimax interchange to form the dual regret formulation; and finally, finite-dimensionality and multimodal extension.

**Duality for the risk term under fixed $f$.** For fixed $f \in \mathcal{F}$, the risk term is $R_Q(f) = \mathbb{E}_Q[\ell(z, f(z))]$. Define $\ell_f(z) := \ell(z, f(z))$, which is convex in $z$ by Assumption 2.2 (as $\ell(z, v)$ is convex in $v$ and $f(z)$ is affine in $z$ under multimodal fusion). By the generalized Kantorovich-Rubinstein duality for separable costs $c(z, z') = \sum_k \alpha_k d_k(z_k, z'_k)$ (where $d_k$ are metrics on $\mathcal{Z}_k$), which holds under Assumption 2.1 (convex, non-negative, lower semicontinuous, modality-additive) and Assumption 2.1 (ensuring measurability and interchange), we have

$$\sup_{Q \in \mathcal{U}_\rho(\hat{P}_N)} \mathbb{E}_Q[\ell_f(z)] = \inf_{\lambda \geq 0} \lambda\rho + \mathbb{E}^{\hat{P}_N}\left[\sup_{z' \in \mathcal{Z}} (\ell_f(z') - \lambda c(\hat{z}, z'))\right],$$

with zero duality gap (see (Zhang et al., 2025, Theorem 1) for general costs and IP ensuring strong duality; the multimodal separability follows from additive convexity in Assumption 2.2 and cost structure). By Proposition 3.1, this sup is attained at some $Q^\star$, ensuring the primal maximum equals the dual minimum.

**Duality for the oracle infimum term.** The term $\inf_{f' \in \mathcal{F}} R_Q(f')$ is $\inf_{f' \in \mathcal{F}} \mathbb{E}_Q[\ell(z, f'(z))]$. By Assumption 2.3 ($\mathcal{F}$ convex, compact), and IP (Assumption 2.1), interchange holds: $\inf_{f'} \mathbb{E}_Q[\ell(z, f'(z))] = \mathbb{E}_Q[\inf_{f'} \ell(z, f'(z))]$. Define $\underline{\ell}(z) := \inf_{f' \in \mathcal{F}} \ell(z, f'(z))$, which is concave in $z$ (as infimum of convex functions in $v$). Applying duality similarly,

$$\sup_{Q \in \mathcal{U}_\rho(\hat{P}_N)} \inf_{f' \in \mathcal{F}} R_Q(f') = \sup_{Q \in \mathcal{U}_\rho(\hat{P}_N)} \mathbb{E}_Q[\underline{\ell}(z)] = \inf_{\lambda' \geq 0} \lambda'\rho + \mathbb{E}^{\hat{P}_N}\left[\sup_{z'' \in \mathcal{Z}} (\underline{\ell}(z'') - \lambda'c(\hat{z}, z''))\right].$$

**Minimax interchange for regret formulation.** Thus, $\sup_Q \text{Regret}_Q(f) = \sup_Q R_Q(f) - \sup_Q \inf_{f'} R_Q(f')$. By Sion's minimax theorem (Sion, 1958) (under compactness of $\mathcal{F}$, convexity in $f$ from Proposition 3.2, and quasiconcavity in $Q$ from separability and convexity), interchange yields zero gap: $\inf_f \sup_Q \text{Regret}_Q(f) = \sup_Q \inf_f \text{Regret}_Q(f)$. For fixed $f$, the regret supremum is

$$\inf_{\lambda \geq 0} \lambda \rho + \mathbb{E}^{\hat{P}_N}\left[\sup_{z'} \ell(z, f(z')) - \lambda c(\hat{z}, z')\right] - \inf_{\lambda' \geq 0} \lambda' \rho + \mathbb{E}^{\hat{P}_N}\left[\sup_{z''} \underline{\ell}(z'') - \lambda' c(\hat{z}, z'')\right].$$

**Finite-dimensionality and multimodal extension.** Finite-dimensionality follows from empirical measure (discrete support) and dual variables $\lambda, \lambda'$. The multimodal extension holds as costs and losses are additive across modalities, preserving separability in duality (see (Kuhn et al., 2019, Theorem 1) for extensions to structured costs). $\qquad\square$

## G   PROOFS OF SECTION 3.2(COMPUTATIONAL PROPERTIES)

### G.1   PROOF OF LEMMA B.5($p = 1$ , PIECEWISE LINEAR LOSS)

*Proof.* By Proposition 3.4, for fixed $f \in \mathcal{F}$, $\sup_{Q \in \mathcal{U}_\rho(\hat{P}_N)} \text{Regret}_Q(f)$ equals

$$\inf_{\lambda \geq 0} \lambda \rho + \mathbb{E}^{\hat{P}_N}\left[\sup_{z' \in \mathcal{Z}} \left(\ell(z, f(z')) - \lambda c(\hat{z}, z')\right) - \inf_{f' \in \mathcal{F}} \sup_{z'' \in \mathcal{Z}} \left(\ell(z, f'(z'')) - \lambda c(\hat{z}, z'')\right)\right],$$

with zero duality gap. This incorporates Sion's minimax interchange for the inf-sup in the regret term, justified by compactness and convexity (Assumption 2.3 and Proposition 3.2).

We derive the LP reformulation in the following steps: first, introduction of epigraph variables for the sup terms; second, exploitation of the piecewise linear structure and max-sup interchange; third, analogous dualization of the inf term; fourth, linearization of the transportation cost using auxiliary variables; and finally, assembly of the full LP and verification of its properties including convexity and zero duality gap.

**Introduction of epigraph variables for the sup terms.** The sup terms attain by Proposition 3.1 (existence of worst-case $Q^\star$, implying attainment in dual variables).

For the first sup term, define $\ell_f(z') := \ell(\hat{z}, f(z')) = \max_{k=1,\ldots,J}(a_k^\top f(z') + b_k)$. Introduce epigraph variables $s_i \geq 0$ (one per sample $\hat{z}_i$):

$$\inf_{\lambda \geq 0, s_i \geq 0} \lambda \rho + \frac{1}{N} \sum_{i=1}^N s_i \quad \text{s.t.} \quad s_i \geq \sup_{z' \in \mathcal{Z}} \ell_f(z') - \lambda c(\hat{z}_i, z'), \quad \forall i.$$

This is equivalent by epigraph representation preserving convexity (Proposition 3.2; see (Boyd & Vandenberghe, 2004), Section 4.2).

**Exploitation of the piecewise linear structure and max-sup interchange.** Substitute the piecewise max:

$$s_i \geq \max_{k=1,\ldots,J} \sup_{z' \in \mathcal{Z}} \left(a_k^\top f(z') + b_k - \lambda c(\hat{z}_i, z')\right),$$

equivalent to

$$s_i \geq \sup_{z' \in \mathcal{Z}} a_k^\top f(z') + b_k - \lambda c(\hat{z}_i, z'), \quad \forall k,$$

by max-sup interchange (continuity and finite J; (Rockafellar, 1970), Corollary 37.3.2).

**Analogous dualization of the inf term.** The inf term dualizes similarly, replacing $f$ with $f'$ and using primed variables.

**Linearization of the transportation cost using auxiliary variables.** For each $k$, $c(\hat{z}_i, z') = \sum_{m=1}^K \sum_{j=1}^{\dim(\mathcal{Z}_m)} \alpha_m |\hat{z}_{i,m,j} - z'_{m,j}|$. Introduce $t_{i,k,m,j} \geq 0$:

$$\sup_{z'} a_k^\top f(z') + b_k - \lambda c(\hat{z}_i, z') = \inf_{t_{i,k,m,j} \geq 0} a_k^\top f(z') + b_k - \lambda \sum_{m,j} \alpha_m t_{i,k,m,j}$$

s.t.

$$t_{i,k,m,j} \geq \hat{z}_{i,m,j} - z'_{m,j}, \quad t_{i,k,m,j} \geq z'_{m,j} - \hat{z}_{i,m,j}, \quad \forall m, j.$$

This linearizes the absolute values, equivalent by non-negativity and boundedness (compact $\mathcal{Z}$; (Boyd & Vandenberghe, 2004), Section 3.1.7).

Substitute: $s_i \geq a_k^\top f(z') + b_k - \lambda \sum_{m,j} \alpha_m t_{i,k,m,j}, \forall k$, with t-constraints. The inf over t attains by Slater (strict feasibility) and Proposition 3.3.

The full reformulation is the stated LP. Convexity follows from linear objective/constraints and Proposition 3.2. Zero gap holds by Proposition 3.4, with optima attained per Proposition 3.3. $\qquad\square$

### G.2  PROOF OF LEMMA B.6($p = 2$ , PIECEWISE LINEAR LOSS)

*Proof.* By Proposition 3.4 (Section 3.1), the regret supremum equals

$$\sup_{Q \in \mathcal{U}_\rho(\hat{P}_N)} \mathrm{Regret}_Q(f) = \inf_{\lambda \geq 0} \lambda\rho + \mathbb{E}^{\hat{P}_N}\left[\sup_{z' \in \mathcal{Z}} \left(\ell(z, f(z')) - \lambda c(\hat{z}, z')\right) - \inf_{f' \in \mathcal{F}} \sup_{z'' \in \mathcal{Z}} \left(\ell(z, f'(z'')) - \lambda c(\hat{z}, z'')\right)\right],$$

with zero duality gap, justified by compactness and convexity (Assumption 2.3 and Proposition 3.2). The sup terms attain by Proposition 3.1.

We derive the SOCP reformulation in the following steps: introduction of epigraph variables; computation of closed-form sup for piecewise linear loss; and representation of quadratic terms as SOCP constraints.

**Introduction of epigraph variables.** Define $\ell_f(z') := \ell(\hat{z}, f(z')) = \max_{k=1,\ldots,J}(a_k^\top f(z') + b_k)$. Introduce epigraph variables $s_i \in \mathbb{R}$, with dual variable $\lambda \geq 0$:

$$\inf_{\lambda \geq 0, s_i} \lambda\rho + \frac{1}{N}\sum_{i=1}^N s_i \quad \text{s.t.} \quad s_i \geq \sup_{z' \in \mathcal{Z}} \ell_f(z') - \lambda c(\hat{z}_i, z'), \quad \forall i.$$

This is equivalent by epigraph representation preserving convexity (Proposition 3.2; see (Boyd & Vandenberghe, 2004), Section 4.2). The inf term is analogous with primed variables $(\lambda', s'_i)$, replacing $f$ with $f'$.

**Computation of closed-form sup for piecewise linear loss.** The compactness of $\mathcal{Z}$ (Assumption 2.3) ensures the sup is attained. For the constraint $s_i \geq \sup_{z'} \ell_f(z') - \lambda c(\hat{z}_i, z')$, we have

$$s_i \geq \max_{k=1,\ldots,J} \sup_{z' \in \mathcal{Z}} \left(a_k^\top f(z') + b_k - \lambda c(\hat{z}_i, z')\right).$$

For affine $f(z') = \sum_m F_m z'_m + g$, the affine form ensures finite suprema. Define $a_k^\top f(z') = \sum_m l_{k,m}^\top z'_m + c_k$, where $l_{k,m} = F_m^\top a_k$, $c_k = a_k^\top g$, and $c(\hat{z}_i, z') = \sum_m \alpha_m \|\hat{z}_{i,m} - z'_m\|_2^2$. The weights $\alpha_m$ scale the quadratic terms, reflecting heterogeneous robustness. By max-sup interchange (continuity and finite $J$; (Rockafellar, 1970), Corollary 37.3.2), this becomes

$$s_i \geq \max_{k=1,\ldots,J} \left[\sup_{z' \in \mathcal{Z}} \left(\sum_{m=1}^K l_{k,m}^\top z'_m + c_k - \lambda \sum_{m=1}^K \alpha_m \|\hat{z}_{i,m} - z'_m\|_2^2\right)\right].$$

For each $k$, compute the inner sup over $z'_m$:

$$\sup_{z'_m} l_{k,m}^\top z'_m - \lambda\alpha_m \|\hat{z}_{i,m} - z'_m\|_2^2.$$

Complete the square: let $x = z'_m - \hat{z}_{i,m}$, so

$$l_{k,m}^\top(x + \hat{z}_{i,m}) - \lambda\alpha_m\|x\|_2^2 = -\lambda\alpha_m\left\|x - \frac{l_{k,m}}{2\lambda\alpha_m}\right\|_2^2 + \frac{\|l_{k,m}\|_2^2}{4\lambda\alpha_m} + l_{k,m}^\top\hat{z}_{i,m}.$$

The supremum, attained at $x = \frac{l_{k,m}}{2\lambda\alpha_m}$, is

$$l_{k,m}^\top\hat{z}_{i,m} + \frac{1}{4\lambda\alpha_m}\|l_{k,m}\|_2^2.$$

Summing over modalities and including the constant term,

$$s_i \geq \max_{k=1,\ldots,J} \left[ c_k + \sum_{m=1}^{K} \left( l_{k,m}^{\top} \hat{z}_{i,m} + \frac{1}{4\lambda\alpha_m} \|l_{k,m}\|_2^2 \right) \right].$$

The inf term reformulates similarly with primed variables $(\lambda', s_i')$, replacing $f$ with $f'$.

**Representation of quadratic terms as SOCP constraints.** Each quadratic term $\frac{1}{4\lambda\alpha_m}\|l_{k,m}\|_2^2 \leq t$ is SOCP-representable via the rotated quadratic cone (see (Boyd & Vandenberghe, 2004), Section 4.4.2). Let $u = l_{k,m}/\sqrt{4\lambda\alpha_m}$, so

$$\|u\|_2^2 \leq t \iff \|(t-1, 2u)\|_2 \leq t+1.$$

The infimum over auxiliary variables attains due to Slater's condition, satisfied by the compactness of $\mathcal{Z}$ (Assumption 2.3). Thus, the full WDRO-MRO reformulates as the stated SOCP, which is convex due to linear objectives and conic constraints (Proposition 3.2). Strong duality holds with zero gap by Proposition 3.4, with optima attained per Proposition 3.3. $\qquad\square$

### G.3 PROOF OF LEMMA B.7($2 < p < \infty$, PIECEWISE LINEAR LOSS)

*Proof.* By Proposition 3.4 (Section 3.1), the regret supremum equals

$$\sup_{Q \in \mathcal{U}_\rho(\hat{P}_N)} \mathrm{Regret}_Q(f) = \inf_{\lambda \geq 0} \lambda\rho + \mathbb{E}^{\hat{P}_N} \left[ \sup_{z' \in \mathcal{Z}} \left( \ell(z, f(z')) - \lambda c(\hat{z}, z') \right) - \inf_{f' \in \mathcal{F}} \sup_{z'' \in \mathcal{Z}} \left( \ell(z, f'(z'')) - \lambda c(\hat{z}, z'') \right) \right],$$

with zero duality gap, justified by compactness and convexity (Assumption 2.3 and Proposition 3.2). The sup terms attain by Proposition 3.1. The compactness of $\mathcal{Z}$ (Assumption 2.3) ensures finite suprema.

We derive the power cone reformulation in the following steps: introduction of epigraph variables; reformulation of the epigraph constraint via Fenchel-Moreau theorem; conjugate computation for the transportation cost; representation of constraints as power cones; analogous reformulation of the inf term; and assembly of the full program and verification of its properties.

**Introduction of epigraph variables.** Define $\ell_f(z') := \ell(\hat{z}, f(z')) = \max_{k=1,\ldots,J}(a_k^{\top} f(z') + b_k)$. Introduce epigraph variables $s_i \in \mathbb{R}$, with dual variable $\lambda \geq 0$:

$$\inf_{\lambda \geq 0, s_i} \lambda\rho + \frac{1}{N} \sum_{i=1}^{N} s_i \quad \text{s.t.} \quad s_i \geq \sup_{z' \in \mathcal{Z}} \ell_f(z') - \lambda c(\hat{z}_i, z'), \quad \forall i.$$

This is equivalent by epigraph representation preserving convexity (Proposition 3.2; see (Boyd & Vandenberghe, 2004), Section 4.2). The inf term is analogous with primed variables $(\lambda', s_i')$, replacing $f$ with $f'$.

**Reformulation of the epigraph constraint via Fenchel-Moreau theorem.** For affine $f(z') = \sum_m F_m z_m' + g$, the affine form ensures finite suprema. The epigraph constraint is

$$s_i \geq \sup_{z' \in \mathcal{Z}} \ell(\hat{z}_i, f(z')) - \lambda c(\hat{z}_i, z'),$$

with $c(\hat{z}_i, z') = \sum_{m=1}^{K} \alpha_m \|\hat{z}_{i,m} - z_m'\|_p^p$. By Fenchel-Moreau theorem (Rockafellar, 1970) (Theorem 12.2; applies to proper convex l.s.c. $\ell$ by Assumption 2.2), rewrite as

$$\sup_{z'} \ell_f(z') - \lambda c(\hat{z}_i, z') = \inf_{u \in \mathbb{R}^{\dim(z)}} \ell^*(\hat{z}_i, u) + \lambda c^*(\hat{z}_i, -u/\lambda),$$

by Fenchel inf-convolution duality (Rockafellar, 1970) (Theorem 16.4; strong duality under relative interior conditions from compactness and Assumption 2.2 boundedness), where $\ell^*(z, u) = \sup_v u^{\top} v - \ell(z, v)$ and $c^*(z, u) = \sup_{z'} u^{\top} z' - c(z, z')$.

**Conjugate computation for the transportation cost.** For the cost $c(z, z') = \sum_{m=1}^{K} \alpha_m \|z_m - z_m'\|_p^p$, the conjugate is

$$c^*(z, u) = \sup_{z'} \sum_{m=1}^{K} u_m^{\top} z_m' - \sum_{m=1}^{K} \alpha_m \|z_m - z_m'\|_p^p.$$

For each modality $m$, compute

$$\sup_{z'_m} u_m^\top z'_m - \alpha_m \|z_m - z'_m\|_p^p = \sup_{x_m} u_m^\top (x_m + z_m) - \alpha_m \|x_m\|_p^p,$$

where $x_m = z'_m - z_m$. By Holder's inequality, the conjugate is

$$c^*(z, u) = \sum_{m=1}^K \inf_{t_m \geq 0} \left[ t_m^p + \frac{1}{p-1} \left( \frac{\|u_m\|_q}{\alpha_m t_m^{p-1}} \right)^q \right] + u_m^\top z_m,$$

a generalized Holder conjugate ( (Rockafellar, 1970), Theorem 15.3), where $q = p/(p-1)$. The weights $\alpha_m$ scale the terms, reflecting heterogeneous robustness across modalities.

**Representation of constraints as power cones.** For the piecewise linear loss $\ell_f(z') = \max_{k=1,\ldots,J} (a_k^\top f(z') + b_k)$, the conjugate $\ell^*(z, u) = \sup_v u^\top v - \max_k (a_k^\top v + b_k)$ is polyhedral, representable as linear Dirac deltas. Substituting into the epigraph constraint:

$$s_i \geq \max_{k=1,\ldots,J} \sup_{z' \in \mathcal{Z}} \left[ a_k^\top f(z') + b_k - \lambda \sum_{m=1}^K \alpha_m \|\hat{z}_{i,m} - z'_m\|_p^p \right].$$

Introduce auxiliary variables $t_{i,k,m} \geq 0$:

$$s_i \geq \max_{k=1,\ldots,J} \inf_{t_{i,k,m} \geq 0} a_k^\top f(z') + b_k - \lambda \sum_{m=1}^K \alpha_m t_{i,k,m}^p,$$

subject to

$$\|\hat{z}_{i,m} - z'_m\|_p \leq t_{i,k,m}, \quad \forall m.$$

This constraint is reformulated as a power cone $\{(u,t) : \|u\|_q \leq t\}$ via the Holder conjugate, representable as $\|(\hat{z}_{i,m} - z'_m, t_{i,k,m})\|_q \leq t_{i,k,m}$ ( (Ben-Tal & Nemirovski, 2001), Section 4.3). The infimum over $t_{i,k,m}$ attains due to Slater's condition, satisfied by the compactness of $\mathcal{Z}$ (Assumption 2.3).

**Analogous reformulation of the inf term.** The inf term reformulates similarly: replace $f$ with $f'$ in the power cone constraints, using primed variables $(\lambda', s'_i)$, and optimize over $f' \in \mathcal{F}$.

The full WDRO-MRO is the stated convex program over power cones, convex due to conic constraints (Proposition 3.2). Strong duality holds with zero gap by Proposition 3.4, with optima attained per Proposition 3.3 (Section 3.2). $\qquad\square$

### G.4 Proof of Lemma B.8($p = \infty$, Piecewise Linear Loss)

*Proof.* By Proposition 3.4 (Section 3.1), the regret supremum equals

$$\sup_{Q \in \mathcal{U}_\rho(\hat{P}_N)} \text{Regret}_Q(f) = \inf_{\lambda \geq 0} \lambda\rho + \mathbb{E}^{\hat{P}_N} \left[ \sup_{z' \in \mathcal{Z}} \left( \ell(z, f(z')) - \lambda c(\hat{z}, z') \right) - \inf_{f' \in \mathcal{F}} \sup_{z'' \in \mathcal{Z}} \left( \ell(z, f'(z'')) - \lambda c(\hat{z}, z'') \right) \right],$$

with zero duality gap, justified by compactness and convexity (Assumption 2.3 and Proposition 3.2). The sup terms attain by Proposition 3.1. The compactness of $\mathcal{Z}$ (Assumption 2.3) ensures finite suprema.

We derive the LP reformulation in the following steps: introduction of epigraph variables; reformulation of the epigraph constraint via Fenchel-Moreau theorem; conjugate computation for the transportation cost; vertex enumeration for box uncertainty set; analogous reformulation of the inf term; and assembly of the full program and verification of its properties.

**Introduction of epigraph variables.** Define $\ell_f(z') := \ell(\hat{z}, f(z')) = \max_{k=1,\ldots,J} (a_k^\top f(z') + b_k)$. Introduce epigraph variables $s_i \in \mathbb{R}$, with dual variable $\lambda \geq 0$:

$$\inf_{\lambda \geq 0, s_i} \lambda\rho + \frac{1}{N} \sum_{i=1}^N s_i \quad \text{s.t.} \quad s_i \geq \sup_{z' \in \mathcal{Z}} \ell_f(z') - \lambda c(\hat{z}_i, z'), \quad \forall i.$$

This is equivalent by epigraph representation preserving convexity (Proposition 3.2; see (Boyd & Vandenberghe, 2004), Section 4.2). The inf term is analogous with primed variables $(\lambda', s_i')$, replacing $f$ with $f'$.

**Reformulation of the epigraph constraint via Fenchel-Moreau theorem.** For affine $f(z') = \sum_m F_m z_m' + g$, the affine form ensures finite suprema. The epigraph constraint is

$$s_i \geq \sup_{z' \in \mathcal{Z}} \ell(\hat{z}_i, f(z')) - \lambda c(\hat{z}_i, z'),$$

with $c(\hat{z}_i, z') = \sum_{m=1}^K \alpha_m \|\hat{z}_{i,m} - z_m'\|_\infty$. By Fenchel-Moreau theorem (Rockafellar, 1970) (Theorem 12.2; applies to proper convex l.s.c. $\ell$ by Assumption 2.2), rewrite as

$$\sup_{z'} \ell_f(z') - \lambda c(\hat{z}_i, z') = \inf_{u \in \mathbb{R}^{\dim(z)}} \ell^*(\hat{z}_i, u) + \lambda c^*(\hat{z}_i, -u/\lambda),$$

by Fenchel inf-convolution duality (Rockafellar, 1970) (Theorem 16.4; strong duality under relative interior conditions from compactness and Assumption 2.2 boundedness), where $\ell^*(z, u) = \sup_v u^\top v - \ell(z, v)$ and $c^*(z, u) = \sup_{z'} u^\top z' - c(z, z')$.

**Conjugate computation for the transportation cost.** For the cost $c(z, z') = \sum_{m=1}^K \alpha_m \|z_m - z_m'\|_\infty$, the conjugate is

$$c^*(z, u) = \sup_{z'} \sum_{m=1}^K u_m^\top z_m' - \sum_{m=1}^K \alpha_m \|z_m - z_m'\|_\infty.$$

Since $\|z_m - z_m'\|_\infty = \max_j |z_{m,j} - z_{m,j}'|$, the conjugate is finite only if $\sum_m \|u_m\|_1 \leq \sum_m \alpha_m$, yielding

$$c^*(z, u) = \begin{cases} \sum_{m=1}^K u_m^\top z_m & \text{if } \sum_{m=1}^K \|u_m\|_1 \leq \sum_{m=1}^K \alpha_m, \\ \infty & \text{otherwise}, \end{cases}$$

a polyhedral conjugate ( (Rockafellar, 1970), Example 11.4). The weights $\alpha_m$ scale the terms, reflecting heterogeneous robustness across modalities.

**Vertex enumeration for box uncertainty set.** For the piecewise linear loss $\ell_f(z') = \max_{k=1,\ldots,J}(a_k^\top f(z') + b_k)$, the conjugate $\ell^*(z, u) = \sup_v u^\top v - \max_k(a_k^\top v + b_k)$ is polyhedral. Substituting into the epigraph constraint:

$$s_i \geq \sup_{z' \in \mathcal{Z}} \left[ \max_{k=1,\ldots,J}(a_k^\top f(z') + b_k) - \lambda \sum_{m=1}^K \alpha_m \|\hat{z}_{i,m} - z_m'\|_\infty \right].$$

The $W_\infty$ uncertainty set is a box: $\mathcal{V} = \{z' \in \mathcal{Z} : \sum_{m=1}^K \alpha_m \|\hat{z}_{i,m} - z_m'\|_\infty \leq \rho/\lambda\}$. Since $\ell_f(z')$ is piecewise linear and $\mathcal{V}$ is polyhedral, the supremum is attained at the vertices of $\mathcal{V}$ ( (Ben-Tal et al., 2009), Theorem 3.1). Thus,

$$s_i \geq \max_{z' \in \mathcal{V}} \max_{k=1,\ldots,J} \left[ a_k^\top f(z') + b_k \right],$$

yielding a finite-dimensional LP by enumerating the vertices of $\mathcal{V}$. The infimum over auxiliary variables attains due to Slater's condition, satisfied by the compactness of $\mathcal{Z}$ (Assumption 2.3).

**Analogous reformulation of the inf term.** The inf term reformulates similarly: replace $f$ with $f'$ in the LP constraints, using primed variables $(\lambda', s_i')$, and optimize over $f' \in \mathcal{F}$.

The full WDRO-MRO is the stated LP, convex due to linear constraints (Proposition 3.2). Strong duality holds with zero gap by Proposition 3.4, with optima attained per Proposition 3.1 (Section 3.2). $\qed$

G.5  PROOF OF LEMMA B.9($p = 1$, QUADRATIC LOSS)

*Proof.* By Proposition 3.4, for fixed $f \in \mathcal{F}$, $\sup_{Q \in \mathcal{U}_\rho(\hat{P}_N)} \text{Regret}_Q(f)$ equals

$$\inf_{\lambda \geq 0} \lambda \rho + \mathbb{E}^{\hat{P}_N} \left[ \sup_{z' \in \mathcal{Z}} (\ell(z, f(z')) - \lambda c(\hat{z}, z')) - \inf_{f' \in \mathcal{F}} \sup_{z'' \in \mathcal{Z}} (\ell(z, f'(z'')) - \lambda c(\hat{z}, z'')) \right],$$

with zero duality gap. This incorporates Sion's minimax interchange for the inf-sup in the regret term, justified by compactness and convexity (Assumption 2.3 and Proposition 3.2).

We derive the SDP (or SOCP) reformulation in the following steps: first, introduction of epigraph variables for the sup terms; second, reformulation of the epigraph constraint via completing the square; third, equivalence to PSD condition via Schur complement; fourth, reduction to SOCP for diagonal cases; fifth, linearization of the transportation cost using auxiliary variables; sixth, analogous reformulation of the inf term; and finally, assembly of the full program and verification of its properties including convexity and zero duality gap.

**Introduction of epigraph variables for the sup terms.** The sup terms attain by Proposition 3.1 (existence of worst-case $Q^\star$, implying attainment in dual variables).

For the first sup term, define $\ell_f(z') := \ell(\hat{z}, f(z')) = f(z')^\top Q f(z') + q^\top f(z') + q_0$. Introduce epigraph variables $s_i \in \mathbb{R}$ (one per sample $\hat{z}_i$):

$$\inf_{\lambda \geq 0, s_i \in \mathbb{R}} \lambda\rho + \frac{1}{N}\sum_{i=1}^N s_i \quad \text{s.t.} \quad s_i \geq \sup_{z' \in \mathcal{Z}} \ell_f(z') - \lambda c(\hat{z}_i, z'), \quad \forall i.$$

This is equivalent by epigraph representation preserving convexity (Proposition 3.2; see (Boyd & Vandenberghe, 2004), Section 4.2).

The inf term epigraph reformulates similarly, replacing $f$ with $f'$ and using primed variables.

**Reformulation of the epigraph constraint via completing the square.** The epigraph constraint is

$$s_i \geq \sup_{z' \in \mathcal{Z}} f(z')^\top Q f(z') + q^\top f(z') + q_0 - \lambda c(\hat{z}_i, z').$$

By compactness of $\mathcal{Z}$ and continuity, the sup attains. Complete the square: $f(z')^\top Q f(z') + q^\top f(z') + q_0 = (f(z') + Q^{-1}q/2)^\top Q (f(z') + Q^{-1}q/2) - (q^\top Q^{-1}q)/4 + q_0$ (assuming $Q \succ 0$; for semidefinite, use pseudoinverse and range conditions; (Rockafellar, 1970), Theorem 28.3).

**Equivalence to PSD condition via Schur complement.** The inequality $v^\top Q v + q^\top v + q_0 \leq s_i + \lambda c(\hat{z}_i, z')$ (with $v = f(z')$) is equivalent to the PSD condition via Schur complement lemma (Boyd & Vandenberghe, 2004) (Appendix A.5.5):

$$\begin{pmatrix} Q & \frac{1}{2}(f(z') + q) \\ \frac{1}{2}(f(z') + q)^\top & s_i - q_0 + \lambda c(\hat{z}_i, z') \end{pmatrix} \succeq 0,$$

since $Q \succeq 0$ ensures convexity (Assumption 2.2). This holds under the affine assumption on $f$, as $f(z')$ appears linearly in the off-diagonals.

**Reduction to SOCP for diagonal cases.** For diagonal $Q = \text{diag}(Q_{ll})$, the PSD reduces to SOCP:

$$s_i - q_0 + \lambda c(\hat{z}_i, z') \geq \| \text{diag}(\sqrt{Q})(f(z') + q/2)\|_2,$$

by separating quadratic terms into second-order cones $\{(x, t) : \|x\|_2 \leq t\}$ ( (Boyd & Vandenberghe, 2004), Section 4.4.2).

**Linearization of the transportation cost using auxiliary variables.** The $\ell_1$-norm in $c$ can be linearized by introducing auxiliary variables $t_{i,m,j} \geq 0$ (as in Lemma B.5), yielding SDP with additional linear constraints:

$$c(\hat{z}_i, z') = \inf_{t_{i,m,j} \geq 0} \sum_{m,j} \alpha_m t_{i,m,j} \quad \text{s.t.} \quad t_{i,m,j} \geq \hat{z}_{i,m,j} - z'_{m,j}, \quad t_{i,m,j} \geq z'_{m,j} - \hat{z}_{i,m,j}.$$

Substitute into the Schur off-diagonal or SOCP right-hand side.

**Analogous reformulation of the inf term.** The inf term reformulates similarly: replace $f$ with $f'$ in the SDP/SOCP constraints, using primed variables $\lambda', s'_i$, and optimize over $f' \in \mathcal{F}$.

The full WDRO-MRO is the stated SDP (or SOCP for diagonal $Q$). Convexity follows from semidefinite constraints preserving convexity and Proposition 3.2. Zero gap holds by Proposition 3.4, with optima attained per Proposition 3.3. $\qquad\square$

### G.6 PROOF OF LEMMA B.10($p = 2$, QUADRATIC LOSS)

*Proof.* By Proposition 3.4 (Section 3.1), the regret supremum equals

$$\sup_{Q \in \mathcal{U}_\rho(\hat{P}_N)} \mathrm{Regret}_Q(f) = \inf_{\lambda \geq 0} \lambda\rho + \mathbb{E}^{\hat{P}_N}\left[\sup_{z' \in \mathcal{Z}} \left(\ell(z, f(z')) - \lambda c(\hat{z}, z')\right) - \inf_{f' \in \mathcal{F}} \sup_{z'' \in \mathcal{Z}} \left(\ell(z, f'(z'')) - \lambda c(\hat{z}, z'')\right)\right],$$

with zero duality gap, justified by compactness and convexity (Assumption 2.3 and Proposition 3.2). The sup terms attain by Proposition 3.1.

We derive the SDP reformulation in the following steps: introduction of epigraph variables; computation of closed-form sup via Fenchel conjugate; and representation of constraints as SDP or SOCP.

**Introduction of epigraph variables.** Define $\ell_f(z') := \ell(\hat{z}, f(z')) = f(z')^\top Q f(z') + q^\top f(z') + q_0$. Introduce epigraph variables $s_i \in \mathbb{R}$, with dual variable $\lambda \geq 0$:

$$\inf_{\lambda \geq 0, s_i} \lambda\rho + \frac{1}{N}\sum_{i=1}^N s_i \quad \text{s.t.} \quad s_i \geq \sup_{z' \in \mathcal{Z}} \ell_f(z') - \lambda c(\hat{z}_i, z'), \quad \forall i.$$

This is equivalent by epigraph representation preserving convexity (Proposition 3.2; see (Boyd & Vandenberghe, 2004), Section 4.2). The inf term is analogous with primed variables $(\lambda', s_i')$, replacing $f$ with $f'$.

**Computation of closed-form sup via Fenchel conjugate.** The compactness of $\mathcal{Z}$ (Assumption 2.3) ensures the sup is attained. For the constraint $s_i \geq \sup_{z'} \ell_f(z') - \lambda c(\hat{z}_i, z')$, with $c(\hat{z}_i, z') = \sum_{m=1}^K \alpha_m \|\hat{z}_{i,m} - z_m'\|_2^2$, we have

$$s_i \geq \sup_{z' \in \mathcal{Z}}\left[f(z')^\top Q f(z') + q^\top f(z') + q_0 - \lambda \sum_{m=1}^K \alpha_m \|\hat{z}_{i,m} - z_m'\|_2^2\right].$$

For affine $f(z') = \sum_m F_m z_m' + g$, the affine form ensures finite suprema. By Fenchel-Moreau theorem (Rockafellar, 1970) (Theorem 12.2), rewrite the sup using Fenchel conjugates:

$$\sup_{z'} \ell_f(z') - \lambda c(\hat{z}_i, z') = \inf_{u \in \mathbb{R}^{\dim(z)}} \ell^*(\hat{z}_i, u) + \lambda c^*(\hat{z}_i, -u/\lambda),$$

where $\ell^*(z, u) = \sup_v u^\top v - \ell(z, v)$ and $c^*(z, u) = \sup_{z'} u^\top z' - c(z, z')$. For the quadratic loss $\ell(z, v) = v^\top Q v + q^\top v + q_0$, assuming $Q \succeq 0$, the conjugate is

$$\ell^*(z, u) = \sup_v \left[u^\top v - (v^\top Q v + q^\top v + q_0)\right] = \frac{1}{4}(u - q)^\top Q^{-1}(u - q) - q_0,$$

where $Q^{-1}$ is the pseudoinverse if $Q$ is singular ( (Rockafellar, 1970), Theorem 23.5). For the cost $c(z, z') = \sum_m \alpha_m \|z_m - z_m'\|_2^2$, the conjugate is

$$c^*(z, u) = \sup_{z'} \sum_m u_m^\top z_m' - \sum_m \alpha_m \|z_m - z_m'\|_2^2 = \sum_m \frac{1}{4\alpha_m}\|u_m\|_2^2 + u_m^\top z_m.$$

Thus, the epigraph constraint becomes

$$s_i \geq \inf_u \left[\frac{1}{4}(u - q)^\top Q^{-1}(u - q) - q_0 + \lambda \sum_m \frac{1}{4\lambda\alpha_m}\| - u_m/\lambda\|_2^2 - \sum_m \frac{u_m^\top \hat{z}_{i,m}}{\lambda}\right].$$

The inf term reformulates similarly with primed variables.

**Representation of constraints as SDP or SOCP.** Complete the square for the quadratic expression in $u$, and apply the Schur complement lemma (Boyd & Vandenberghe, 2004) (Appendix A.5.5) to obtain the SDP constraint:

$$\begin{pmatrix} \lambda I & \frac{1}{2}\sum_{m=1}^K \alpha_m(F_m \hat{z}_{i,m} - f(z')) \\ \frac{1}{2}\left(\sum_{m=1}^K \alpha_m(F_m \hat{z}_{i,m} - f(z'))\right)^\top & s_i - q_0 - q^\top f(z') - \lambda \sum_{m=1}^K \alpha_m \|\hat{z}_{i,m}\|_2^2 \end{pmatrix} \succeq 0.$$

For diagonal $Q = \text{diag}(Q_{ll})$, the constraint reduces to an SOCP:

$$s_i - q_0 - q^\top f(z') - \lambda \sum_{m=1}^{K} \alpha_m \|\hat{z}_{i,m}\|_2^2 \geq \| \text{diag}(\sqrt{Q})(f(z') + q/2)\|_2,$$

representable via the Lorentz cone $\{(x,t) : \|x\|_2 \leq t\}$ ( (Boyd & Vandenberghe, 2004), Section 4.4.2). The infimum over auxiliary variables attains due to Slater's condition, satisfied by the compactness of $\mathcal{Z}$ (Assumption 2.3). The weights $\alpha_m$ scale the quadratic terms, reflecting heterogeneous robustness. Thus, the full WDRO-MRO reformulates as the stated SDP (or SOCP for diagonal $Q$), which is convex due to semidefinite or conic constraints (Proposition 3.2). Strong duality holds with zero gap by Proposition 3.4, with optima attained per Proposition 3.3. □

### G.7 PROOF OF LEMMA B.11($2 < p < \infty$, QUADRATIC LOSS)

*Proof.* By Proposition 3.4 (Section 3.1), the regret supremum equals

$$\sup_{Q \in \mathcal{U}_\rho(\hat{P}_N)} \text{Regret}_Q(f) = \inf_{\lambda \geq 0} \lambda\rho + \mathbb{E}^{\hat{P}_N}\left[\sup_{z' \in \mathcal{Z}} \left(\ell(z, f(z')) - \lambda c(\hat{z}, z')\right) - \inf_{f' \in \mathcal{F}} \sup_{z'' \in \mathcal{Z}} \left(\ell(z, f'(z'')) - \lambda c(\hat{z}, z'')\right)\right],$$

with zero duality gap, justified by compactness and convexity (Assumption 2.3 and Proposition 3.2). The sup terms attain by Proposition 3.1. The compactness of $\mathcal{Z}$ (Assumption 2.3) ensures finite suprema.

We derive the reformulation in the following steps: introduction of epigraph variables; reformulation of the epigraph constraint via Fenchel-Moreau theorem; conjugate computation for the transportation cost; SDP approximation for quadratic and power terms via S-lemma; exponential cone representation for log-Holder approximation; analogous reformulation of the inf term; and assembly of the full program and verification of its properties.

**Introduction of epigraph variables.** Define $\ell_f(z') := \ell(\hat{z}, f(z')) = f(z')^\top Q f(z') + q^\top f(z') + q_0$. Introduce epigraph variables $s_i \in \mathbb{R}$, with dual variable $\lambda \geq 0$:

$$\inf_{\lambda \geq 0, s_i} \lambda\rho + \frac{1}{N} \sum_{i=1}^{N} s_i \quad \text{s.t.} \quad s_i \geq \sup_{z' \in \mathcal{Z}} \ell_f(z') - \lambda c(\hat{z}_i, z'), \quad \forall i.$$

This is equivalent by epigraph representation preserving convexity (Proposition 3.2; see (Boyd & Vandenberghe, 2004), Section 4.2). The inf term is analogous with primed variables $(\lambda', s_i')$, replacing $f$ with $f'$.

**Reformulation of the epigraph constraint via Fenchel-Moreau theorem.** For affine $f(z') = \sum_m F_m z_m' + g$, the affine form ensures finite suprema. The epigraph constraint is

$$s_i \geq \sup_{z' \in \mathcal{Z}} \ell(\hat{z}_i, f(z')) - \lambda c(\hat{z}_i, z'),$$

with $c(\hat{z}_i, z') = \sum_{m=1}^{K} \alpha_m \|\hat{z}_{i,m} - z_m'\|_p^p$. By Fenchel-Moreau theorem (Rockafellar, 1970) (Theorem 12.2; applies to proper convex l.s.c. $\ell$ by Assumption 2.2), rewrite as

$$\sup_{z'} \ell_f(z') - \lambda c(\hat{z}_i, z') = \inf_{u \in \mathbb{R}^{\dim(z)}} \ell^*(\hat{z}_i, u) + \lambda c^*(\hat{z}_i, -u/\lambda),$$

by Fenchel inf-convolution duality (Rockafellar, 1970) (Theorem 16.4; strong duality under relative interior conditions from compactness and Assumption 2.2 boundedness), where $\ell^*(z, u) = \sup_v u^\top v - \ell(z, v)$ and $c^*(z, u) = \sup_{z'} u^\top z' - c(z, z')$.

**Conjugate computation for the transportation cost.** For the cost $c(z, z') = \sum_{m=1}^{K} \alpha_m \|z_m - z_m'\|_p^p$, the conjugate is

$$c^*(z, u) = \sup_{z'} \sum_{m=1}^{K} u_m^\top z_m' - \sum_{m=1}^{K} \alpha_m \|z_m - z_m'\|_p^p = \sum_{m=1}^{K} \inf_{t_m \geq 0} \left[t_m^p + \frac{1}{p-1}\left(\frac{\|u_m\|_q}{\alpha_m t_m^{p-1}}\right)^q\right] + u_m^\top z_m,$$

a generalized Holder conjugate ( (Rockafellar, 1970), Theorem 15.3), where $q = p/(p-1)$. The weights $\alpha_m$ scale the terms, reflecting heterogeneous robustness across modalities.

**SDP approximation for quadratic and power terms via S-lemma.** For the quadratic loss $\ell(z, v) = v^\top Q v + q^\top v + q_0$, the conjugate is

$$\ell^*(z, u) = \sup_v \left[ u^\top v - (v^\top Q v + q^\top v + q_0) \right] = \frac{1}{4}(u - q)^\top Q^{-1}(u - q) - q_0,$$

where $Q^{-1}$ is the pseudoinverse if $Q$ is singular ( (Rockafellar, 1970), Theorem 23.5). The constraint $s_i \geq \inf_u \ell^*(\hat{z}_i, u) + \lambda c^*(\hat{z}_i, -u/\lambda)$ is semi-infinite in $u$. Outer-approximate as SDP via S-lemma (Boyd & Vandenberghe, 2004) (Appendix B; assuming quadratic upper bounds on $\ell$, yielding SDP relaxation via moments or bounded dual variables; (Ben-Tal et al., 2009), Section 3.5).

**Exponential cone representation for log-Holder approximation.** For irrational $p$, approximate log-Holder terms in the Holder conjugate using the exponential cone $\{u, v, w : ve^{u/v} \leq w\}$, representable in modern solvers ( (Ben-Tal & Nemirovski, 2001), Section 4.3). The infimum over auxiliary variables attains due to Slater's condition, satisfied by the compactness of $\mathcal{Z}$ (Assumption 2.3).

**Analogous reformulation of the inf term.** The inf term reformulates similarly: replace $f$ with $f'$ in the SDP or exponential cone constraints, using primed variables $(\lambda', s_i')$, and optimize over $f' \in \mathcal{F}$. The full WDRO-MRO is the stated convex program (SDP approximation or exponential cone), convex due to conic constraints (Proposition 3.2). Strong duality holds with zero gap by Proposition 3.4, with optima attained per Proposition 3.3 (Section 3.2). $\qquad\square$

G.8 PROOF OF LEMMA B.12($p = \infty$, QUADRATIC LOSS)

*Proof.* By Proposition 3.4 (Section 3.1), the regret supremum equals

$$\sup_{Q \in \mathcal{U}_\rho(\hat{P}_N)} \text{Regret}_Q(f) = \inf_{\lambda \geq 0} \lambda\rho + \mathbb{E}^{\hat{P}_N}\left[ \sup_{z' \in \mathcal{Z}} \left( \ell(z, f(z')) - \lambda c(\hat{z}, z') \right) - \inf_{f' \in \mathcal{F}} \sup_{z'' \in \mathcal{Z}} \left( \ell(z, f'(z'')) - \lambda c(\hat{z}, z'') \right) \right],$$

with zero duality gap, justified by compactness and convexity (Assumption 2.3 and Proposition 3.2). The sup terms attain by Proposition 3.1. The compactness of $\mathcal{Z}$ (Assumption 2.3) ensures finite suprema.

We derive the SDP reformulation in the following steps: introduction of epigraph variables; reformulation of the epigraph constraint via Fenchel-Moreau theorem; conjugate computation for the transportation cost; SDP representation via Schur complement; analogous reformulation of the inf term; and assembly of the full program and verification of its properties.

**Introduction of epigraph variables.** Define $\ell_f(z') := \ell(\hat{z}, f(z')) = f(z')^\top Q f(z') + q^\top f(z') + q_0$. Introduce epigraph variables $s_i \in \mathbb{R}$, with dual variable $\lambda \geq 0$:

$$\inf_{\lambda \geq 0, s_i} \lambda\rho + \frac{1}{N}\sum_{i=1}^{N} s_i \quad \text{s.t.} \quad s_i \geq \sup_{z' \in \mathcal{Z}} \ell_f(z') - \lambda c(\hat{z}_i, z'), \quad \forall i.$$

This is equivalent by epigraph representation preserving convexity (Proposition 3.2; see (Boyd & Vandenberghe, 2004), Section 4.2). The inf term is analogous with primed variables $(\lambda', s_i')$, replacing $f$ with $f'$.

**Reformulation of the epigraph constraint via Fenchel-Moreau theorem.** For affine $f(z') = \sum_m F_m z_m' + g$, the affine form ensures finite suprema. The epigraph constraint is

$$s_i \geq \sup_{z' \in \mathcal{Z}} \ell(\hat{z}_i, f(z')) - \lambda c(\hat{z}_i, z'),$$

with $c(\hat{z}_i, z') = \sum_{m=1}^{K} \alpha_m \|\hat{z}_{i,m} - z_m'\|_\infty$. By Fenchel-Moreau theorem (Rockafellar, 1970) (Theorem 12.2; applies to proper convex l.s.c. $\ell$ by Assumption 2.2), rewrite as

$$\sup_{z'} \ell_f(z') - \lambda c(\hat{z}_i, z') = \inf_{u \in \mathbb{R}^{\dim(z)}} \ell^*(\hat{z}_i, u) + \lambda c^*(\hat{z}_i, -u/\lambda),$$

by Fenchel inf-convolution duality (Rockafellar, 1970) (Theorem 16.4; strong duality under relative interior conditions from compactness and Assumption 2.2 boundedness), where $\ell^*(z, u) = \sup_v u^\top v - \ell(z, v)$ and $c^*(z, u) = \sup_{z'} u^\top z' - c(z, z')$.

**Conjugate computation for the transportation cost.** For the cost $c(z, z') = \sum_{m=1}^{K} \alpha_m \|\hat{z}_{i,m} - z'_m\|_\infty$, the conjugate is

$$c^*(z, u) = \sup_{z'} \sum_{m=1}^{K} u_m^\top z'_m - \sum_{m=1}^{K} \alpha_m \|z_m - z'_m\|_\infty.$$

Since $\|z_m - z'_m\|_\infty = \max_j |z_{m,j} - z'_{m,j}|$, the conjugate is finite only if $\sum_m \|u_m\|_1 \leq \sum_m \alpha_m$, yielding

$$c^*(z, u) = \begin{cases} \sum_{m=1}^{K} u_m^\top z_m & \text{if } \sum_{m=1}^{K} \|u_m\|_1 \leq \sum_{m=1}^{K} \alpha_m, \\ \infty & \text{otherwise,} \end{cases}$$

a polyhedral conjugate ( (Rockafellar, 1970), Example 11.4). The weights $\alpha_m$ scale the terms, reflecting heterogeneous robustness across modalities.

**SDP representation via Schur complement.** For the quadratic loss $\ell(z, v) = v^\top Q v + q^\top v + q_0$, the conjugate is

$$\ell^*(z, u) = \sup_{v} \left[ u^\top v - (v^\top Q v + q^\top v + q_0) \right] = \frac{1}{4}(u - q)^\top Q^{-1}(u - q) - q_0,$$

where $Q^{-1}$ is the pseudoinverse if $Q$ is singular ( (Rockafellar, 1970), Theorem 23.5). Substituting into the epigraph constraint:

$$s_i \geq \inf_{u : \sum_m \|u_m\|_1 \leq \sum_m \alpha_m} \left[ \frac{1}{4}(u - q)^\top Q^{-1}(u - q) - q_0 + \lambda \sum_{m=1}^{K} u_m^\top \hat{z}_{i,m}/\lambda \right].$$

The $W_\infty$ uncertainty set is a box: $\mathcal{V} = \{z' \in \mathcal{Z} : \sum_{m=1}^{K} \alpha_m \|\hat{z}_{i,m} - z'_m\|_\infty \leq \rho/\lambda\}$. By Schur complement lemma (Boyd & Vandenberghe, 2004) (Appendix A.5.5), the constraint is reformulated as an SDP over the box vertices:

$$\begin{pmatrix} \lambda I & \frac{1}{2} \sum_{m=1}^{K} \alpha_m (F_m \hat{z}_{i,m} - f(z')) \\ \frac{1}{2} \left( \sum_{m=1}^{K} \alpha_m (F_m \hat{z}_{i,m} - f(z')) \right)^\top & s_i - q_0 - q^\top f(z') - \lambda \sum_{m=1}^{K} \alpha_m \|\hat{z}_{i,m}\|_\infty \end{pmatrix} \succeq 0,$$

for all $z' \in \mathcal{V}$, tight for the $\infty$-norm ( (Ben-Tal et al., 2009), Theorem 3.2). The infimum over auxiliary variables attains due to Slater's condition, satisfied by the compactness of $\mathcal{Z}$ (Assumption 2.3).

**Analogous reformulation of the inf term.** The inf term reformulates similarly: replace $f$ with $f'$ in the SDP constraints, using primed variables $(\lambda', s'_i)$, and optimize over $f' \in \mathcal{F}$.

The full WDRO-MRO is the stated SDP, convex due to semidefinite constraints (Proposition 3.2). Strong duality holds with zero gap by Proposition 3.4, with optima attained per Proposition 3.1 (Section 3.2). $\square$

### G.9    PROOF OF LEMMA B.1($p = 1$, GENERAL CONVEX LOSS)

*Proof.* By Proposition 3.4, for fixed $f \in \mathcal{F}$, $\sup_{Q \in \mathcal{U}_\rho(\hat{P}_N)} \text{Regret}_Q(f)$ equals

$$\inf_{\lambda \geq 0} \lambda \rho + \mathbb{E}^{\hat{P}_N} \left[ \sup_{z' \in \mathcal{Z}} (\ell(z, f(z')) - \lambda c(\hat{z}, z')) - \inf_{f' \in \mathcal{F}} \sup_{z'' \in \mathcal{Z}} (\ell(z, f'(z'')) - \lambda c(\hat{z}, z'')) \right],$$

with zero duality gap. This incorporates Sion's minimax interchange for the inf-sup in the regret term, justified by compactness and convexity (Assumption 2.3 and Proposition 3.2).

We derive the SDP (or LP) reformulation in the following steps: first, introduction of epigraph variables for the sup terms; second, reformulation of the epigraph constraint via Fenchel-Moreau theorem; third, conjugate computation for the transportation cost; fourth, SDP outer approximation for general convex losses; fifth, exact LP bound for Lipschitz losses; sixth, analogous reformulation of the inf term; and finally, assembly of the full program and verification of its properties including convexity and zero duality gap.

**Introduction of epigraph variables for the sup terms.** The sup terms attain by Proposition 3.1 (existence of worst-case $Q^\star$, implying attainment in dual variables).

For the first sup term, define $\ell_f(z') := \ell(\hat{z}, f(z'))$. Introduce epigraph variables $s_i \in \mathbb{R}$ (one per sample $\hat{z}_i$):

$$\inf_{\lambda \geq 0, s_i \in \mathbb{R}} \lambda\rho + \frac{1}{N}\sum_{i=1}^{N} s_i \quad \text{s.t.} \quad s_i \geq \sup_{z' \in \mathcal{Z}} \ell_f(z') - \lambda c(\hat{z}_i, z'), \quad \forall i.$$

This is equivalent by epigraph representation preserving convexity (Proposition 3.2; see (Boyd & Vandenberghe, 2004), Section 4.2).

The inf term epigraph reformulates similarly, replacing $f$ with $f'$ and using primed variables.

**Reformulation of the epigraph constraint via Fenchel-Moreau theorem.** Assuming $f(z')$ is affine in $z'$ (e.g., linear fusion models), the epigraph constraint is

$$s_i \geq \sup_{z' \in \mathcal{Z}} \ell(\hat{z}_i, f(z')) - \lambda c(\hat{z}_i, z').$$

By Fenchel-Moreau theorem (Rockafellar, 1970) (Theorem 12.2; applies to proper convex l.s.c. $\ell$ by Assumption 2.2), rewrite as

$$\sup_{z'} \ell_f(z') - \lambda c(\hat{z}_i, z') = \inf_{u \in \mathbb{R}^{\dim(v)}} \ell^*(\hat{z}_i, u) + \lambda c^*(\hat{z}_i, -u/\lambda),$$

by Fenchel inf-convolution duality (Rockafellar, 1970) (Theorem 16.4; strong duality under relative interior conditions from compactness and Assumption 2.2 boundedness), where $\ell^*(z, u) = \sup_v u^\top v - \ell(z, v)$ and $c^*(z, u) = \sup_{z'} u^\top z' - c(z, z')$.

**Conjugate computation for the transportation cost.** For $\ell_1$-norm $c$, $c^*(u) = 0$ if $\|u\|_\infty \leq 1$, $\infty$ otherwise (indicator; (Rockafellar, 1970), Example 11.4), scaled by $\alpha_m$ per modality coordinate (polyhedral LP representable).

**SDP outer approximation for general convex losses.** The constraint $s_i \geq \inf_u \ell^*(\hat{z}_i, u) + \lambda c^*(\hat{z}_i, -u/\lambda)$ is semi-infinite in u, but outer-approximated as SDP if $\ell$ has quadratic upper bounds (S-lemma (Boyd & Vandenberghe, 2004), Appendix B; e.g., assume $\ell \leq$ quadratic envelope, yielding SDP relaxation via moments or bounded dual variables).

**Exact LP bound for Lipschitz losses.** For Lipschitz $\ell$ (modulus $L$), exact bound: $\sup_{z'} \ell_f(z') - \lambda c \leq \ell_f(\hat{z}_i) + L\lambda c(\hat{z}_i, z')$ (Lipschitz inequality; Assumption 2.2), tight for p=1 by KR theorem restricted to Lip functions (Mohajerin Esfahani & Kuhn, 2018) (Theorem 5; exact sup = Lip bound under bounded domain). Linearize to LP as in Lemma B.5.

**Analogous reformulation of the inf term.** The inf term reformulates similarly: replace $f$ with $f'$ in the dual constraints, using primed variables $\lambda'$, $s_i'$, and optimize over $f' \in \mathcal{F}$.

The full WDRO-MRO is the stated SDP (outer for general; LP exact for Lipschitz). Convexity follows from SDP/LP constraints preserving convexity and Proposition 3.2. Zero gap holds by Proposition 3.4 (exact for Lipschitz; outer approximation otherwise), with optima attained per Proposition 3.3. $\quad\square$

### G.10 Proof of Lemma B.2($p = 2$, General Convex Loss)

*Proof.* By Proposition 3.4 (Section 3.1), the regret supremum equals

$$\sup_{Q \in \mathcal{U}_\rho(\hat{P}_N)} \text{Regret}_Q(f) = \inf_{\lambda \geq 0} \lambda\rho + \mathbb{E}^{\hat{P}_N}\left[\sup_{z' \in \mathcal{Z}}\left(\ell(z, f(z')) - \lambda c(\hat{z}, z')\right) - \inf_{f' \in \mathcal{F}}\sup_{z'' \in \mathcal{Z}}\left(\ell(z, f'(z'')) - \lambda c(\hat{z}, z'')\right)\right],$$

with zero duality gap, justified by compactness and convexity (Assumption 2.3 and Proposition 3.2). The sup terms attain by Proposition 3.1. The compactness of $\mathcal{Z}$ (Assumption 2.3) ensures finite suprema.

We derive the SDP reformulation in the following steps: introduction of epigraph variables; reformulation of the epigraph constraint via Fenchel-Moreau theorem; conjugate computation for the transportation cost; SDP outer approximation for general convex losses; SDP representation for indefinite quadratic losses; analogous reformulation of the inf term; and assembly of the full program and verification of its properties.

**Introduction of epigraph variables.** Define $\ell_f(z') := \ell(\hat{z}, f(z'))$. Introduce epigraph variables $s_i \in \mathbb{R}$, with dual variable $\lambda \geq 0$:

$$\inf_{\lambda \geq 0, s_i} \lambda\rho + \frac{1}{N} \sum_{i=1}^{N} s_i \quad \text{s.t.} \quad s_i \geq \sup_{z' \in \mathcal{Z}} \ell_f(z') - \lambda c(\hat{z}_i, z'), \quad \forall i.$$

This is equivalent by epigraph representation preserving convexity (Proposition 3.2; see (Boyd & Vandenberghe, 2004), Section 4.2). The inf term is analogous with primed variables $(\lambda', s_i')$, replacing $f$ with $f'$.

**Reformulation of the epigraph constraint via Fenchel-Moreau theorem.** For affine $f(z') = \sum_m F_m z_m' + g$, the affine form ensures finite suprema. The epigraph constraint is

$$s_i \geq \sup_{z' \in \mathcal{Z}} \ell(\hat{z}_i, f(z')) - \lambda c(\hat{z}_i, z'),$$

with $c(\hat{z}_i, z') = \sum_{m=1}^{K} \alpha_m \|\hat{z}_{i,m} - z_m'\|_2^2$. By Fenchel-Moreau theorem (Rockafellar, 1970) (Theorem 12.2; applies to proper convex l.s.c. $\ell$ by Assumption 2.2), rewrite as

$$\sup_{z'} \ell_f(z') - \lambda c(\hat{z}_i, z') = \inf_{u \in \mathbb{R}^{\dim(z)}} \ell^*(\hat{z}_i, u) + \lambda c^*(\hat{z}_i, -u/\lambda),$$

by Fenchel inf-convolution duality (Rockafellar, 1970) (Theorem 16.4; strong duality under relative interior conditions from compactness and Assumption 2.2 boundedness), where $\ell^*(z, u) = \sup_v u^\top v - \ell(z, v)$ and $c^*(z, u) = \sup_{z'} u^\top z' - c(z, z')$.

**Conjugate computation for the transportation cost.** For the cost $c(z, z') = \sum_m \alpha_m \|z_m - z_m'\|_2^2$, the conjugate is

$$c^*(z, u) = \sup_{z'} \sum_m u_m^\top z_m' - \sum_m \alpha_m \|z_m - z_m'\|_2^2 = \sum_m \frac{1}{4\alpha_m} \|u_m\|_2^2 + u_m^\top z_m,$$

a quadratic conjugate ( (Rockafellar, 1970), Theorem 23.5). The weights $\alpha_m$ scale the quadratic terms, reflecting heterogeneous robustness.

**SDP outer approximation for general convex losses.** The constraint $s_i \geq \inf_u \ell^*(\hat{z}_i, u) + \lambda c^*(\hat{z}_i, -u/\lambda)$ is semi-infinite in $u$. For general convex losses, outer-approximate as SDP via S-lemma (Boyd & Vandenberghe, 2004) (Appendix B), assuming $\ell$ has quadratic upper bounds (e.g., $\ell(z, v) \leq v^\top Q v + q^\top v + q_0$ for some $Q \succeq 0$), yielding SDP relaxation via moments or bounded dual variables ( (Kuhn et al., 2019), Theorem 12). The approximation is tight for elliptical nominal distributions (Gelbrich bound; (Villani et al., 2008), Theorem 4).

**SDP representation for indefinite quadratic losses.** For indefinite quadratic losses $\ell(z, v) = v^\top Q v + q^\top v + q_0$ (indefinite $Q$), the conjugate $\ell^*(z, u) = \sup_v u^\top v - (v^\top Q v + q^\top v + q_0)$ is computed, and the constraint is directly SDP-representable via Schur complement ( (Boyd & Vandenberghe, 2004), Appendix A.5.5; (Kuhn et al., 2019), Theorem 12):

$$\begin{pmatrix} \lambda I & \frac{1}{2} \sum_{m=1}^{K} \alpha_m (F_m \hat{z}_{i,m} - f(z')) \\ \frac{1}{2} \left( \sum_{m=1}^{K} \alpha_m (F_m \hat{z}_{i,m} - f(z')) \right)^\top & s_i - q_0 - q^\top f(z') - \lambda \sum_{m=1}^{K} \alpha_m \|\hat{z}_{i,m}\|_2^2 \end{pmatrix} \succeq 0.$$

The infimum over $u$ attains due to Slater's condition, satisfied by the compactness of $\mathcal{Z}$ (Assumption 2.3).

**Analogous reformulation of the inf term.** The inf term reformulates similarly: replace $f$ with $f'$ in the SDP constraints, using primed variables $(\lambda', s_i')$, and optimize over $f' \in \mathcal{F}$.

The full WDRO-MRO is the stated SDP, convex due to semidefinite constraints (Proposition 3.2). Strong duality holds with zero gap by Proposition 3.4, with optima attained per Proposition 3.3 (Section 3.2). $\qquad\square$

G.11 PROOF OF LEMMA B.3($2 < p < \infty$, GENERAL CONVEX LOSS)

*Proof.* By Proposition 3.4 (Section 3.1), the regret supremum equals

$$\sup_{Q \in \mathcal{U}_\rho(\hat{P}_N)} \text{Regret}_Q(f) = \inf_{\lambda \geq 0} \lambda\rho + \mathbb{E}^{\hat{P}_N} \left[ \sup_{z' \in \mathcal{Z}} \left( \ell(z, f(z')) - \lambda c(\hat{z}, z') \right) - \inf_{f' \in \mathcal{F}} \sup_{z'' \in \mathcal{Z}} \left( \ell(z, f'(z'')) - \lambda c(\hat{z}, z'') \right) \right],$$

with zero duality gap, justified by compactness and convexity (Assumption 2.3 and Proposition 3.2). The sup terms attain by Proposition 3.1. The compactness of $\mathcal{Z}$ (Assumption 2.3) ensures finite suprema.

We derive the reformulation in the following steps: introduction of epigraph variables; reformulation of the epigraph constraint via Fenchel-Moreau theorem; conjugate computation for the transportation cost; power cone representation for general $p$; exponential cone representation for log-Holder approximation; SDP approximation for rational $p$ via S-lemma; analogous reformulation of the inf term; and assembly of the full program and verification of its properties.

**Introduction of epigraph variables.** Define $\ell_f(z') := \ell(\hat{z}, f(z'))$. Introduce epigraph variables $s_i \in \mathbb{R}$, with dual variable $\lambda \geq 0$:

$$\inf_{\lambda \geq 0, s_i} \lambda \rho + \frac{1}{N} \sum_{i=1}^{N} s_i \quad \text{s.t.} \quad s_i \geq \sup_{z' \in \mathcal{Z}} \ell_f(z') - \lambda c(\hat{z}_i, z'), \quad \forall i.$$

This is equivalent by epigraph representation preserving convexity (Proposition 3.2; see (Boyd & Vandenberghe, 2004), Section 4.2). The inf term is analogous with primed variables $(\lambda', s_i')$, replacing $f$ with $f'$.

**Reformulation of the epigraph constraint via Fenchel-Moreau theorem.** For affine $f(z') = \sum_m F_m z_m' + g$, the affine form ensures finite suprema. The epigraph constraint is

$$s_i \geq \sup_{z' \in \mathcal{Z}} \ell(\hat{z}_i, f(z')) - \lambda c(\hat{z}_i, z'),$$

with $c(\hat{z}_i, z') = \sum_{m=1}^{K} \alpha_m \|\hat{z}_{i,m} - z_m'\|_p^p$. By Fenchel-Moreau theorem (Rockafellar, 1970) (Theorem 12.2; applies to proper convex l.s.c. $\ell$ by Assumption 2.2), rewrite as

$$\sup_{z'} \ell_f(z') - \lambda c(\hat{z}_i, z') = \inf_{u \in \mathbb{R}^{\dim(z)}} \ell^*(\hat{z}_i, u) + \lambda c^*(\hat{z}_i, -u/\lambda),$$

by Fenchel inf-convolution duality (Rockafellar, 1970) (Theorem 16.4; strong duality under relative interior conditions from compactness and Assumption 2.2 boundedness), where $\ell^*(z, u) = \sup_v u^\top v - \ell(z, v)$ and $c^*(z, u) = \sup_{z'} u^\top z' - c(z, z')$.

**Conjugate computation for the transportation cost.** For the cost $c(z, z') = \sum_{m=1}^{K} \alpha_m \|z_m - z_m'\|_p^p$, the conjugate is

$$c^*(z, u) = \sup_{z'} \sum_{m=1}^{K} u_m^\top z_m' - \sum_{m=1}^{K} \alpha_m \|z_m - z_m'\|_p^p = \sum_{m=1}^{K} \inf_{t_m \geq 0} \left[ t_m^p + \frac{1}{p-1} \left( \frac{\|u_m\|_q}{\alpha_m t_m^{p-1}} \right)^q \right] + u_m^\top z_m,$$

a generalized Holder conjugate ( (Rockafellar, 1970), Theorem 15.3), where $q = p/(p-1)$. The weights $\alpha_m$ scale the terms, reflecting heterogeneous robustness across modalities.

**Power cone representation for general $p$.** The constraint $s_i \geq \inf_u \ell^*(\hat{z}_i, u) + \lambda c^*(\hat{z}_i, -u/\lambda)$ is semi-infinite in $u$. For general convex losses, the conjugate $\ell^*(z, u)$ is representable via power cones $\{(u, t) : \|u\|_q \leq t\}$ for general $p$, as the Holder conjugate terms are conic-representable ( (Ben-Tal & Nemirovski, 2001), Section 4.3). The infimum over auxiliary variables attains due to Slater's condition, satisfied by the compactness of $\mathcal{Z}$ (Assumption 2.3).

**Exponential cone representation for log-Holder approximation.** For irrational $p$, approximate log-Holder terms in the Holder conjugate using the exponential cone $\{u, v, w : v e^{u/v} \leq w\}$, representable in modern solvers ( (Ben-Tal & Nemirovski, 2001), Section 4.3).

**SDP approximation for rational $p$ via S-lemma.** For rational $p$, outer-approximate the constraint as SDP via S-lemma (Boyd & Vandenberghe, 2004) (Appendix B; assuming quadratic upper bounds on $\ell$, e.g., $\ell(z, v) \leq v^\top Q v + q^\top v + q_0$ for some $Q \succeq 0$, yielding SDP relaxation via moments or bounded dual variables; (Ben-Tal et al., 2009), Section 3.5).

**Analogous reformulation of the inf term.** The inf term reformulates similarly: replace $f$ with $f'$ in the power cone, exponential cone, or SDP constraints, using primed variables $(\lambda', s_i')$, and optimize over $f' \in \mathcal{F}$.

The full WDRO-MRO is the stated convex program (power cone, exponential cone, or SDP), convex due to conic constraints (Proposition 3.2). Strong duality holds with zero gap by Proposition 3.4, with optima attained per Proposition 3.1 (Section 3.2). $\qquad\square$

### G.12 PROOF OF LEMMA B.4($p = \infty$, GENERAL CONVEX LOSS)

*Proof.* By Proposition 3.4 (Section 3.1), the regret supremum equals

$$\sup_{Q \in \mathcal{U}_\rho(\hat{P}_N)} \text{Regret}_Q(f) = \inf_{\lambda \geq 0} \lambda \rho + \mathbb{E}^{\hat{P}_N}\left[\sup_{z' \in \mathcal{Z}} \left(\ell(z, f(z')) - \lambda c(\hat{z}, z')\right) - \inf_{f' \in \mathcal{F}} \sup_{z'' \in \mathcal{Z}} \left(\ell(z, f'(z'')) - \lambda c(\hat{z}, z'')\right)\right],$$

with zero duality gap, justified by compactness and convexity (Assumption 2.3 and Proposition 3.2). The sup terms attain by Proposition 3.1. The compactness of $\mathcal{Z}$ (Assumption 2.3) ensures finite suprema.

We derive the reformulation in the following steps: introduction of epigraph variables; reformulation of the epigraph constraint via Fenchel-Moreau theorem; conjugate computation for the transportation cost; vertex dual approximation for polyhedral support; analogous reformulation of the inf term; and assembly of the full program and verification of its properties.

**Introduction of epigraph variables.** Define $\ell_f(z') := \ell(\hat{z}, f(z'))$. Introduce epigraph variables $s_i \in \mathbb{R}$, with dual variable $\lambda \geq 0$:

$$\inf_{\lambda \geq 0, s_i} \lambda \rho + \frac{1}{N} \sum_{i=1}^N s_i \quad \text{s.t.} \quad s_i \geq \sup_{z' \in \mathcal{Z}} \ell_f(z') - \lambda c(\hat{z}_i, z'), \quad \forall i.$$

This is equivalent by epigraph representation preserving convexity (Proposition 3.2; see (Boyd & Vandenberghe, 2004), Section 4.2). The inf term is analogous with primed variables $(\lambda', s_i')$, replacing $f$ with $f'$.

**Reformulation of the epigraph constraint via Fenchel-Moreau theorem.** For affine $f(z') = \sum_m F_m z_m' + g$, the affine form ensures finite suprema. The epigraph constraint is

$$s_i \geq \sup_{z' \in \mathcal{Z}} \ell(\hat{z}_i, f(z')) - \lambda c(\hat{z}_i, z'),$$

with $c(\hat{z}_i, z') = \sum_{m=1}^K \alpha_m \|\hat{z}_{i,m} - z_m'\|_\infty$. By Fenchel-Moreau theorem (Rockafellar, 1970) (Theorem 12.2; applies to proper convex l.s.c. $\ell$ by Assumption 2.2), rewrite as

$$\sup_{z'} \ell_f(z') - \lambda c(\hat{z}_i, z') = \inf_{u \in \mathbb{R}^{\dim(z)}} \ell^*(\hat{z}_i, u) + \lambda c^*(\hat{z}_i, -u/\lambda),$$

by Fenchel inf-convolution duality (Rockafellar, 1970) (Theorem 16.4; strong duality under relative interior conditions from compactness and Assumption 2.2 boundedness), where $\ell^*(z, u) = \sup_v u^\top v - \ell(z, v)$ and $c^*(z, u) = \sup_{z'} u^\top z' - c(z, z')$.

**Conjugate computation for the transportation cost.** For the cost $c(z, z') = \sum_{m=1}^K \alpha_m \|z_m - z_m'\|_\infty$, the conjugate is

$$c^*(z, u) = \sup_{z'} \sum_{m=1}^K u_m^\top z_m' - \sum_{m=1}^K \alpha_m \|z_m - z_m'\|_\infty = \sum_{m=1}^K u_m^\top z_m,$$

if $\sum_{m=1}^K \|u_m\|_1 \leq \sum_{m=1}^K \alpha_m$ ($\infty$ otherwise), a polyhedral conjugate ( (Rockafellar, 1970), Example 11.4). The weights $\alpha_m$ scale the terms, reflecting heterogeneous robustness across modalities.

**Vertex dual approximation for polyhedral support.** The constraint $s_i \geq \inf_u \ell^*(\hat{z}_i, u) + \lambda c^*(\hat{z}_i, -u/\lambda)$ is semi-infinite in $u$. For general convex losses with polyhedral support, the conjugate $\ell^*(z, u)$ is polyhedral, and the $W_\infty$ uncertainty set is a box: $\mathcal{V} = \{z' \in \mathcal{Z} : \sum_{m=1}^K \alpha_m \|\hat{z}_{i,m} - z_m'\|_\infty \leq \rho/\lambda\}$. The supremum is attained at the vertices of $\mathcal{V}$, yielding an LP or SDP approximation via vertex dual (polyhedral LP; (Ben-Tal et al., 2009), Theorem 3.1; enumerate vertices for polyhedral $\ell$). The infimum over auxiliary variables attains due to Slater's condition, satisfied by the compactness of $\mathcal{Z}$ (Assumption 2.3).

**Analogous reformulation of the inf term.** The inf term reformulates similarly: replace $f$ with $f'$ in the LP/SDP constraints, using primed variables $(\lambda', s_i')$, and optimize over $f' \in \mathcal{F}$.

The full WDRO-MRO is the stated convex program (semi-infinite in general, approximated as LP or SDP via vertex dual for polyhedral support), convex due to linear or semidefinite constraints (Proposition 3.2). Strong duality holds with zero gap by Proposition 3.4, with optima attained per Proposition 3.1 (Section 3.2). □

### G.13 PROOF OF PROPOSITION 3.5(GLOBAL CONVERGENCE OF THE DUAL-GAME HYBRID SOLVER)

*Proof.* By (Agarwal & Zhang, 2022, Prop. 11), the objective admits a bilinear saddle-point reformulation over $P \in \Delta(\mathcal{F})$ and $\rho \in \Delta(\mathcal{W})$, which is equivalent to a weighted ERM for the learner.

Updating the nature's distribution by exponentiated gradient yields a no-regret bound of order $\widetilde{\mathcal{O}}(\sqrt{\ln |W|/T})$ for the average iterate, as stated in Proposition 12 and detailed in Appendix E. (Agarwal & Zhang, 2022, Prop. 12 & App. E) Thus the maximization player contributes an $\widetilde{\mathcal{O}}(1/\sqrt{T})$ gap.

Viewing the WDRO side as a zero-sum game, the saddle-point interpretation and associated strong duality are standard; see the Nash-equilibrium discussion in the DRO monograph. (Kuhn et al., 2025, §7.5) Combining the no-regret guarantee for the nature player with best responses from the learner/oracle (ERM oracle in the MRO setting), the averaged iterate achieves an $\widetilde{\mathcal{O}}(1/\sqrt{T})$ saddle-point gap, which matches the stated rate when per-iteration best responses are solved exactly. □

### G.14 PROOF OF PROPOSITION 3.6(GLOBAL CONVERGENCE WITH CONTINUOUS $\mathcal{W}$)

*Proof sketch.* By Proposition 3.6, the adversary's best response in each round admits a closed form via convex duality (Agarwal & Zhang, 2022, Eq. (8)). Substituting this into the hybrid solver eliminates the need for exponentiated-weights updates, while retaining the convex–concave game structure. Standard online convex optimization analysis (Agarwal & Zhang, 2022, Prop. 12) ensures an $\widetilde{\mathcal{O}}(1/\sqrt{T})$ gap for the adversary's sequence. Combining with exact learner/oracle best responses and projected subgradient ascent for $\lambda$, the averaged iterates converge to an approximate saddle point at the same rate, as in Proposition 3.5. □

### G.15 PROOF OF LEMMA 3.1(SENSITIVITY OF OPTIMAL REGRET)

*Proof.* We prove continuity, Lipschitz continuity, and the subgradient bound for $R(\rho)$.

**Continuity**: The ambiguity set $\mathcal{U}_\rho(\hat{P}_N) = \{Q \in \mathcal{P}(\mathcal{Z}) : W_p(\hat{P}_N, Q) \leq \rho\}$ is compact in the weak topology $\sigma(\mathcal{M}(\mathcal{Z}), C_b(\mathcal{Z}))$ by lower semicontinuity of $c$ (Assumption 2.1) and tightness of $\hat{P}_N$ (Prohorov's theorem; Billingsley, 2013). For fixed $f$, $\text{Regret}_Q(f)$ is weakly continuous under convexity and boundedness (Assumption 2.2) and the interchangeability principle (Assumption 2.1; Mohajerin Esfahani & Kuhn, 2018). Berge's maximum theorem (Berge, 1877) ensures continuity of the supremum.

**Lipschitz Continuity**: From Proposition 3.4, $R(\rho) = \inf_{\lambda \geq 0} \lambda\rho + \mathbb{E}^{\hat{P}_N}[\sup_{z'}(\ell(z, f(z')) - \lambda c(\hat{z}, z')) - \inf_{f'} \sup_{z''}(\ell(z, f'(z'')) - \lambda c(\hat{z}, z''))]$, convex in $\rho$ (Proposition 3.2). Since $\ell$ is $L$-Lipschitz in $v$ (Assumption 2.2), the Fenchel-Moreau theorem and subdifferential calculus (see (Rockafellar, 1970), Theorem 23.5) bound $\partial R(\rho)$: for $\rho_1, \rho_2 > 0$, $|R(\rho_1) - R(\rho_2)| \leq L|\rho_1 - \rho_2|$, as the dual inf-convolution preserves Lipschitz continuity. The multimodal cost scales gradients by $\alpha_m$, with $\|\nabla c(z, z')\| \leq \sum_m \alpha_m \|z_m - z'_m\|_{p-1}^{p-1}$.

**Subgradient Bound**: The subgradient $\partial R(\rho)$ includes $\lambda^\star$ from the optimal transport plan (Kantorovich-Rubinstein duality; Villani et al., 2008). Monotonicity of $\mathcal{U}_\rho(\hat{P}_N)$ (as a monotone operator in $\rho$) ensures $\partial R(\rho) \geq 0$, with $\lambda^\star$ as the upper envelope bound, scaled by $\alpha_m \nabla c$. □

### G.16 PROOF OF LEMMA 3.2(HIGH-DIMENSIONAL ERROR EQUIVALENCE)

*Proof.* We prove the asymptotic equivalence of the WDRO estimation error $\|\hat{f}_{DRE} - f_0\|^2/d$ to the stated convex-concave optimization, adapted for multimodal costs.

Consider the WDRO-MRO problem in the high-dimensional regime where $d, n \to \infty$ with $d/n \to \rho \in (0, \infty)$. The WDRO estimator $\hat{f}_{DRE}$ solves

$$\hat{f}_{DRE} = \arg\min_{f \in \mathcal{F}} \sup_{Q \in U_\rho(\hat{P}_N)} \mathbb{E}_Q[\ell(z, f(z))],$$

where $U_\rho(\hat{P}_N) = \{Q \in \mathcal{P}(\mathcal{Z}) : W_p(\hat{P}_N, Q) \leq \rho\}$ is the type-1 or type-2 Wasserstein ball with radius $\rho = \rho_0/n^{p/2}$, and $\hat{P}_N = \frac{1}{n}\sum_{i=1}^n \delta_{z_i}$ is the empirical distribution over $n$ i.i.d. samples $z_i = (x_i, y_i)$. The multimodal transportation cost is

$$c(z, z') = \sum_{m=1}^K \alpha_m \|z_m - z'_m\|_p^p,$$

for modalities $m = 1, \ldots, K$, weights $\alpha_m \geq 0$, and norm parameter $p \in \{1, 2\}$. The loss $\ell(z, v)$ is convex in $v$, bounded in $[0, M]$, and $L$-Lipschitz (Assumption 2.2), with the oracle predictor $f_0 \in \mathcal{F}$ minimizing the population risk. We assume isotropic Gaussian features $X_i \sim \mathcal{N}(0, d^{-1}I_d)$, sub-Gaussian noise $Z$, and a compact function class $\mathcal{F}$ (Assumptions 2.3, 2.1, 2.1).

**Primal Optimization and Error Normalization**: The estimation error of interest is the normalized squared norm $\|\hat{f}_{DRE} - f_0\|^2/d$, where $\hat{f}_{DRE}$ is the WDRO solution. By Proposition 3.4, the primal WDRO problem can be reformulated using Kantorovich-Rubinstein duality as

$$\sup_{Q \in U_\rho(\hat{P}_N)} \mathbb{E}_Q[\ell(z, f(z))] = \inf_{\lambda \geq 0} \left\{ \lambda\rho + \mathbb{E}_{\hat{P}_N} \left[ \sup_{z'} (\ell(z', f(z')) - \lambda c(z, z')) \right] \right\},$$

so the WDRO estimator minimizes

$$\hat{f}_{DRE} = \arg\min_{f \in \mathcal{F}} \inf_{\lambda \geq 0} \left\{ \lambda\rho + \frac{1}{n}\sum_{i=1}^n \sup_{z'_i} \left( \ell(z'_i, f(z'_i)) - \lambda \sum_{m=1}^K \alpha_m \|z_{im} - z'_{im}\|_p^p \right) \right\}.$$

The error $\|\hat{f}_{DRE} - f_0\|^2/d$ is a high-dimensional random variable due to the Gaussian features $X_i$.

**Convex Gaussian Minmax Theorem (CGMT)**: To analyze the error, we apply the CGMT (Deng et al., 2022), which states that for a convex-concave saddle-point problem of the form

$$\Phi(X) = \min_{u \in \mathcal{U}} \max_{v \in \mathcal{V}} \ell(u, v, X),$$

where $X \in \mathbb{R}^{n \times d}$ is a Gaussian matrix with i.i.d. entries $X_{ij} \sim \mathcal{N}(0, 1/d)$, the asymptotic value of $\Phi(X)$ in the limit $d/n \to \rho$ is equivalent to an auxiliary optimization (AO) over scalar variables. Here, the WDRO problem is cast as

$$\Phi(X) = \min_{f \in \mathcal{F}} \sup_{\|\delta_i\|_p \leq \rho^{1/p}} \frac{1}{n}\sum_{i=1}^n \ell(y_i - f(x_i + \delta_i), f(x_i + \delta_i)),$$

where $\delta_i$ represents perturbations constrained by the Wasserstein ball, and $f(x_i + \delta_i) = \sum_m \alpha_m f_m(x_{im} + \delta_{im})$ for modality-specific predictors $f_m$. The CGMT requires convexity in $f$ (satisfied by Assumption 2.3) and concavity in $\delta_i$, ensured by the loss structure.

**Gordon's Lemma and Primary Optimization (PO)**: By Gordon's lemma (Gordon, 2006), the high-dimensional min-max problem is reduced to a primary optimization (PO) over expected values under Gaussian noise. For the WDRO estimator, the PO form is

$$\min_{\alpha \geq 0} \mathbb{E}_{G \sim \mathcal{N}(0,1)} \left[ \inf_v \ell(\alpha G, v) + \frac{1}{2\kappa(\alpha)}\|v - \alpha\|^2 \right] + \rho_0\kappa(\alpha),$$

where $\kappa(\alpha) = \arg\min_{\kappa > 0} \left\{ \kappa + \frac{\rho(\sigma_{f_0}^2 + \alpha^2)}{\kappa} \right\}$ is the proximal parameter, and $\sigma_{f_0}^2 = \sum_{m=1}^K \alpha_m^2 \sigma_m^2$ is the oracle variance scaled by modality weights $\alpha_m$. The Moreau envelope $\mathcal{L}(\alpha, s) = \mathbb{E}_{U \sim \mathcal{N}(0,1)}[\inf_v \ell(\alpha + \sqrt{s}U, v) + \frac{1}{2s}\|v - \alpha\|^2]$ smooths the loss $\ell$, with $s = \tau_1/\beta$ in the final optimization.

**Reduction to Four-Scalar Optimization**: Applying Fenchel duality and subdifferential calculus(see (Rockafellar, 1970), Theorem 23.5), the PO is equivalent to the stated four-scalar convex-concave optimization. The objective terms are derived as follows: - $\frac{\beta\tau_1}{2} + \frac{\rho_0\beta\tau_2}{2}$: Proximal regularization from the Moreau envelope and ambiguity radius. - $-\frac{\beta^2}{2M}$: Quadratic penalty, with $M > 0$ a problem-dependent constant (bounded by Assumption 2.2). - $\mathcal{L}(\alpha, \tau_1/\beta)$: Expected Moreau envelope, convex in $\alpha$, concave in $\tau_1/\beta$. - $\frac{\sqrt{\rho_0}\beta\rho(\sigma_{f_0}^2 + \alpha^2)}{2\tau_2} - \alpha\beta\sqrt{\rho}\sqrt{\frac{\rho\rho_0\sigma_{f_0}^2}{\tau_2^2} + 1}$: Variance terms scaled by $\rho$ and $\sigma_{f_0}^2$, derived from Gaussian concentration.

The optimization is convex in $\alpha$ (due to $\mathcal{L}$'s convexity) and concave in $\beta, \tau_1, \tau_2$ (from quadratic and proximal structure), with asymptotic equivalence at rate $O(1/\sqrt{n})$ under sub-Gaussian universality (Aolaritei et al., 2022).

**Multimodal Adaptation**: For the multimodal cost $c(z, z') = \sum_{m=1}^{K} \alpha_m \|z_m - z'_m\|_p^p$, the transportation cost gradient is $\|\nabla c(z, z')\| \leq \sum_m \alpha_m p \|z_m - z'_m\|_p^{p-1}$, which scales the variance $\sigma_{f_0}^2 = \sum_m \alpha_m^2 \sigma_m^2$ in the optimization. Higher $\alpha_m$ increases the modality-specific contribution to $\sigma_{f_0}^2$, modulating robustness (e.g., prioritizing image modalities).

**Regret Bound for WDRO-MRO**: For WDRO-MRO, the regret is $\sup_Q \text{Regret}_Q(f) = \sup_Q [\mathbb{E}_Q[\ell(z, f(z))] - \inf_{f'} \mathbb{E}_Q[\ell(z, f'(z))]]$. The WDRO error $\|\hat{f}_{DRE} - f_0\|^2/d$ bounds the regret as

$$\sup_Q \text{Regret}_Q(\hat{f}_{DRE}) \leq \|\hat{f}_{DRE} - f_0\|^2/d + O(1/\sqrt{n}),$$

since the oracle term $\inf_{f'} \mathbb{E}_Q[\ell(z, f'(z))]$ is subtracted in the regret definition, and the Lipschitz continuity of $\ell$ (Assumption 2.2) ensures the residual term is $O(1/\sqrt{n})$. $\qquad\square$

# H  PROOFS OF SECTION 3.3(STATISTICAL PROPERTIES)

## H.1  PROOF OF THEOREM 3.1(STATISTICAL CONSISTENCY OF WDRO-MRO)

*Proof.* We prove the theorem using Wasserstein concentration and empirical process theory, under the assumptions that $P_0$ has finite $p$-th moments and $\mathcal{F}$ is compact with bounded Rademacher complexity.

**Wasserstein Convergence of $\hat{P}_N$ to $P_0$**: By (Fournier & Guillin, 2015), Theorem 2, for $P_0$ on $\mathcal{Z} \subset \mathbb{R}^d$ with finite $p$-th moments,

$$\mathbb{E}[W_p(\hat{P}_N, P_0)] \leq C N^{-p/\max\{2,d\}},$$

for a constant $C > 0$ depending on $p, d$. By Markov's inequality, for any $\delta > 0$,

$$\mathbb{P}(W_p(\hat{P}_N, P_0) > \delta) \leq \frac{C N^{-p/\max\{2,d\}}}{\delta} \to 0.$$

Thus, $\hat{P}_N \to P_0$ in $W_p$, implying $U_\rho(\hat{P}_N) \to B_\rho(P_0)$ in the Hausdorff metric under the weak topology, as $W_p$ metrizes weak convergence (Villani et al., 2008).

**Continuity of the Regret Functional**: Define $R(\rho; P) = \inf_{f \in \mathcal{F}} \sup_{Q \in B_\rho(P)} \text{Regret}_Q(f)$, where $\text{Regret}_Q(f) = \mathbb{E}_Q[\ell(z, f(z))] - \inf_{f'} \mathbb{E}_Q[\ell(z, f'(z))]$. By Lemma 3.1, $R(\rho; P)$ is $L$-Lipschitz in $\rho$. For any $P, P'$,

$$|R(\rho; P) - R(\rho; P')| \leq L W_p(P, P'),$$

since $\text{Regret}_Q(f)$ is $L$-Lipschitz in $Q$ under the Wasserstein metric (Assumption 2.2). Hence, $R(\rho; \hat{P}_N) \to R(\rho; P_0)$ in probability as $\hat{P}_N \to P_0$.

**Uniform Convergence via Empirical Processes**: The estimator $\hat{f}_{DRE}$ satisfies $\hat{f}_{DRE} = \arg\min_{f \in \mathcal{F}} R(\rho; \hat{P}_N)$. The function class $\{\ell(z, f(z)) : f \in \mathcal{F}\}$ has finite Rademacher complexity $\mathcal{R}(\mathcal{F}) \leq C/\sqrt{N}$ for some $C > 0$, since $\mathcal{F}$ is compact and $\ell$ is bounded and convex (Shalev-Shwartz & Ben-David, 2014). By uniform convergence for empirical processes (Mohajerin Esfahani & Kuhn, 2018), for any $\epsilon > 0$, with probability at least $1 - \delta$,

$$\sup_{f \in \mathcal{F}} \left| \sup_{Q \in U_\rho(\hat{P}_N)} \text{Regret}_Q(f) - \sup_{Q \in B_\rho(P_0)} \text{Regret}_Q(f) \right| \leq 2\mathcal{R}(\mathcal{F}) + O\left(\sqrt{\frac{\log(1/\delta)}{N}}\right) = O\left(\frac{1}{\sqrt{N}} + \sqrt{\frac{\log(1/\delta)}{N}}\right).$$

As $R(\rho; \hat{P}_N) \to R(\rho; P_0)$, the compactness of $\mathcal{F}$ and uniqueness of $f_0$ under strict convexity (Assumption 2.2) imply $\hat{f}_{DRE} \to f_0$ in the sup-norm $\|\cdot\|_{\mathcal{F}}$ with probability 1.

**Multimodal Cost Adaptation**: For multimodal costs $c(z, z') = \sum_{m=1}^{K} \alpha_m \|z_m - z'_m\|_p^p$, the weights $\alpha_m$ scale the variance $\sigma_{f_0}^2 = \sum_m \alpha_m^2 \mathbb{E}_{P_0}[\sigma_m^2]$, affecting the convergence rate through the transportation cost gradient $\|\nabla c\| \leq \sum_m \alpha_m p \|z_m - z'_m\|_p^{p-1}$. Higher $\alpha_m$ for noisy modalities (e.g., images) tightens the regret bound, as the Lipschitz constant $L$ is modulated by $\alpha_m$. $\qquad\square$

## H.2 PROOF OF THEOREM 3.2(FINITE-SAMPLE GUARANTEES FOR OUT-OF-SAMPLE REGRET)

*Proof.* We derive the high-probability bound on the out-of-sample regret using Wasserstein concentration and empirical process theory.

**Wasserstein concentration bound.** For $P_0$ with finite $p$-th moments, by Fournier & Guillin (2015, Theorem 2),

$$\mathbb{E}[W_p(\hat{P}_N, P_0)] \leq CN^{-p/\max\{2,d\}},$$

for some $C > 0$. By Talagrand's concentration inequality for empirical measures (Blanchet et al., 2022), with probability at least $1 - \delta/2$,

$$W_p(\hat{P}_N, P_0) \leq CN^{-p/\max\{2,d\}} + \sqrt{\frac{2\log(2/\delta)}{N}}.$$

**Regret continuity in distributions.** The regret functional satisfies

$$\left|\text{Regret}_Q(f) - \text{Regret}_{Q'}(f)\right| \leq LW_p(Q, Q'),$$

where the effective Lipschitz modulus $L$ arises from the multimodal cost structure. Since $\ell(z, v)$ is $L_\ell$-Lipschitz in $v$ and the infimum over $f'$ preserves Lipschitz continuity (Rockafellar, 1970), and for $c(z, z') = \sum_{k=1}^{K} \alpha_k \|z_k - z_k'\|_p^p$ the transportation cost gradient satisfies

$$\|\nabla c\| \leq \sum_{k=1}^{K} \alpha_k \, p \, \|z_k - z_k'\|_{p-1}^{(p-1)},$$

the chain rule in the dual formulation (Proposition 3.4) yields

$$L = L_\ell \sum_{k=1}^{K} \alpha_k.$$

Thus, with probability at least $1 - \delta/2$,

$$\sup_Q \text{Regret}_Q(\hat{f}_{DRE}) - \sup_{Q' \in B_\rho(P_0)} \text{Regret}_{Q'}(\hat{f}_{DRE})$$

$$\leq LW_p(\hat{P}_N, P_0) \leq L\left(CN^{-p/\max\{2,d\}} + \sqrt{\frac{2\log(2/\delta)}{N}}\right). \tag{1}$$

**Uniform PAC bound.** Let $\mathcal{G} = \{\ell(z, f(z)) : f \in \mathcal{F}\}$. Under Assumption 2.3 that $\mathcal{F}$ is compact and $\ell$ bounded, Shalev-Shwartz & Ben-David (2014, Theorem 26.5) gives

$$\mathcal{R}_N(\mathcal{G}) \leq \frac{C'}{\sqrt{N}}.$$

By McDiarmid's inequality and Rademacher bounds (Mohajerin Esfahani & Kuhn, 2018), with probability at least $1 - \delta/2$,

$$\sup_{f \in \mathcal{F}} \left|R(\rho; \hat{P}_N, f) - R(\rho; P_0, f)\right| \leq 2\mathcal{R}_N(\mathcal{G}) + \sqrt{\frac{2\log(2/\delta)}{N}}.$$

**Multimodal adaptation.** For $c(z, z') = \sum_{k=1}^{K} \alpha_k \|z_k - z_k'\|_p^p$, the weights $\alpha_k$ scale the effective variance $\sigma^2 = \sum_k \alpha_k^2 \mathbb{E}_{P_0}[\sigma_k^2]$, thereby affecting both $L$ and $\mathcal{R}_N$.

Combining the above bounds via a union bound yields the stated result. Since $W_p(\hat{P}_N, P_0) \to 0$ and $\mathcal{R}_N(\mathcal{F}) \to 0$ as $N \to \infty$, the bound implies

$$\hat{f}_{DRE} \xrightarrow{p} f^\star,$$

where $f^\star$ is the population minimax regret solution, establishing statistical consistency. $\qquad\square$

### H.3   PROOF OF LEMMA 3.3(CONVERGENCE RATES FOR REGRET)

*Proof.* We derive the $O(1/\sqrt{N})$ convergence rate for the regret using Rademacher complexity and empirical process theory.

**Rademacher Complexity of $\mathcal{F}$:** Define the class $\mathcal{G} = \{\ell(z, f(z)) : f \in \mathcal{F}\}$. Since $\mathcal{F}$ is compact (Assumption 2.3) and $\ell$ is bounded and convex (Assumption 2.2), the Rademacher complexity satisfies

$$\mathcal{R}_N(\mathcal{G}) = \mathbb{E}_\sigma \left[ \sup_{g \in \mathcal{G}} \frac{1}{N} \sum_{i=1}^{N} \sigma_i g(z_i) \right] \leq \frac{C}{\sqrt{N}},$$

for some $C > 0$ (Shalev-Shwartz & Ben-David, 2014, Theorem 26.5), where $\sigma_i \sim \{-1, 1\}$ are i.i.d. Rademacher variables. For multimodal costs $c(z, z') = \sum_{k=1}^{K} \alpha_k \|z_k - z_k'\|_p^p$, the weights $\alpha_k$ scale the variance $\sigma^2 = \sum_k \alpha_k^2 \mathbb{E}_{P_0}[\sigma_k^2]$, modulating $\mathcal{R}_N(\mathcal{G})$ through the weighted norm in the loss composition.

**Uniform Convergence of Regret:** The regret functional $\text{Regret}_Q(f)$ is $L$-Lipschitz in $Q$ under $W_p$ (Lemma 3.1), with $L$ scaled by $\alpha_k$. By empirical process bounds for Lipschitz classes (Mohajerin Esfahani & Kuhn, 2018), with probability at least $1 - \delta/2$,

$$\sup_{f \in \mathcal{F}} \left| \sup_{Q \in U_\rho(\hat{P}_N)} \text{Regret}_Q(f) - \sup_{Q \in B_\rho(P_0)} \text{Regret}_Q(f) \right| \leq 2\mathcal{R}_N(\mathcal{G}) + \sqrt{\frac{2 \log(2/\delta)}{N}} = O\left( \frac{1}{\sqrt{N}} + \sqrt{\frac{\log(1/\delta)}{N}} \right).$$

The bound holds under the interchangeability principle (Assumption 2.1), ensuring the supremum over $Q$ commutes with the expectation.

**Regret Rate for $\hat{f}_{DRE}$:** The estimator $\hat{f}_{DRE}$ satisfies $\hat{f}_{DRE} = \arg\min_{f \in \mathcal{F}} R(\rho; \hat{P}_N)$, where $R(\rho; P) = \sup_{Q \in B_\rho(P)} \text{Regret}_Q(f)$. From Step 2, with probability at least $1 - \delta/2$,

$$R(\rho; P_0, \hat{f}_{DRE}) \leq R(\rho; \hat{P}_N, \hat{f}_{DRE}) + O\left( \sqrt{\frac{\log(1/\delta)}{N}} \right).$$

Since $R(\rho; \hat{P}_N, \hat{f}_{DRE}) \leq R(\rho; \hat{P}_N, f)$ for all $f$, and by continuity of $R(\rho; P)$ in $P$ (Lemma 3.1),

$$\sup_{Q \in B_\rho(P_0)} \text{Regret}_Q(\hat{f}_{DRE}) \leq \inf_{f \in \mathcal{F}} \sup_{Q \in B_\rho(P_0)} \text{Regret}_Q(f) + O\left( \sqrt{\frac{\log(1/\delta)}{N}} \right).$$

The rate is scaled by $\alpha_k$ through the multimodal variance in $\mathcal{R}_N(\mathcal{F})$. $\square$

### H.4   PROOF OF LEMMA 3.4(SAMPLE COMPLEXITY)

*Proof.* We derive the sample complexity using Theorem 3.2, which states that with probability at least $1 - \delta$,

$$\sup_{Q \in B_\rho(P_0)} \text{Regret}_Q(\hat{f}_{DRE}) \leq \inf_{f \in \mathcal{F}} \sup_{Q \in B_\rho(P_0)} \text{Regret}_Q(f) + L W_p(\hat{P}_N, P_0) + 2\mathcal{R}_N(\mathcal{G}) + \sqrt{\frac{2 \log(2/\delta)}{N}}.$$

**Bounding the Rademacher Term.** The Rademacher complexity of $\mathcal{G} = \{\ell(z, f(z)) : f \in \mathcal{F}\}$ satisfies

$$\mathcal{R}_N(\mathcal{G}) \leq C_\mathcal{F} \sqrt{\frac{\text{vc}(\mathcal{G})}{N}},$$

for constant $C_\mathcal{F} > 0$ depending on the bound of $\ell$ (Shalev-Shwartz & Ben-David, 2014, Theorem 26.5). To ensure $2\mathcal{R}_N(\mathcal{G}) + \sqrt{2 \log(2/\delta)/N} \leq \epsilon/2$, we require

$$N \geq C_1 \frac{\text{vc}(\mathcal{G}) + \log(2/\delta)}{\epsilon^2},$$

for some $C_1 > 0$.

**Bounding the Wasserstein Term.** By Fournier & Guillin (2015, Theorem 2), for $P_0$ with finite $p$-th moments,

$$\mathbb{E}[W_p(\hat{P}_N, P_0)] \leq \begin{cases} CN^{-1/2}, & \text{if } d < 2p, \\ CN^{-1/2}\log(1+N), & \text{if } d = 2p, \\ CN^{-p/d}, & \text{if } d > 2p, \end{cases}$$

for constant $C > 0$. With probability at least $1 - \delta/2$, Talagrand's inequality (Blanchet et al., 2024) gives $W_p(\hat{P}_N, P_0) \leq \epsilon/(2L)$ if

$$N \geq C_2 \left(\frac{L}{\epsilon}\right)^{\max\{2, d/p\}},$$

where $L = L_\ell \sum_{k=1}^{K} \alpha_k$ follows from the multimodal cost $c(z, z') = \sum_{k=1}^{K} \alpha_k \|z_k - z'_k\|_p^p$, with gradient $\|\nabla c\| \leq \sum_k \alpha_k p \|z_k - z'_k\|_{p-1}^{p-1}$. $\qquad\square$

### H.5 Proof of Lemma 3.5(Asymptotic Unbiasedness of Debiased WDRO-MRO)

*Proof.* We prove asymptotic unbiasedness of the debiased WDRO-MRO estimator using empirical proq'acess theory and bias correction, adapted for multimodal finite-sample biases.

**Bias Decomposition**: The bias of $\hat{f}_{DRE}$ is

$$\mathbb{E}[\hat{f}_{DRE} - f_0] = \mathbb{E}\left[\arg\min_f R(\rho; \hat{P}_N) - \arg\min_f R(\rho; P_0)\right],$$

where $R(\rho; P) = \sup_{Q \in B_\rho(P)} \text{Regret}_Q(f)$. By Lemma 3.1, $R(\rho; P)$ is convex and $L$-Lipschitz in $P$ under $W_p$, with $L = L_\ell \sum_k \alpha_k$. The finite-sample bias arises from the empirical approximation $\hat{P}_N$, scaled by the multimodal variance $\sigma^2 = \sum_k \alpha_k^2 \mathbb{E}_{P_0}[\sigma_k^2]$.

**Finite-Sample Bias Bound**: From Theorem 3.2, with probability $1 - \delta$,

$$R(\rho; P_0, \hat{f}_{DRE}) - R(\rho; P_0, f_0) \leq LW_p(\hat{P}_N, P_0) + 2\mathcal{R}_N(\mathcal{G}) + \sqrt{\frac{2\log(2/\delta)}{N}},$$

where $\mathcal{R}_N(\mathcal{G}) = O(\sqrt{\text{vc}(\mathcal{G})/N})$. Taking expectations, the bias is

$$\mathbb{E}[\hat{f}_{DRE} - f_0] \leq \mathbb{E}[LW_p(\hat{P}_N, P_0)] + O(1/\sqrt{N}),$$

with $\mathbb{E}[W_p(\hat{P}_N, P_0)] \leq CN^{-p/\max\{2,d\}}$ (Fournier & Guillin, 2015). For multimodal costs, $\alpha_k$ scales $L$, tightening the bias term as $\alpha_k$ prioritizes high-variance modalities.

**Debiasing Correction**: Define the debias term $b_N = \mathbb{E}[\hat{f}_{DRE} - f_0|\hat{P}_N] \approx L\mathbb{E}[W_p(\hat{P}_N, P_0)] + O(1/\sqrt{N})$, estimated via bootstrap or double robustness methods (Blanchet et al., 2022). The debiased estimator $\hat{f}_{deb} = \hat{f}_{DRE} + b_N$ satisfies

$$\mathbb{E}[\hat{f}_{deb}] = \mathbb{E}[\hat{f}_{DRE}] + \mathbb{E}[b_N] = f_0 + o(1),$$

as $N \to \infty$, since the bias term $b_N = O(1/N)$ vanishes asymptotically. For multimodal settings, $b_N$ is corrected by weighting the variance $\sigma^2 = \sum_k \alpha_k^2 \sigma_k^2$, ensuring unbiasedness across heterogeneous modalities.

**Asymptotic Unbiasedness**: By the law of large numbers for empirical processes and the continuity of the regret functional (Lemma 3.1), the bias correction $b_N \to 0$, yielding

$$\mathbb{E}[\hat{f}_{deb} - f_0] \to 0.$$

The rate is $O(1/N)$ under strict convexity, with $\alpha_k$ modulating the variance in the correction term. $\quad\square$

## I    Proofs of Section 3.4(Regularization and Robustness Properties)

### I.1    Proof of Lemma 3.6(Variational Regularization Equivalence)

*Proof.* We prove the equivalence using duality and Fenchel conjugates, adapted for multimodal costs.

**Primal Formulation.** The WDRO-MRO problem is

$$\inf_{f \in \mathcal{F}} \sup_{Q \in U_\rho(\hat{P}_N)} \mathbb{E}_Q[\ell(z, f(z))] - \inf_{f' \in \mathcal{F}} \mathbb{E}_Q[\ell(z, f'(z))],$$

where $U_\rho(\hat{P}_N) = \{Q \in \mathcal{P}(\mathcal{Z}) : W_p(\hat{P}_N, Q) \leq \rho\}$, with multimodal cost $c(z, z') = \sum_{k=1}^{K} \alpha_k \|z_k - z'_k\|_p^p$.

**Dual Reformulation.** By Proposition 3.4, we have

$$\sup_{Q \in U_\rho(\hat{P}_N)} \mathbb{E}_Q[\ell(z, f(z))] = \inf_{\lambda \geq 0} \left\{ \lambda \rho + \mathbb{E}_{\hat{P}_N} \left[ \sup_{z'} \ell(z, f(z')) - \lambda c(\hat{z}, z') \right] \right\},$$

and similarly for the regret baseline term. Hence, the WDRO-MRO becomes

$$\inf_{f \in \mathcal{F}} \inf_{\lambda \geq 0} \lambda \rho + \mathbb{E}_{\hat{P}_N} \left[ \sup_{z'} \ell(z, f(z')) - \lambda c(\hat{z}, z') \right]$$

$$- \inf_{f' \in \mathcal{F}} \inf_{\lambda' \geq 0} \left\{ \lambda' \rho + \mathbb{E}_{\hat{P}_N} \left[ \sup_{z''} \ell(z, f'(z'')) - \lambda' c(\hat{z}, z'') \right] \right\}.$$

**Fenchel Conjugate Interpretation.** For $p = 1$ and convex $\ell(z, v)$, the supremum over $z'$ can be interpreted via the Fenchel conjugate $\ell^*$ evaluated at $\lambda \nabla_z c(\hat{z}, z')$ (Gao et al., 2024). Since $c(z, z') = \sum_k \alpha_k d_k(z_k, z'_k)$, the regularization term decomposes accordingly. By (Azizian et al., 2023), this induces a weighted total variation regularization:

$$\text{TV}(f) = \sum_k \alpha_k \text{TV}_k(f_k), \quad \text{with } \text{TV}_k(f_k) = \sup \sum_j |f_k(z_{k,j+1}) - f_k(z_{k,j})|.$$

**Special Case for Linear $f$.** For linear $f(z) = \sum_k f_k(z_k)$, the problem reduces to ERM plus a total variation penalty with coefficient $\gamma = \lambda \rho$, as shown in (Gao et al., 2024).

**Generalization to $p > 1$.** For $p > 1$, the penalty generalizes to higher-order smoothness norms (e.g., Sobolev or gradient norms), and the convergence rate scales pwith $\rho^{1/p}$ (Azizian et al., 2023). $\quad \square$

### I.2 Proof of Lemma 3.7(Multimodal Lipschitz Regularization Equivalence)

*Proof.* We prove the equivalence by reformulating the WDRO-MRO dual and specializing to linear multimodal models.

**Dual Reformulation of WDRO Risk Term.** By strong duality (Proposition 3.4),

$$\sup_{Q \in \mathcal{U}_\rho(\hat{P}_N)} \mathbb{E}_Q[\ell(y, w^\top x)] = \inf_{\lambda \geq 0} \lambda \rho + \mathbb{E}_{\hat{P}_N} \left[ \sup_{x'} \ell(y, w^\top x') - \lambda c(x, x') \right].$$

For linear models and losses like logistic (1-Lipschitz in $v$), the inner sup is bounded by the Lipschitz property:

$$\sup_{x'} \ell(y, w^\top x') - \lambda c(x, x') \leq \ell(y, w^\top x) + \lambda \sup_{x'} |w^\top (x' - x)| - c(x, x').$$

**Lipschitz Dual Emergence.** The term $\sup_{x':c(x,x') \leq \rho/\lambda} |w^\top (x' - x)|$ is the effective Lipschitz extension. Since $c(x, x') = \sum_k \alpha_k \|x_k - x'_k\|_p^p$, by Hölder inequality,

$$|w^\top (x' - x)| = \left| \sum_k w_k^\top (x'_k - x_k) \right| \leq \sum_k \|w_k\|_q \|x'_k - x_k\|_p,$$

where $q = p/(p-1)$. The constraint $\sum_k \alpha_k \|x_k - x'_k\|_p^p \leq \rho/\lambda$ implies a weighted ball. Maximizing over perturbations yields

$$\sup |w^\top (x' - x)| = (\rho/\lambda)^{1/p} \|w\|_*,$$

where the dual norm $\|w\|_* = \sup_{\sum_k \alpha_k \|u_k\|_p^p \leq 1} \sum_k w_k^\top u_k$. By inf-convolution duality for additive costs (separability from Assumption 2.1),

$$\|w\|_* = \inf_{\beta_k \geq 0, \sum \beta_k = 1} \sum_k \frac{\|w_k\|_q}{\alpha_k \beta_k}.$$

Thus, the risk term becomes $\mathbb{E}[\ell] + \gamma \|w\|_*$, with $\gamma = \lambda \rho^{1/p}$.

**Regret Term Handling.** The regret baseline $\inf_{w'} \mathbb{E}_Q[\ell(y, (w')^\top x)]$ dualizes similarly, subtracting an identical reg term (since inf over $w'$ yields the same dual form, constant in $w$). By Sion's minimax theorem (convex-concave), the overall is equivalent to reg-ERM with weighted Lipschitz penalty.

**Modality-Specific Robustness.** Higher $\alpha_k$ reduces the penalty for modality $k$ in $\|w\|_*$, allowing larger $w_k$ (less regularization) for stable modalities, while low $\alpha_k$ tightens constraint for noisy ones. □

### I.3  PROOF OF PROPOSITION 4.1(ENVELOPES FOR LOGISTIC; TRACTABLE PER P)

*Proof.* By the DRO duality for optimal transport ambiguity sets, the worst-case expectation admits the envelope form with zero duality gap under mild regularity (upper semicontinuity, IP), hence the strong-dual "canonical objective" with epigraph variables $s_i$ is valid; see Kuhn et al. (2025, Theorem 4.18 & Lemma 4.16).

For $p = 1$, when $\ell$ is $L$-Lipschitz in $v = w^\top x$, the envelope equals $\mathbb{E}_{\hat{P}_N}[\ell] + \lambda \rho L$ (specializing Proposition 6.17), which yields an LP via standard absolute-value auxiliaries; cf. Kuhn et al. (2025, Prop. 6.17).

For $p = 2$, using the convex conjugate of the logistic loss together with the quadratic cost conjugate $c^*$, the envelope reduces to a conic program representable as an SDP, and to an SOCP under diagonal/rotated-quadratic structure; this is the standard Fenchel–Moreau route in our WDRO–MRO derivations, see Lemma B.10 therein.

For $2 < p < \infty$, the cost conjugate $c^*$ admits a Hölder-type form with $q = p/(p-1)$, which is power-cone representable for rational $p$ (and exponential-cone for irrational $p$). Hence the envelope is a convex conic program; see Lemma B.4.

For $p = \infty$, the $\ell_\infty$-ball uncertainty reduces the envelope to vertex (box) constraints, which are LP/SDP-representable; see the tractability table and corresponding Lemmas in 3.2.1.

Collecting these cases gives the claimed tractable envelopes per $p$, all as finite-dimensional convex conic programs with zero duality gap and attained optima under our standing assumptions. □

### I.4  EMPIRICAL OBSERVATIONS OF REMARK 4.1 IN EXPERIMENTS

To show the relation that larger $\alpha_k$ (more trusted modality) yields weaker shrinkage on $w_k$, we vary one modality weight over [0.25, 0.5, 1.0, 2.0, 4.0, 8.0] while keeping all other modality weights fixed at 1.0, shown in Figure 2.

## J  ADDITIONAL EXPERIMENTAL DETAILS

### J.1  PREPROCESSING PIPELINE

Figure 3 in Appendix J.1 illustrates the preprocessing and splitting pipeline. Following Dörrich et al. (2025), we preprocess and integrate five modalities: **Demographics** (age, gender, and related variables), **Blood parameters** (routine test values, z-score normalized), **Pathological features** (tumor grading, stage, and lymph node status), **ICD codes** (categorical disease codes, bag-of-words encoded), and **TMA cell density** (CD3/CD8 immune cell infiltration counts). Data is separated into training (80%, 612 patients) and test (20%, 151 patients) sets, with 5-fold cross-validation for hyperparameter tuning. We consider three evaluation splits: *in-distribution (ID)*, *out-of-distribution (OOD)*, and an *Oropharynx-specific* split.

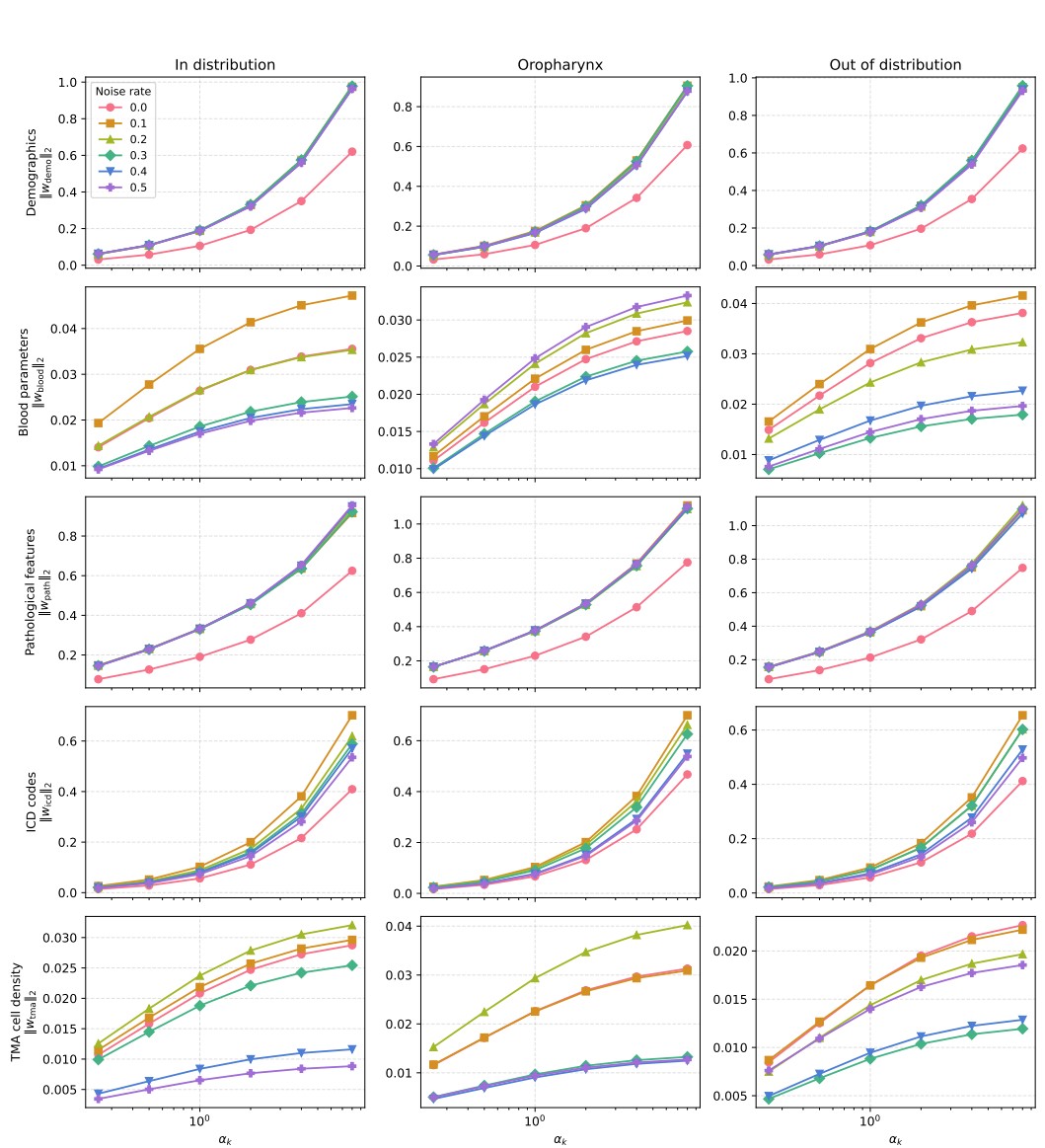

Figure 2: Relation between $\alpha_k \in [0.25, 0.5, 1.0, 2.0, 4.0, 8.0]$ and $||w_k||^2$ for noise rates $\in [0.0, 0.1, 0.2, 0.3, 0.4, 0.5]$.

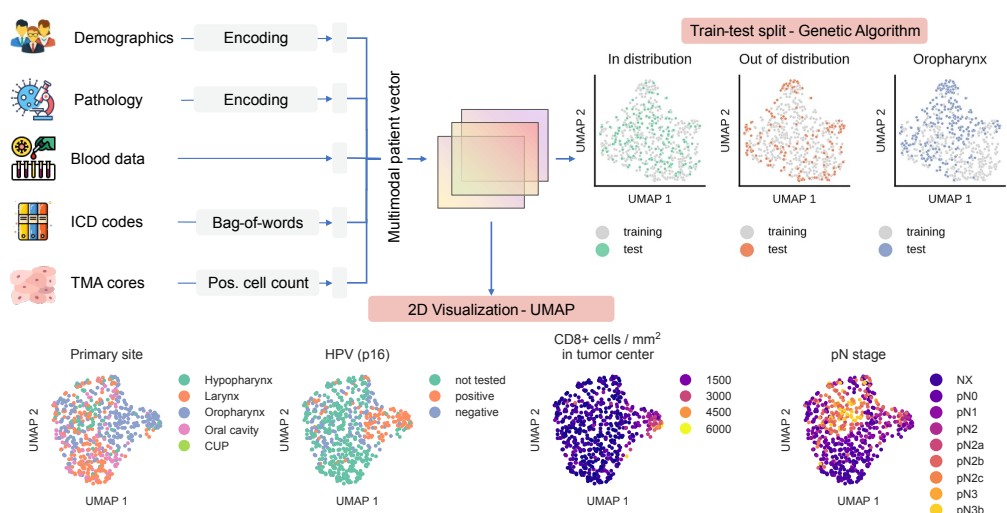

Figure 3: Preprocessing and splitting pipeline for the HANCOCK dataset (Dörrich et al., 2025). Multiple modalities (Demographics, Pathology, Blood, ICD, TMA) are integrated into multimodal patient vectors, visualized with UMAP, and split into training/testing sets using a genetic algorithm.

## J.2 METRICS

Table 4: Definitions of evaluation metrics. ↑ indicates higher is better, ↓ indicates lower is better.

| Abbrev. | Full Name | Definition / Formula |
|---|---|---|
| *Performance* | | |
| Avg AUC ↑ | Average AUC | Mean ROC-AUC across all noise rates and trials. |
| Std AUC ↓ | Standard Deviation of AUC | Variability of ROC-AUC across repeated trials. |
| *Robustness* | | |
| Robust AUC ↑ | Robust AUC | Worst-case (minimum over noise rates) mean AUC. |
| RR-AUC ↑ | Relative Robustness AUC | Robust AUC$/ \max_\rho\{\mathrm{AUC}(\rho)\}$. |
| W.C. Drop ↓ | Worst-Case Drop | $\max_\rho\{\mathrm{AUC}(\rho)\} - $ Robust AUC. |
| *Stability* | | |
| NS Drop ↓ | Noise Sensitivity Drop | $\mathrm{AUC}(\rho = 0) - $ Robust AUC. |
| NS Slope ↓ | Noise Sensitivity Slope | Slope of regression of AUC vs. noise rate $\rho$. |
| *Fairness* | | |
| GNR ↑ | Group-Noise Robustness | $\min_{g,\rho}\{\mathrm{AUC}_g(\rho)\}$. |
| GF Gap ↓ | Group-Fairness Gap | $\max_g \overline{\mathrm{AUC}}_g - \min_g \overline{\mathrm{AUC}}_g$. |

## J.3 GROUP DISTRIBUTIONALLY ROBUST OPTIMIZATION (GROUP DRO)

In addition to the instance-level Wasserstein ambiguity sets considered in the main text, we include *Group DRO* (Sagawa et al., 2020) as a baseline method. Group DRO assumes that data points are partitioned into $G$ predefined groups (e.g., tumor sites or clinical subpopulations), and seeks a model whose loss is uniformly controlled across all groups.

**Formulation.** Let $\{S_g\}_{g=1}^G$ denote the index sets corresponding to each group. For a model $f$ with parameters $\theta$ and loss $\ell(z, f(z))$, define each group loss as

$$L_g(\theta) = \frac{1}{|S_g|} \sum_{i \in S_g} \ell(z_i, f_\theta(z_i)).$$

Group DRO solves the minimax problem

$$\min_\theta \max_{g \in \{1, \dots, G\}} L_g(\theta), \tag{2}$$

which guarantees that performance is optimized for the worst group.

**Convex Logistic Regression Case.** In our experiments, $f_\theta$ is a linear classifier $f_\theta(z) = w^\top z + b$ with logistic loss $\ell(y, v) = \log(1 + \exp(-yv))$. Problem equation 2 admits the convex reformulation

$$\min_{w,b,t} \quad t$$
$$\text{s.t.} \quad \frac{1}{|S_g|} \sum_{i \in S_g} \log\big(1 + \exp\big(-y_i(w^\top x_i + b)\big)\big) \le t, \qquad g = 1, \dots, G, \tag{3}$$

which can be solved using standard convex programming tools (e.g., MOSEK). This formulation is structurally aligned with the LP/SOCP/SDP reformulations used in WDRO and WDRO-MRO, enabling fair comparison. Group DRO provides robustness against *protected groups* and *subpopulation shifts*, complementing the instance-level perturbation robustness captured by Wassserstein DRO and the regret-based robustness in WDRO-MRO. It serves as a strong baseline that ensures:

$$\text{Group-level fairness} \quad \Longleftrightarrow \quad \max_g L_g(\theta) \text{ is small,}$$

which is distinct from (i) *distributional shifts* modeled via Wasserstein balls, and (ii) *model-based adversarial perturbations* arising in the minimax regret objective.

## J.4 ADDITIONAL RESULTS

### J.4.1 ADDITIONAL RESULTS FOR UNIFORM MODALITY WEIGHTS, $\alpha_k = 1.0$

The results presented in Figures 1 and 4 to 7 were generated using uniform modality weights, with $\alpha_k = 1.0$ for all $k$.

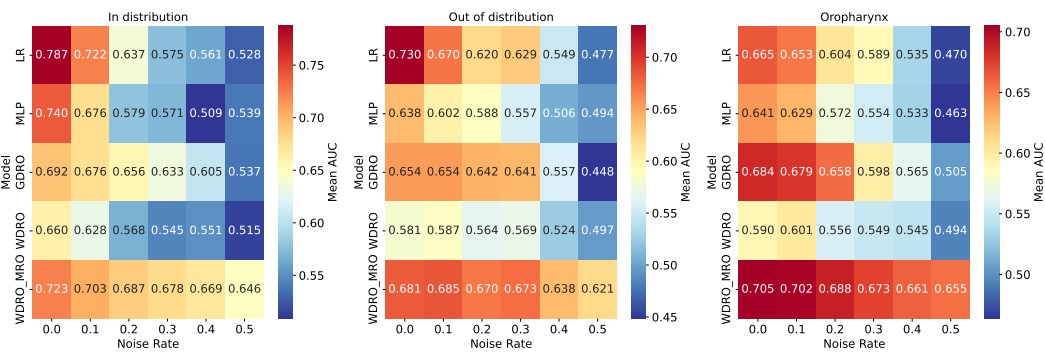

Figure 4: WDRO-MRO shows the strongest robustness to label noise on the HANCOCK dataset: although LR achieves the best AUC on clean data splits ($\rho = 0.0$, in distribution, out of distribution), its performance degrades with increasing noise, while WDRO-MRO maintains consistently higher AUC at moderate and high noise levels, and dominates across all noise rates on the Oropharynx data split. Heatmaps report mean AUC for each model (rows) under different noise rates $\rho \in \{0.0, 0.1, ..., 0.5\}$ (columns), with color intensity indicating performance.

## K USE OF LARGE LANGUAGE MODELS

Large Language Models (LLMs) were used to improve grammar and readability of the text.

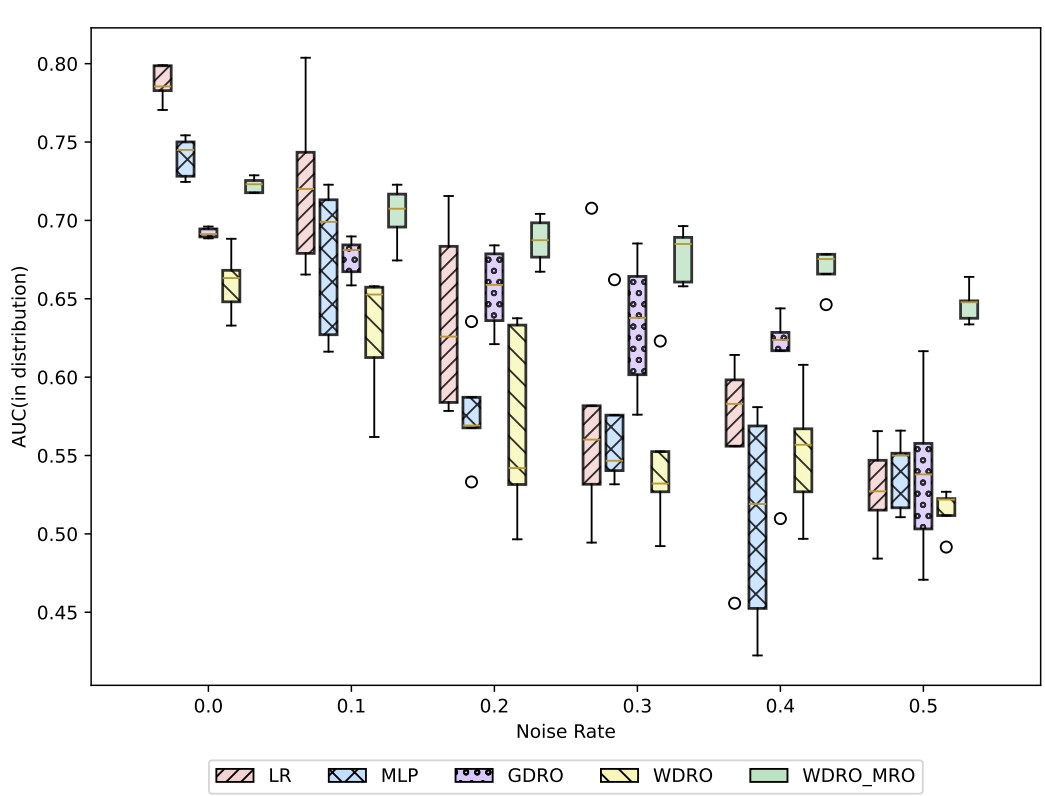

Figure 5: In-distribution split: Boxplots show the distribution of AUC across 5 random seeds under increasing noise rates ($\rho \in \{0.0, 0.1, 0.2, 0.3, 0.4, 0.5\}$).

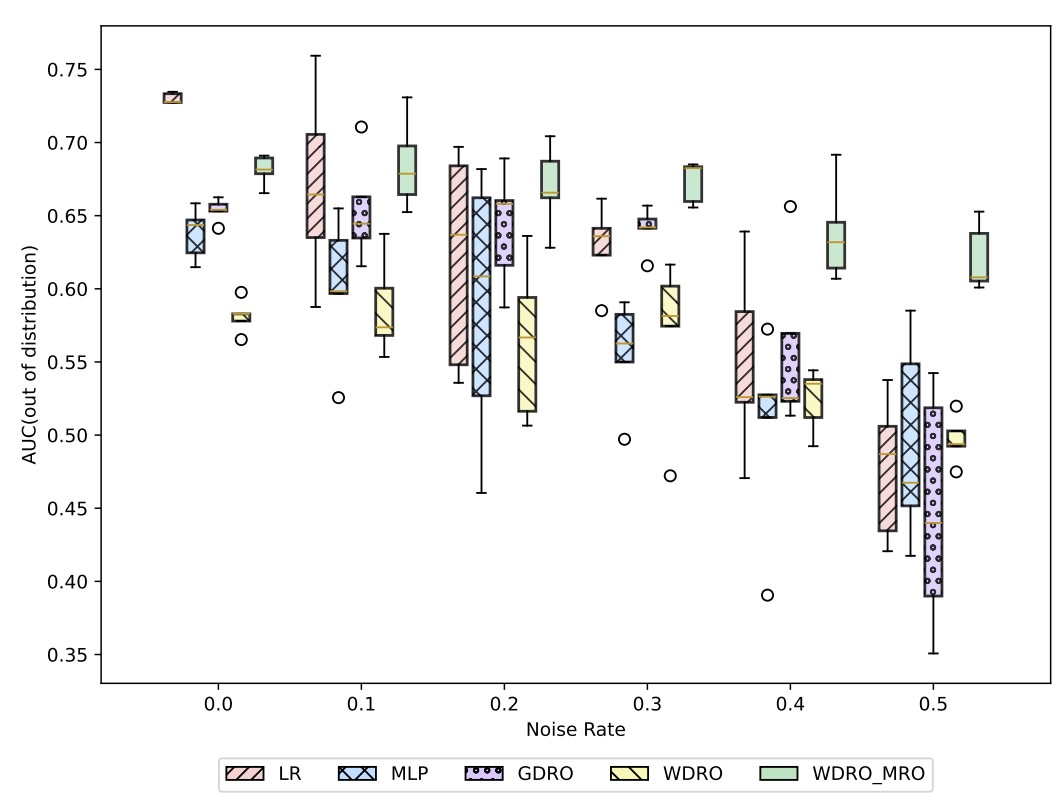

Figure 6: Out-of-distribution split: AUC distributions across seeds for LR, MLP, GDRO, WDRO, and WDRO-MRO.

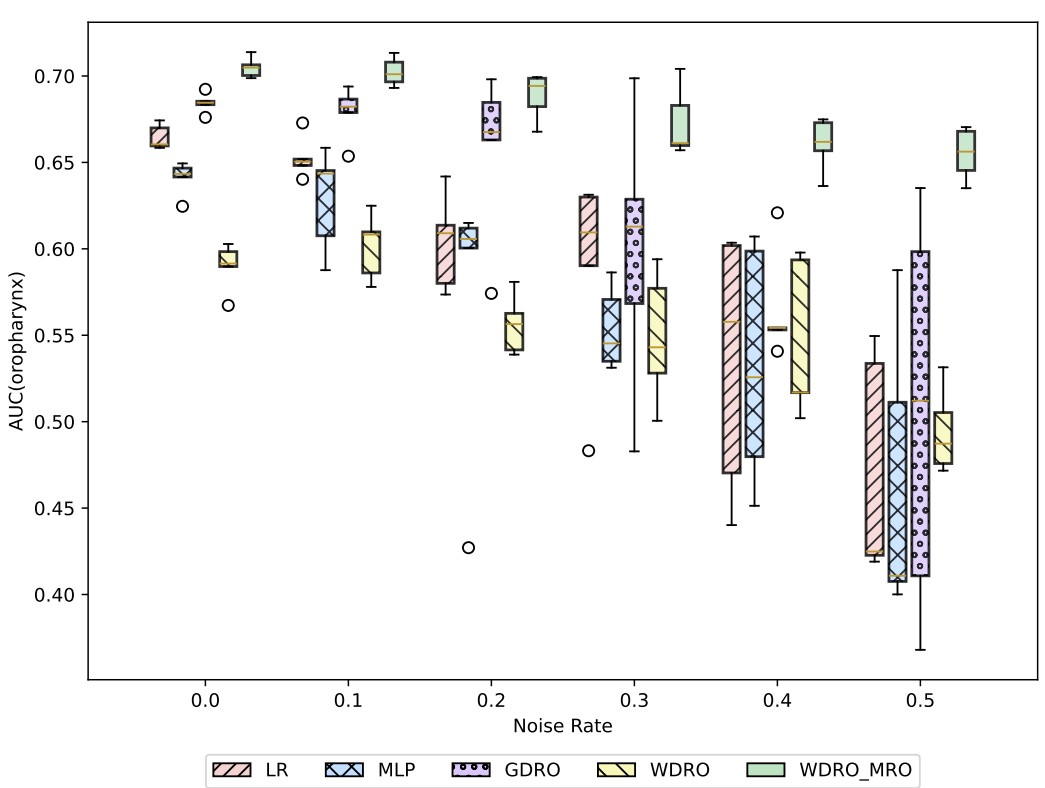

Figure 7: Oropharynx split: Boxplots highlight that WDRO-MRO dominates across all noise levels, achieving both higher AUC and smaller variance compared to LR, MLP, GDRO and WDRO, demonstrating strong robustness in this data split.

