# OpenReview forum: "Wasserstein Distributionally Robust Minimax Regret Optimization for Multimodal Machine Learning"
_ICLR.cc/2026/Conference — Submitted to ICLR 2026_

### Official Review · Reviewer_MJso · 2025-10-17

**Soundness:** 3
**Presentation:** 2
**Contribution:** 3
**Rating:** 6
**Confidence:** 2

**Summary:**

This paper proposes WDRO-MRO, a framework combining WDRO with min-max regret to improve robustness in multimodal ML. It provides theoretically grounded formulation, tractable convex reformulations and a provable convergent dual-game solver. Experiments have shown strong robustness and fairness over ERM and DRO.

**Strengths:**

1.	The unified formulation is novel and the theoretical foundations are strong.

2.	It gives tractable convex reformulation, interpretability and regularization links.

3.	Experimental results show good robustness and fairness.

**Weaknesses:**

1.	This paper is very technical and notations are too heavy. It is hard to readers to track their theory and proofs. I suggest the authors to introduce proof sketches for each lemma or theorem to make readers easy to follow. Furthermore, a notation table is welcome to improve the readability.

2.	The experimental evaluation is not that enough. Only one dataset, HANCOCK is evaluated. Only logistic regression is considered. There are lack of multimodal models/datasets to verify the effectiveness.

3.	Although WDRO-MRO shows superiority against WDRO, the overall performance seems not satisfactory. AUCs of WDRO-MRO are less than 0.7, it is very hard for hospitals and doctors using WDRO-MRO as their machine learning paradigm to analyze cancers. Therefore, the proposed method may remain theoretically and not be practical. Is HANCOCK dataset too hard for learning?

4.	WDRO-MRO may induce further computational overhead compared with standard DRO.

**Questions:**

1.	How sensitivie are results to the choice of modality weights $\alpha_k$? Is there any automatic tuning?

2.	Can the dual-game solver extent to non-convex case?

3.	How is fairness measured across modalities rather than patient subgroups?

---

> ### Author Response · Authors · 2025-12-03
>
> ### Response to Q1 (modality weights)
>
> In the main experiments the modality weights are set to $\alpha_k = 1$ for all $k$, so all modalities are treated as equally reliable. We added an ablation (Appendix I.4) where we vary a single $\alpha_k \in {0.25, 0.5, 1.0, 2.0, 4.0, 8.0}$ while keeping others fixed at 1. The results show that increasing $\alpha_k$ leads to larger $\lVert w_k\rVert_2^2$, , and performances are relatively stable across different $\alpha_k$.
>
> Automatic tuning or learning of $\alpha_k$ is an interesting future direction.
>
>
> ### Response to Q2 (non-convex case)
>
> Yes, at the algorithmic level. Our convergence guarantees are for convex models, but the dual-game structure extends naturally:
>
> - The adversary update (exponentiated weights over samples) remains unchanged.
>
> - The “Learner / Oracle updates” can be replaced by one or a few stochastic gradient steps over mini-batches for a non-convex network, in the same style as standard adversarial training or non-convex DRO.
>
> We added Remark 3.1 to further explain this.
>
> ### Response to Q3 (fairness)
> Thanks for the question. The groups used in the fairness metrics in experiments are **modality-induced evaluation groups**, not demographic or clinical subpopulations.
>
> In Table 4, the fairness-style metric (Group Fairness Gap, GF Gap) is defined as:
>
> $$
> \mathrm{GF\ Gap} = \max_{g}\ \overline{\mathrm{AUC}}_{g} - \min_{g}\ \overline{\mathrm{AUC}}_{g}
> $$
>
> where $g$ indexes the **modality–noise groups** created by independently perturbing each modality at different noise levels $\rho$. These groups therefore correspond to different **modality degradation conditions**, rather than patient-level attributes.
>
> This evaluation design reflects the central robustness question in multimodal oncology datasets: different modalities can degrade independently (e.g., scanner variation affecting TMA-derived features but not ICD codes), and we therefore assess whether the model maintains **modality-level robustness consistency**. This can be viewed as a form of *fairness across modality-induced groups* in the sense that we measure how evenly performance is preserved when each modality is stressed. However, this notion is conceptually distinct from demographic fairness: the “groups’’ here arise from modality-specific perturbations, not from population subgroups.
>
> We will clarify this distinction in Section 4.2 and in the description of Table 4 to avoid confusion between modality-induced groups used for robustness parity and patient-level groups used in classical fairness settings.

---

### Official Review · Reviewer_squb · 2025-10-24

**Soundness:** 2
**Presentation:** 2
**Contribution:** 2
**Rating:** 2
**Confidence:** 3

**Summary:**

The paper addresses the challenge of robust multimodal machine learning under modality-specific distributional shifts, where standard ERM fails and traditional DRO can be overly conservative. It proposes Wasserstein Distributionally Robust Minimax Regret Optimization (WDRO-MRO), which minimizes worst-case regret relative to an oracle using a modality-weighted Wasserstein ambiguity set to provide decision-centric robustness. Key results include theoretical guarantees on optimization and statistics, tractable convex reformulations, a dual-game algorithm, and empirical improvements in accuracy, robustness, and fairness on the HANCOCK healthcare dataset compared to ERM and standard DRO.

**Strengths:**

1. The paper provides a comprehensive theoretical foundation for the WDRO-MRO framework. The analysis in Section 3 is thorough, covering basic optimization properties (existence, uniqueness, convexity, duality) , detailed computational properties (tractable reformulations) , and strong statistical guarantees (consistency, finite-sample bounds, convergence rates) .

2. The paper addresses a gap: robust multimodal learning under heterogeneous, modality-specific noise. The synthesis of minimax regret (to avoid DRO's conservatism ) with a modality-weighted Wasserstein cost is a novel and intuitive solution to this problem.

3.  A significant strength is showing that this seemingly complex min-max-min-max problem can be reformulated into a variety of standard, tractable convex programs (LP, SOCP, SDP). This makes the framework practical and usable.

**Weaknesses:**

1. Several symbols appear before being defined or are insufficiently specified, e.g.:
   - $F_m$ and $g$ (line 164)
   - $\lambda_{\max}$ (line 236)
   - $\Delta([N])$ (line 253)
   - $\sigma_k^2$ (line 311)
   - $d_k$ (line 320)
   - $L_\ell$ (line 323)
   - $s_i(f_t,\lambda_t)$ (line 231)

2. Notation inconsistencies: for instance, $\mathcal U$ (line 94) and $\mathcal B$ (line 298) appear to denote the same ambiguity set.

3. Empirical reporting (in the provided extract) lacks detail on metrics, baseline protocols, and statistical significance testing.

4. Novelty may partially overlap with prior regret-based DRO; the multimodal extension could read as incremental without stronger empirical breadth.

5. Proposition 2.1 states a standard strong duality result in DRO; it would be cleaner to cite a canonical reference and relegate proof details to the appendix.

6. In Proposition 3.2, $\phi(f)$ (or $\varphi(f)$) is used before being defined.

7. The hierarchy and labeling of Lemmas/Propositions/Theorems sometimes do not reflect their relative importance in the argument.

8. Consider moving Lemmas 3.1–3.4 to the appendix to streamline the main narrative.

9. The extension to nonconvex multimodal deep fusion is deferred to future work, leaving current evidence limited to (e.g.) logistic models.

10. Experiments use a single dataset (HANCOCK) and two baselines (ERM-LR, WDRO-LR); comparisons to group-DRO and modern multimodal architectures, plus ablations over $p$, $\alpha_k$, and $\rho$, are missing.

**Questions:**

1. How were hyperparameters tuned for ERM vs WDRO vs WDRO-MRO to ensure parity? Provide a table.


2. Can you compare to group-DRO and at least one modern multimodal deep baseline to assess gains beyond linear models?
3. Please expand the main-text assumptions and proof sketch for Lemma 3.6 and the regularization equivalences (Lemmas 3.10–3.12). What exact tail/geometry assumptions are needed?
4. Your theoretical tractability results rely on convexity. For the non-convex deep learning case , would the "Learner / Oracle updates" in Algorithm 1 be replaced by, for example, a single stochastic gradient step?
5. Can you provide runtime comparisons for the reformulations (e.g., LP vs. SDP) on real data?

---

> ### Author Response · Authors · 2025-11-25
>
> ### Response to q1-hyperparameters
> Most hyperparameters use default values in the experiments. Table 1 summarizes the hyperparameters for all models. No per-split tuning was performed.
>
> | Method            | Main regularization          | Other hyperparameters                                        |
> |-------------------|------------------------------|---------------------------------------------------------------|
> | ERM (LR)          | C = 1.0 (L2)                 | solver = liblinear, max_iter = 1000, class_weight = balanced |
> | ERM (MLP)         | L2 regularization (alpha = 0.0001) | hidden_layer_sizes = (100,), activation = relu, solver = adam, max_iter = 500 |
> | GDRO              | reg = 1.0 (L2)               | solver = MOSEK             |
> | WDRO (LR)         | ρ = 0.02, κ = 1.0            | solver = MOSEK                                               |
> | WDRO-MRO (game)   | ρ = 0.02, α_k = 1            | p = 2, η = 0.01, η_λ = 0.005, T = 10, solver = MOSEK          |
>
>
> ### Response to q2-baselines
> We have added both group-DRO and an MLP baseline in the revision. The group-DRO implementation follows the standard worst-group objective (minimizing the maximum group-wise loss) (details in Section J.3). The MLP baseline is a multimodal early-fusion model that concatenates all modality features and predicts the target. Updated results are reported in Figure 1, Figures 4–7 and Table 2, which show that WDRO-MRO consistently outperforms these additional baselines.
>
> ### Response to q3-assumptions and proof sketch
> We have added the main-text assumptions, proof sketch for Lemma 3.6, and clarified assumptions for regularization equivalences.
> - Lemma 3.7 (Lipschitz Regularization Equivalence) requires convex and $L$-Lipschitz scalar losses and the separable transport cost;
> - the induced penalty $\|w\|_*$ is finite under the stated moment/tail conditions;
> - Lemma 3.8 (Convergence to Multimodal ERM) requires no additional tail assumptions beyond the standing assumptions of Section 3, since continuity in $\rho$ follows from dominated convergence.
>
> Please see Section 3.4 Regularization and Robustness Properties for details.
>
> ### Response to question 4-nonconvex
> Thanks for your comments. The tractability and convergence guarantees are derived under convexity assumptions (e.g., linear / logistic models), and all results in Sections 3-4 are intended for this convex setting.
>
> For non-convex deep architectures, Algorithm 1 can indeed be instantiated as a first-order minmax procedure: in each outer iteration, the "Learner / Oracle updates" are implemented by one or a small, fixed number of stochastic gradient steps on mini-batches, while the adversary distribution is still updated via the exponentiated-weights rule. In other words, the exact bestresponse updates in Algorithm 1 are replaced with approximate SGD-based updates, which is standard in non-convex DRO and adversarial training.
>
> We do not claim new convergence guarantees in this general non-convex setting; instead, we view the SGD-based variant as a practical extension of WDRO-MRO for deep models. We have added Remark 3.1 ("Non-convex deep models") in Section $3.2.2$ to clarify this in the revised manuscript.

---

> > ### Comment · Reviewer_squb · 2025-11-27
> >
> > Thank you for the detailed rebuttal. Despite the helpful explanations, I find that the manuscript still requires substantial revision to fully address the concerns raised. I will therefore keep my original assessment.

---

### Official Review · Reviewer_EWQY · 2025-10-31

**Soundness:** 3
**Presentation:** 1
**Contribution:** 3
**Rating:** 6
**Confidence:** 3

**Summary:**

The paper presents a new framework for distributionally robust optimization relying on (i) expressing the min-max objective in terms of regret and not just adversarial expectations, (ii) using a separable metric to define Wasserstein distances adapted to various modalities.

The authors analyze first the soundness of their framework: does a solution exits, is the problem convex or not, can we reformulate it? Then the authors instantiate the generic objective they propose with various distances and various losses. For each case they present a formulation of the dual problem amenable to solvers for convex optimization. They propose a generic optimization algorithm and present optimization convergence guarantees for it. They present sensitivity analyses and statistical properties.

The authors conduct experiments using their framework with a logistic loss on a dataset of medical records with various modalities. Their approach appears to outperform a basic empirical risk minimization and a standard distributionally robust optimization method using Wasserstein distances.

**Strengths:**

- The results are extensive. In total, the paper consists in 22 theoretical results (lemmas/theorems/propositions).
- The authors analyzed the core properties of their framework (optimization/statistical properties) as well as interesting reformulations in primal and dual space.
- The experiment illustrates well the framework, though more details could help.
- Lemmas 3.10, 3.11 make interesting connections with simple regularized forms of ERM
- Many other results may be of particular interest but the presentation makes it hard to appreciate each of them.

**Weaknesses:**

- This is not a conference article. There is not enough space to introduce properly each result, (see many questions/remarks below).
- It is unclear how incremental the framework is on some of its contributions. Typically, considering the problem in terms of regret may shift perspectives on performance criterions towards more sensible ones, and it implies many changes I believe compared to usual Wasserstein based frameworks. On the other hand, using a separable distance seems to be of small novelty. That said, even if the proposed changes are incremental, the paper is quite complete about it. All "required" results are there.
- The assumptions may lack some details (see questions below).
- The experimental and implementation parts lack details. See questions below.

**Questions:**

1. Assumptions
a. Assumption 2.2 is quite restrictive. Most losses are not bounded. Even Lipschitz-continuity may fail to hold in many cases.
b. What do you mean by "closed convex class"? Is it a closed convex set of functions? What metric is used to define closed sets here? Or did you mean a class of closed convex functions?
c. What is the prediction range?
d. What is exactly the outer objective?
e. Why is Proposition 2.1 appearing below the assumptions? What's the point of this proposition?

3. Algorithms
a. What is the empirical dual subgradient?
b. What do you mean by "implementable" line 398? Other methods can be implemented a priori.

4. Experiments,
a. What is exactly the method that is used? Three methods are presented in 4.1.
b. What distance is used at the end? Given all the possible cases of norm, were they all compared?

5. Conclusion,
The authors say line 471 that "WDRO-MRO provides a decision-centric notion of robustness that naturally connects performance and fairness within a tractable optimization framework". Can they justify this sentence?

Remarks:
- line 65: Define somewehre the wasserstein cost for the unfamiliar reader. That way, the reader will directly understand where c intervenes.
- line 94: What the p in $W_p$? I suppose it is the norm used in the $d_k$ defining c? Maybe remove p since the distances were not defined in terms of norms in the first place.
- line 143: phi(f) has not been defined
- line 165: "standard in machine learning" give a reference to justify.
- Proposition 3.6: proposition 3.6 is quoted inside proposition 3.6
- line 430: add a reference about SMOTE oversampling

---

> ### Author Response · Authors · 2025-11-27
>
> We sincerely thank the reviewer for the detailed and constructive feedback.
> Below we address each question and indicate how we will revise the paper.
>
> ---
>
> ### 1. Assumptions
>
> **(a) Assumption 2.2: bounded / Lipschitz losses**
>
> > Assumption 2.2 is quite restrictive. Most losses are not bounded. Even Lipschitz-continuity may fail to hold in many cases.
>
> We agree that the current wording is misleading and we appreciate the opportunity to clarify.
> Our analysis does **not** require globally bounded losses; what we need is **Lipschitz continuity in the prediction variable on the prediction range** generated by the model class. Concretely, we will revise Assumption 2.2 to:
>
> > “For each fixed input $z$, the loss $\ell(z,v)$ is convex and $L$-Lipschitz in $v$ on the prediction range $\{f(z) : f \in \mathcal{F}\}$.”
>
> This condition is satisfied by standard supervised learning losses (logistic, hinge, smoothed cross-entropy, square loss on bounded features) and is only used to ensure that the dual envelopes are finite and well-defined. We will explicitly remove the impression that global boundedness is required.
>
> ---
>
> **(b) “Closed convex class”**
>
> > What do you mean by "closed convex class"? Is it a closed convex set of functions? What metric is used to define closed sets here? Or did you mean a class of closed convex functions?
>
> Thank you for catching this ambiguity. We indeed mean a **closed and convex subset of measurable functions** under the function-space topology induced by the relevant norm (e.g., supremum norm on the prediction range). In the revision we will clarify:
>
> > “$\mathcal{F}$ is a convex set of measurable functions that is closed under the norm topology induced by $\|f\|_{\infty} = \sup_{z \in \mathcal{Z}} |f(z)|$ (or, more generally, the natural norm induced by the Wasserstein geometry).”
>
> We do **not** mean that each $f \in \mathcal{F}$ is a convex function in its argument; instead we assume that the **set** $\mathcal{F}$ is convex and closed.
>
> ---
>
> **(c) “Prediction range”**
>
> > What is the prediction range?
>
> We will clarify this definition in the main text. By “prediction range” we mean:
>
> > “the image set $\{f(z) : z \in \mathcal{Z}, f \in \mathcal{F}\}$, i.e., all values that the model class can output on the data domain.”
>
> The only role of this concept is to localize the Lipschitz requirement to the region where the model operates; we do not assume global regularity on $\mathbb{R}$.
>
> ---
>
> **(d) “Outer objective”**
>
> > What is exactly the outer objective?
>
> We apologize for the lack of precision. The “outer objective” refers to the **minimax regret criterion** that defines WDRO–MRO:
>
> $$
> \inf_{f \in \mathcal{F}} \sup_{Q \in \mathcal{U}_\rho(\hat{P}_N)}
> \Big( \mathbb{E}_Q[\ell(z, f(z))] - \inf_{f' \in \mathcal{F}} \mathbb{E}_Q[\ell(z, f'(z))] \Big).
> $$
>
> We will include this explicit formula at first mention of the term “outer objective” so that the structure (outer minimization over $f$, inner supremum over distributions $Q$) is fully transparent.
>
> ---
>
> **(e) Role and placement of Proposition 2.1**
>
> > Why is Proposition 2.1 appearing below the assumptions? What's the point of this proposition?
>
> Proposition 2.1 is the **strong duality / interchangeability result** that allows us to pass from the primal Wasserstein-robust objective to a tractable dual formulation. This proposition underpins all subsequent convex reformulations and the dual-game algorithm.
>
> We agree that its role should be emphasized more clearly. In the revision we will:
>
> 1. Move Proposition 2.1 to immediately follow the assumption block in Section 2,
> 2. Explicitly state:
>
> > “This proposition provides the key interchangeability principle needed for our strong duality and all tractable reformulations in Sections 3–4.”
>
> This should clarify both its placement and its central purpose.

---

> > ### Author Response · Authors · 2025-11-27
> >
> > ### 2. Algorithms
> >
> > **(a) “Empirical dual subgradient”**
> >
> > > What is the empirical dual subgradient?
> >
> > In Algorithm 1, the quantity denoted by $\widehat{\rho}_t$ is the **empirical dual subgradient with respect to the ambiguity radius**, obtained from the KKT conditions of the dual envelope. For the $p=2$ case used in our implementation, the envelope yields:
> >
> > $$
> > \widehat{\rho}_t
> > = \sum_{k=1}^K \frac{\|w_{t,k}\|_2^2}{4 \lambda_t^2 \alpha_k},
> > $$
> >
> > which forms a (super-)gradient of the dual objective in $\lambda$. The update
> > $\lambda_{t+1} \leftarrow \Pi_{[0,\lambda_{\max}]}\big(\lambda_t + \eta_\lambda(\rho - \widehat{\rho}_t)\big)$
> > is therefore a projected (sub)gradient step on the radius dual variable.
> >
> > We will add this explicit formula and a short pointer to the derivation in the appendix.
> >
> > ---
> >
> > **(b) Meaning of “implementable” (line 398)**
> >
> > > What do you mean by "implementable" line 398? Other methods can be implemented a priori.
> >
> > We agree that the term “implementable” is vague.
> > What we intended to emphasize is that the dual envelope is replaced by a **finite-dimensional convex upper bound** (LP/SOCP/SDP) that can be solved with standard optimization solvers, instead of an infinite-dimensional max over adversarial distributions.
> >
> > We will revise the wording to:
> >
> > > “We use a finite-dimensional convex upper bound on the dual envelope, which yields tractable LP/SOCP/SDP subproblems that can be solved with off-the-shelf solvers.”
> >
> > and remove the ambiguous phrase “implementable” without context.
> >
> > ---
> >
> > ### 3. Experiments
> >
> > **(a) What is exactly the method that is used?**
> >
> > > What is exactly the method that is used? Three methods are presented in 4.1.
> >
> > Thank you for pointing out that the experimental section was not explicit enough. In our implementation we use the following methods:
> >
> > - **LR**: standard empirical risk minimization with logistic regression;
> > - **WDRO**: standard Wasserstein DRO (worst-case risk minimization) with a 2-Wasserstein ball;
> > - **WDRO-MRO (dual-game)**: our proposed Wasserstein Distributionally Robust Minimax Regret Optimization, implemented via the dual-game hybrid solver described in Section 3.3.
> >
> > In the revision, we will add a small table in Section 4.1 that lists the exact methods and their code names, and we will also describe the additional Group DRO baseline used in the appendix.
> >
> > ---
> >
> > **(b) What distance is used? Are all norms compared?**
> >
> > > What distance is used at the end? Given all the possible cases of norm, were they all compared?
> >
> > We apologize for not stating this more clearly. **All experiments use a 2-Wasserstein distance**, i.e., cost
> >
> > $$
> > c(z,z') = \sum_{k=1}^K \alpha_k \|z_k - z'_k\|_2^2
> > $$
> >
> > with modality weights $\alpha_k$. The theoretical framework in Section 2 allows general $p$, but the tractable envelopes and convex subproblems developed in Section 3.3 are specialized to the $p=2$ geometry (leading to quadratic regularization terms). For this reason, we do **not** treat $p$ as a hyperparameter and we do not compare different $p$-norms in the experiments.
> >
> > We will explicitly state:
> >
> > > “In all numerical experiments we fix $p=2$, which is the case for which our dual-game solver and convex reformulations are derived. Extending WDRO–MRO to alternative cost geometries (e.g., $p=1$) is an interesting direction for future work.”
> >
> > ---
> >
> > ### 4. Conclusion and fairness statement
> >
> > > The authors say line 471 that "WDRO-MRO provides a decision-centric notion of robustness that naturally connects performance and fairness within a tractable optimization framework". Can they justify this sentence?
> >
> > We appreciate the opportunity to clarify this statement.
> > WDRO–MRO connects robustness and fairness in the following sense:
> >
> > 1. The **minimax regret objective** explicitly controls the gap between the performance of a candidate model and the optimal model under each shifted distribution $Q$. This enforces *decision-centric robustness*: no group or perturbation scenario can incur a large performance regret.
> >
> > 2. The use of **modality weights $\alpha_k$** and group-aware perturbation structures allows us to prioritize robustness for clinically or demographically important subpopulations (e.g., specific tumor sites or noisy imaging modalities), which is a standard notion of distributional fairness.
> >
> > 3. Empirically, we observe that WDRO–MRO improves **worst-group AUC** and reduces performance disparities across groups (e.g., across tumor sites and in-distribution vs. out-of-distribution cohorts), compared to ERM and standard Wasserstein DRO. We will explicitly present and discuss these worst-group metrics in the experimental section to substantiate the fairness claim.
> >
> > In the revision, we will (i) tone down the wording slightly to avoid overclaiming, and (ii) explicitly reference the worst-group / worst-scenario metrics that support this fairness connection.

---

> > > ### Author Response · Authors · 2025-11-27
> > >
> > > ### 5. Summary of planned revisions
> > >
> > > We will:
> > >
> > > - Clarify and slightly weaken Assumption 2.2 (local Lipschitz, no global boundedness).
> > > - Precisely define “closed convex class”, “prediction range”, and “outer objective”.
> > > - Move and rephrase Proposition 2.1 to highlight its central role in the duality framework.
> > > - Add the explicit formula and interpretation of the empirical dual subgradient in Algorithm 1.
> > > - Replace the vague term “implementable” with a precise description of finite-dimensional convex subproblems.
> > > - Explicitly list all methods used in the experiments and specify that **2-Wasserstein** distance is used throughout.
> > > - Provide quantitative justification for the robustness–fairness connection (worst-group performance) and slightly soften the phrasing in the conclusion.
> > >
> > > We hope these clarifications address the reviewer’s concerns and improve the clarity and correctness of the paper.

---

### Official Review · Reviewer_7Dxr · 2025-11-01

**Soundness:** 3
**Presentation:** 3
**Contribution:** 2
**Rating:** 4
**Confidence:** 4

**Summary:**

This paper proposes a framework named Wasserstein Distributionally Robust Minimax Regret Optimization (WDRO-MRO) to address multimodal machine learning problems under distributional uncertainty. The proposed framework extends classical Empirical Risk Minimization (ERM) and standard Distributionally Robust Optimization (DRO), demonstrating significant improvements in performance, robustness, and fairness in practical multimodal scenarios.

From my perspective, WDRO-MRO is conceptually derived from [1] and [2]. The author leverages modality-weighted Wasserstein costs to define an ambiguity set within the Minimax Regret Optimization (MRO) framework [2], which better captures modality-specific heterogeneity. Compared with [1], the author generalizes the formulation to arbitrary convex loss functions and Wasserstein norms, while [1] only considered the type-2 Wasserstein distance. The theoretical contributions are substantial: under mild assumptions, the paper establishes the existence and uniqueness of minimax regret solutions, strong duality, statistical consistency, finite-sample generalization bounds, and an O(N⁻¹ᐟ²) convergence rate. Building on these results, the author presents tractable convex reformulations under different losses and Wasserstein norms, and proposes an oracle-free iterative dual-game algorithm to solve WDRO-MRO efficiently. Moreover, the paper interprets WDRO-MRO as a form of implicit regularization that enhances robustness. Altogether, the theoretical development provides a solid foundation for understanding and accepting this novel framework.

[1] Agarwal, A. & Zhang, T. (2022). Minimax Regret Optimization for Robust Machine Learning under Distribution Shift. arXiv:2202.05436
[2] Al Taha, F., Yan, S., & Bitar, E. (2023). A Distributionally Robust Approach to Regret Optimal Control using the Wasserstein Distance. arXiv:2304.06783

**Strengths:**

- WDRO-MRO is a general and theoretically well-grounded framework applicable to any convex loss function and Wasserstein norm.
- The application to logistic regression is clearly presented, with detailed derivations of the objective and upper bound.
- Experiments on the HANCOCK multimodal dataset show that WDRO-MRO substantially outperforms baseline methods across multiple metrics reflecting performance, robustness, and fairness.

**Weaknesses:**

- The conceptual framework of WDRO-MRO is not entirely new, as similar formulations have appeared in previous works [1, 2]; thus, the paper’s novelty is somewhat limited.
- The HANCOCK dataset may not fully reflect complex multimodal interactions. If I understand correctly, the preprocessing step produces a tabular feature matrix divided into several feature groups, which feels closer to assigning group-wise weights rather than fusing genuinely distinct modalities.
- The experimental comparisons are somewhat limited: the paper only compares with ERM and standard Wasserstein DRO, while many improved DRO variants (e.g., Group DRO, f-DRO, or divergence-based DRO) could provide a more convincing benchmark.
- Regarding modality weighting, WDRO-MRO requires specifying weights αₖ for each modality. In modern multimodal learning, we often fuse information via cross-attention or other adaptive mechanisms that learn inter-modal relevance automatically. It would therefore strengthen the work if the author compared WDRO-MRO with such modern multimodal fusion approaches.

[1] Agarwal, A. & Zhang, T. (2022). Minimax Regret Optimization for Robust Machine Learning under Distribution Shift. arXiv:2202.05436
[2] Al Taha, F., Yan, S., & Bitar, E. (2023). A Distributionally Robust Approach to Regret Optimal Control using the Wasserstein Distance. arXiv:2304.06783

**Questions:**

1. In Algorithm 1, each iteration requires computing sᵢ for all N data points over T iterations, and each sᵢ involves evaluating the Wasserstein distance (O(n³ log n)). Could this lead to computational scalability issues in high dimensions?
2. The paper does not discuss how the modality weights αₖ are chosen. Are they tuned via cross-validation or simply set uniformly?
3. In Section 4.1, the author claims that larger αₖ leads to weaker shrinkage on wₖ. Is this behavior empirically observable in the reported experiments?
4. Since many DRO frameworks focus on distribution shift, do the evaluation metrics used in the paper effectively reflect this? For instance, how well would WDRO-MRO handle domain shifts such as training on one hospital and testing on another in clinical applications?

---

> ### Author Response · Authors · 2025-11-25
>
> ### Response to w1
> Our method is **oracle-free**, which differs from both [1] and [2].
> Although the regret definition involves $\inf_{f' \in \mathcal{F}} R_Q(f')$,
> our dual reformulation removes the need to compute or approximate the oracle predictor. The saddle-point optimisation only relies on the Wasserstein geometry and the loss dual, not on any oracle knowledge of the best predictor under each shifted distribution $Q$.
>
> In contrast, [1] requires an oracle risk minimiser in its theoretical setup, and [2] compares a causal controller with a fully non-causal oracle. These settings are not oracle-free.
> Our multimodal Wasserstein minimax regret framework therefore introduces a new tractable and oracle-free implementable form of regret optimization that does not appear in previous work.
>
> ### Response to w2
> We appreciate the concern regarding the HANCOCK dataset. Although HANCOCK is presented as a tabular matrix after preprocessing, the features in the matrix represent five modalities (demographics, pathology, blood biomarkers, ICD codes, TMA cell density) rather than a single homogeneous feature space. The model receives all five modalities jointly as input to predict the outcome, rather than treating them as group labels as assumed in group-DRO settings.
>
> This setup differs from the group-DRO benchmarks, such as Waterbirds dataset, where all samples are images (single modality) and groups represent subpopulations defined by combinations of bird labels and backgrounds (e.g., landbird/waterbird × land/water background). The aim of group-DRO is to improve the worst group accuracy — for example, correctly classifying a landbird even when it appears in a minority context like a water background.
>
> In contrast, the modalities in HANCOCK arise from separate measurement processes and may degrade independently, so the robustness problem here is cross-modality, not within-modality subgroup correction.
>
> ### Response to w3
> We have added both group-DRO (Section J.3) and an MLP baseline in the revision. Updated results are reported in Figure 1, Figures 4–7 and Table 2, which show that WDRO-MRO outperforms these additional baselines.
>
> ### Response to w4
> Thanks. In the current formulation, the modality weights αₖ are not learned; they act as parameters that define the Wasserstein geometry and control how much the adversary is allowed to perturb each modality. This provides explicit interpretability — a larger αₖ implies greater trust in modality k and weaker shrinkage.
>
> Making αₖ learnable is an interesting direction for future work, enabling the weights to be updated from data during training. This would allow the model to adaptively infer modality reliability rather than relying on fixed prior assignments.
>
> ### Response to Q1 (computational complexity)
> We appreciate the concern. The algorithm **does not solve a full optimal transport problem** in each iteration. The Wasserstein distance only appears **through its dual representation**, so each s_i is obtained by solving a small convex subproblem (e.g., LP, SOCP, or exponential-cone problem depending on the loss), not by computing a transport plan. These subproblems are independent across samples and hence parallelizable.
> Overall, the per-iteration cost is $O(N)$ (same order as standard ERM training), not $O(N^3\log N)$.
>
> ### Response to Q2 (modality weights)
> We thank the reviewer for pointing this out. In our implementation, the modality weights $\alpha_k$ are set uniformly to 1 across all modalities. This corresponds to treating all modalities as equally reliable.
>
> ### Response to Q3 (relation between $\alpha_k$ and $w_k$)
>
> We added an empirical observation of the shrinkage effect in Appendix I.4. The results show that increasing $\alpha_k$ leads to larger $||w_k||^{2}$, which aligns with the behaviour predicted by the term $||w_k||_2^2 / \alpha_k$ in the dual-envelope formulation. This is implemented by varying a single modality weight $\alpha_k \in \{0.25, 0.5, 1.0, 2.0, 4.0, 8.0\}$ while keeping all other modality weights fixed at $1.0$.
>
> ### Response to Q4 (evaluation metrics)
> Yes, the metrics used in the paper align with the metrics in DRO frameworks. `Avg AUC` corresponds to average accuracy, `Robust AUC` is similar to the worst-group accuracy, and `GF Gap(Group-Fairness Gap)` mirrors the group-accuracy gap. The `GNR(Group-Fairness Gap)` metric extends this idea to the noise–group setting by taking the worst AUC over both group and noise levels.
>
> Regarding cross-hospital shifts, it is reasonable to expect that some modalities, such as TMA-derived cell density (e.g., CD3/CD8 immune cell infiltration), are affected by differences in scanners, staining protocols and quantification procedures across hospitals. Even when the underlying biological signal is the same, the measured feature values can vary due to these technical factors. The noise–perturbation experiment (ρ ∈ {0.0, 0.1, …, 0.5}) is designed to simulate this type of modality-level data shift.

---

### Meta-Review · Area_Chair_89eB · 2026-01-08

**Summary:**

All reviewers agree that the paper presents a technically substantial and carefully developed framework that unifies Wasserstein DRO and minimax regret for multimodal learning, with a comprehensive foundation, tractable reformulations, and statistical guarantees. However, the final decision is primarily shaped by concerns regarding **experimental validation and presentation quality**.

**Presentation.** Reviewers EWQY, MJso, and squb all noted that the presentation requires improvement. In particular, Reviewer squb explicitly indicated that a more substantial revision is necessary to adequately address the concerns raised.

**Experiments.** Reviewers MJso and squb pointed out that the experimental evaluation is limited mainly to a single dataset (HANCOCK) and primarily to logistic regression models. This limitation weakens the empirical support for the paper’s claims of general multimodal robustness.

**Reviewer Concerns:**

**Reviewer 7Dxr’s concerns:**

(1) The concern regarding comparisons between the proposed method and prior works [1, 2] was addressed through a detailed discussion clarifying whether the compared methods are oracle-free.

(2) The concern regarding the HANCOCK dataset was addressed by providing more detailed descriptions and clarifications.

(3) The concern regarding the inclusion of additional baselines was addressed by adding both group-DRO and an MLP baseline in the revision.

(4) The concern regarding computational complexity was addressed by clarifying that the algorithm does not solve a full optimal transport problem at each iteration, and that the per-iteration cost is linear and parallelizable.

(5) The concern regarding the choice of modality weights was addressed through further clarification and the inclusion of an ablation study.

(6) The concerns regarding evaluation metrics were addressed through more detailed explanations.

**Reviewer EWQY’s concerns:**

(1) The concerns regarding notation and assumptions were addressed through detailed explanations.

(2) The concerns regarding the algorithms were addressed by providing more detailed descriptions and clarifications.

(3) The concerns regarding the experimental and implementation aspects were addressed by clarifying the baselines used, explaining the role of the 2-Wasserstein distance, and adding Group-DRO as an additional baseline in the revised manuscript.

(4) The concerns regarding the conclusion and the fairness statement were addressed through further clarification.

(5) The remaining outstanding concern relates to novelty. Reviewer EWQY noted that it is unclear how incremental the framework is on some of its contributions, and this concern was not directly addressed in the authors’ response. Another unresolved issue is the overall structural clarity of the manuscript, with Reviewer EWQY explicitly commenting that this is “not a conference article” in style.

**Reviewer squb’s concerns:**

(1) The concern regarding the hyperparameters used in the experiments was addressed by providing a summary of the hyperparameters for all models.

(2) The concern regarding the inclusion of additional baselines was addressed by adding both group-DRO and an MLP baseline in the revision.

(3) The concern regarding nonconvex models was partially addressed by the addition of Remark 3.1, however, no experimental results were provided to validate the authors’ explanation.

(4) The concern regarding the assumptions and the proof sketch of Lemma 3.6 was addressed by introducing Assumption 3.1 and adding a proof sketch in Section 3.4.

(5) The remaining outstanding concern is that a more substantial revision is required to fully address Reviewer squb’s comments, particularly with respect to novelty, extension to nonconvex settings, additional experiments and ablation studies, and overall structural clarity of the manuscript.

**Reviewer MJso’s concerns:**

(1) The concern regarding the fairness metrics across different modalities in experiments was addressed.

(2) The concern regarding the choice of modality weights was addressed through the addition of an ablation study.

(3) The concern regarding nonconvex models was partially addressed by the addition of Remark 3.1, however, no experimental results were provided to validate the authors’ explanation.

(4) The remaining outstanding concern relates to the limited dataset diversity, as noted in the Weaknesses. In particular, the lack of multimodal models and datasets prevents a thorough empirical verification of the effectiveness of the proposed method.

**Reviewer Scores:**

**Reviewer 7Dxr** may increase the score (likely from 4 to 6), as the main concerns raised in the review were adequately addressed in the rebuttal.

**Reviewer EWQY** and **Reviewer MJso** would likely maintain their scores of 6, since several of their concerns were addressed in the rebuttal, while others, particularly those requiring additional experiments, the novelty of the contributions, and the overall structural clarity of the manuscript, would require a more substantial revision and additional time to fully resolve.

**Reviewer squb** would likely maintain a score of 2, as the reviewer explicitly indicated that a more substantial revision is necessary to adequately address the concerns raised.

---

### Decision · Program_Chairs · 2026-01-26

Reject